# Evaluating Semantic Variation in Text-to-Image Synthesis: A Causal Perspective

**Xiangru Zhu**[1][*] **Penglei Sun**[2] **Yaoxian Song**[3] **Yanghua Xiao**[1][†] **Zhixu Li**[4]
**Chengyu Wang**[5] **Jun Huang**[5] **Bei Yang**[5] **Xiaoxiao Xu**[5]

[1]Shanghai Key Laboratory of Data Science, School of Computer Science, Fudan University
[2]Hong Kong University of Science and Technology (Guangzhou)   [3]Zhejiang University
[4]School of Information, Renmin University of China; School of Smart Governance,
Renmin University of China   [5]Alibaba Group

## Abstract

Accurate interpretation and visualization of human instructions are crucial for text-to-image (T2I) synthesis. However, current models struggle to capture semantic variations from word order changes, and existing evaluations, relying on indirect metrics like text-image similarity, fail to reliably assess these challenges. This often obscures poor performance on complex or uncommon linguistic patterns by the focus on frequent word combinations. To address these deficiencies, we propose a novel metric called SemVarEffect and a benchmark named SemVarBench, designed to evaluate the causality between semantic variations in inputs and outputs in T2I synthesis. Semantic variations are achieved through two types of linguistic permutations, while avoiding easily predictable literal variations. Experiments reveal that the CogView-3-Plus and Ideogram 2 performed the best, achieving a score of 0.2/1. Semantic variations in object relations are less understood than attributes, scoring 0.07/1 compared to 0.17-0.19/1. We found that cross-modal alignment in UNet or Transformers plays a crucial role in handling semantic variations, a factor previously overlooked by a focus on textual encoders. Our work establishes an effective evaluation framework that advances the T2I synthesis community's exploration of human instruction understanding.[1]

Input Prompt: A **cat** chasing a **mouse**.

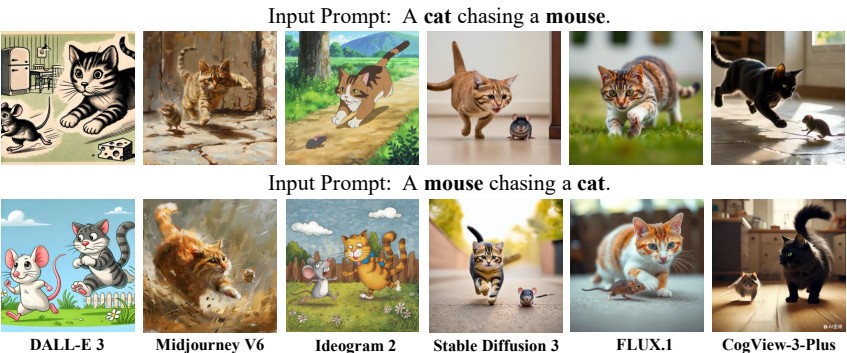

Input Prompt: A **mouse** chasing a **cat**.

DALL-E 3    Midjourney V6    Ideogram 2    Stable Diffusion 3    FLUX.1    CogView-3-Plus

Figure 1: Failed state-of-the-art (SOTA) T2I model examples: different permutations of the same words, different textual semantics, yet similar visual semantics.

## 1 Introduction

Accurately interpreting and visually depicting human instructions is essential for text-to-image (T2I) synthesis ([Cao et al., 2024](#)). Despite advancements in alignment ([Lee et al., 2023a](#); [Wu et al., 2023](#);

---

[*]Email: xrzhu19@fudan.edu.cn
[†]Corresponding author. Email: shawyh@fudan.edu.cn

[1]Our benchmark and code are available at https://github.com/zhuxiangru/SemVarBench.

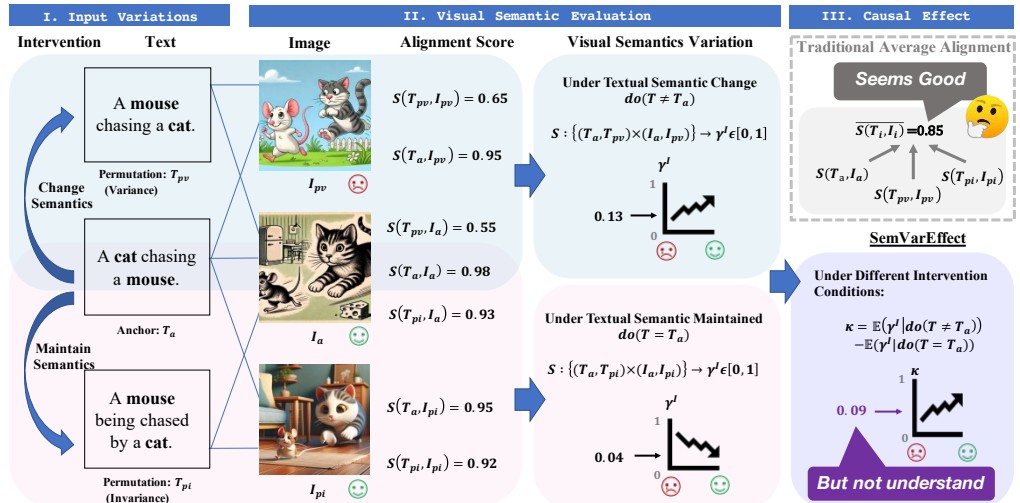

Figure 2: Framework for measuring semantic variation causality in T2I models. Our evaluation consists of three components: (I) **Input Variations** with semantic change/maintenance interventions, (II) **Visual Semantic Evaluation** under both interventions (**blue** for semantic change, **pink** for semantic maintenance), and (III) **Causal Effect Calculation** where SemVarEffect (**purple**) quantifies the difference between intervention outcomes. For Comparison, traditional alignment scores (**gray**) only measure surface similarity, as shown in the cat-mouse example where high alignment coexists with poor semantic consistency. See Section 2 for mathematical details.

Kirstain et al., 2023), composition (Liu et al., 2022; Wang et al., 2024; Li et al., 2024; Feng et al., 2023b), and long instructions (Yang et al., 2024; Gani et al., 2024), these models still treat text prompts as bags of words, failing to depict the semantic variations in human instructions (Yu et al., 2024; Mo et al., 2024). As shown in Fig. 1, existing T2I models generate images with identical semantics, even when the inputs differ semantically (e.g., "a mouse chasing a cat" vs. "a cat chasing a mouse"). This indicates that existing T2I models struggle to accurately capture the semantic variations caused by word orders changes.

There is a lack of direct metric to evaluate a T2I model's ability to understand semantic variations caused by word order changes. Existing NLP research typically evaluates semantic variation indirectly through downstream tasks. For example, in language generation (Gordon et al., 2020), the input sequences with different word orders are used as the actions in a navigation game and the model is evaluated based on the game's accuracy. Similarly, in visual-language understanding (Thrush et al., 2022; Diwan et al., 2022; Yüksekgönül et al., 2023; Wang et al., 2023; Burapacheep et al., 2024), models are evaluated via cross-modal retrieval and image-text matching, focusing on text-image similarity. In T2I synthesis, the text-image alignment score offers an indirect performance measure but may not fully capture a model's sensitivity and robustness to word order. For example, as shown in the upper-right of Fig. 2, an average alignment score of 85, as evaluated by GPT-4, might seem satisfying, but it may conceal the model's proficiency with common word combinations while masking its inadequacy with less frequent or more complex linguistic patterns.

We propose a novel metric, called SemVarEffect, to evaluate the causality of semantic variations between inputs and outputs of T2I models. Our approach uses inputs' semantics as the only intervention to evaluate the average causal effect (ACE) of this intervention on outputs' semantic variations, that is, the contribution of inputs to outputs. A significant ACE would indicate that the T2I model can effectively capture and reflect input semantic variations. On the contrary, a small ACE, such as the 0.09 shown in Fig. 2, exposes a considerable weakness in the T2I model's ability to understand and respond to sentence semantics.

To facilitate the evaluation, we present a new benchmark, called SemVarBench. To avoid overt literal differences, semantic variations are achieved through two types of linguistic permutations (Gerner, 2012): permutation-variance, where different word orders result in different meanings, and permutation-invariance, where the meaning remains unchanged regardless of word orders. Utilizing pre-defined templates and rules as the guidance in the generation stage, followed by a large

amount of annotation and hard sample selection in the validation stage, we constructed a benchmark comprising 11,454 samples, where 10,806 are in the training set and 648 are in the test set. We experimented with a variety of T2I models using our proposed benchmark and metric. The results show that even SOTA models like CogView-3-Plus and Ideogram 2 struggle, achieving scores far from the ideal, which highlights the need for further advancements in handling semantic variations.

Our key contributions are: (1) A first systematic study of semantic variations in T2I synthesis, investigating the causal relationship between input text variations and output images. (2) Sem-VarEffect: A metric quantifying how semantic variations in input text affect T2I output quality. (3) SemVarBench: An expert-annotated benchmark evaluating semantic variations in T2I synthesis through permutation-variance and permutation-invariance tests. (4) Comprehensive evaluation of SOTA T2I models, revealing significant limitations in handling semantic variations and distinct challenges posed by different variation types, while identifying specific areas for improvement.

## 2 SEMANTIC VARIATION EVALUATION FOR TEXT-TO-IMAGE SYNTHESIS

### 2.1 PRELIMINARY

The T2I model $f$ generates images $I$ for each input sentence $T$, represented as $I = f(T)$. $S(T, I)$ is the text-image alignment score, measuring text-image similarity. $S(\cdot)$ represents the scoring method.

**Linguistic Permutation.** Linguistic permutation refers to changes in word order. Given an anchor sentence $T_a$, $T_{pv}$ and $T_{pi}$ are two permutations of $T_a$. $T_{pv}$ exemplifies permutation-variance, which shows a change in meaning, while $T_{pi}$ exemplifies permutation-invariance, where the meaning remains unchanged. The expected $I_{pv}$ is a permutation of objects or relations from $I_a$, while $I_{pi}$ is semantically equivalent to $I_a$, preserving the same visual objects and relations after transformation.

### 2.2 DEFINITION OF VISUAL SEMANTIC VARIATIONS

First, we define the visual semantic variations observed in a single sentence $T$. We decompose complex semantic variation into minimal discrete steps, called localized changes. For each image $I$, the visual semantic variation of a single minimal discrete step $I + \Delta I$, denoted as $\mu_I(T, I)$, is the difference in alignment scores: $\mu_I(T, I) = S(T, I + \Delta I) - S(T, I)$. When the anchor image $I_a$ transforms to a permutation image $I_{p*}$ through these minimal discrete steps, the integrated visual semantic variation can be measured by initial and final states: $\sum_{I_a}^{I_{p*}} \mu_I(T, I) = S(T, I_{p*}) - S(T, I_a)$.

Second, we sum the visual semantic variations from multiple text-image pairs to comprehensively measure these variations. For the sentence $T_a$, the visual semantic variations $\sum_{I_a}^{I_{p*}} \mu(T_a, I)$ demonstrate a shift from a matched to a mismatched image-text pair, indicating a negative change. For the sentence $T_{p*}$, the visual semantic variations $\sum_{I_a}^{I_{p*}} \mu(T_{p*}, I)$ demonstrate a shift from a mismatched to a matched image-text pair, indicating a positive change. To measure the total magnitude of these variations regardless of direction, we use the absolute values. Therefore, the summation of visual semantic variations is defined as $\gamma^I$:

$$\gamma^I = \sum_{T \in \{T_a, T_{p*}\}} \left| \sum_{I_a}^{I_{p*}} \mu(T, I) \right| = |S(T_a, I_{p*}) - S(T_a, I_a)| + |S(T_{p*}, I_{p*}) - S(T_{p*}, I_a)|. \quad (1)$$

### 2.3 THE CAUSALITY BETWEEN TEXTUAL AND VISUAL SEMANTIC VARIATIONS

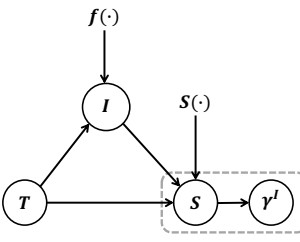

Figure 3: Causal relationship between the input and the output semantic variations.

Fig. 3 illustrates the causal relationship between input and output semantic variations. $T$ is the text input, serving as the input variable, while $I$ is the generated image, acting as a mediator. $S$ is the text-image alignment score, influenced by both $T$ and $I$, and serves as an intermediate result variable. $\gamma^I$ denotes visual semantic variation and is the final comparison result variable. $f(\cdot)$ is an exogenous variable representing a T2I model that maps $T$ to $I$. $S(\cdot)$ is an exogenous variable representing a scoring function that maps $T$ and $I$ to $S$. The dashed line between $S$ and $\gamma^I$ indicates their derived relationship: $\gamma^I$ measures visual semantic variation by summing the absolute differences in alignment scores $S$ between original and permuted text-image pairs.

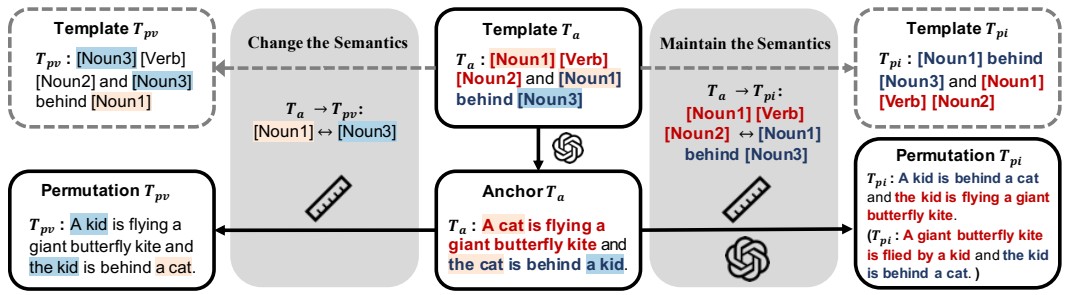

Figure 4: The data collection process of SemVarBench. Top: Templates. Bottom: Generated Sentences. The templates are extracted from the seed pair "a dog is using a wheelchair and the dog is next to a person"/"a person is using a wheelchair and the person is next to a dog".

According to causal inference theory, we define the average causal effect (ACE) of textual semantic variations on visual semantic variations as the SemVarEffect score. It quantifies the influence of input semantic variations on output semantic variations. As shown in Fig. 3, the sentence $T$ serves as an independent variable that influences the generated image $I$. The visual semantics variations is jointly influenced by $T$, $I$ and $S(\cdot)$. Let $\mathrm{do}(T \neq T_a)$ and $\mathrm{do}(T = T_a)$ represent two types of interventions. $\mathrm{do}(T \neq T_a)$ represents an intervention where $T$ differs in meaning from the anchor sentence $T_a$. The visual semantic variation caused by this intervention is denoted as:

$$\gamma_{w/}^I = \mathbb{E}[\gamma^I \mid \mathrm{do}(T \neq T_a)] = \mathbb{E}[\gamma^I \mid T = T_{pv}] = |S(T_a, I_{pv}) - S(T_a, I_a)| + |S(T_{pv}, I_{pv}) - S(T_{pv}, I_a)|. \quad (2)$$

$\mathrm{do}(T = T_a)$ represents an intervention where $T$ match the meaning of the anchor sentence $T_a$. The visual semantic variation caused by this intervention is denoted as:

$$\gamma_{w/o}^I = \mathbb{E}[\gamma^I \mid \mathrm{do}(T = T_a)] = \mathbb{E}[\gamma^I \mid T = T_{pi}], = |S(T_a, I_{pi}) - S(T_a, I_a)| + |S(T_{pi}, I_{pi}) - S(T_{pi}, I_a)|. \quad (3)$$

By comparing visual variations under semantic change interventions and semantic maintenance interventions, we determine the ACE of textual semantic variations:

$$\begin{aligned}
\kappa &= \mathbb{E}[\gamma^I \mid \mathrm{do}(T \neq T_a)] - \mathbb{E}[\gamma^I \mid \mathrm{do}(T = T_a)] = \gamma_{w/}^I - \gamma_{w/o}^I \\
&= |S(T_a, I_{pv}) - S(T_a, I_a)| + |S(T_{pv}, I_{pv}) - S(T_{pv}, I_a)| \\
&\quad - |S(T_a, I_{pi}) - S(T_a, I_a)| - |S(T_{pi}, I_{pi}) - S(T_{pi}, I_a)|,
\end{aligned} \quad (4)$$

The SemVarEffect score $\kappa$ provides theoretically justified bounds (0.5-1.0) for the evaluation. The alignment score $S(\cdot)$ consists of object and relation (triple) components, each contributing up to 0.5 to the total score. $\kappa$ ranges from 0 to 1 under ideal conditions, where $f(\cdot)$ accurately represents the text through images and $S(\cdot)$ faithfully measures text-image alignment. $\kappa > 0.5$ indicates semantically meaningful variations; $0 < \kappa < 0.5$ indicates variations insensitive to semantics; $\kappa \leq 0$ indicates random variations. When no triple remains identical under two interventions, $\kappa$ is maximized, with $\gamma_{w/}^I$ reaching its upper bound and $\gamma_{w/o}^I$ reaching its lower bound. More analysis are in Appendix B.

## 3 SEMANTIC VARIATION DATASET FOR TEXT-TO-IMAGE SYNTHESIS

We create a semantic variation dataset for T2I synthesis through two types of linguistic permutations. In this Section, we first describe the data characteristics and then introduce collection pipeline.

### 3.1 CHARACTERISTICS OF DATA

Each sample $(T_a, T_{pv}, T_{pi})$ consists of three sentences: an anchor sentence $T_a$ and two permutations $T_{pv}$ and $T_{pi}$. They should adhere to the following characteristics:

**Literal Similarity**: $T_a$, $T_{pv}$ and $T_{pi}$ are literally similar, differing only in word order.

**Distinct Semantics**: $T_a$ and $T_{pv}$ have distinct semantics. $T_a$ and $T_{pi}$ share the same semantics.

**Reasonability**: $T_a$, $T_{pv}$ and $T_{pi}$ are semantically reasonable in either the real or fictional world.

**Visualizability**: $T_a$, $T_{pv}$ and $T_{pi}$ describe something humans can visualize. If text is unimaginable to humans, it cannot be meaningfully visualized by T2I models.

**Discrimination**: The images evoked by $T_a$ and $T_{pv}$ present distinguishable differences. The images evoked by $T_a$ and $T_{pi}$ appear similar.

**Recognizability**: The image evoked by $T_a$, $T_{pv}$ and $T_{pi}$ maintain key elements necessary for recognizing typical scenes and characters.

## 3.2 DATA COLLECTION

We use LLMs (GPT-3.5) to generate anchor sentences and their permutations, guided by templates. However, LLMs tend to produce patterns common in their training data, which leads to the neglect of less common combinations specified by templates and rules. To address this issue, we employ a different process for generating $T_a$, $T_{pv}$ and $T_{pi}$.

**Template Acquisition**. We choose all 171 sentence pairs suitable for T2I synthesis from Winoground (Thrush et al., 2022; Diwan et al., 2022) as seed pairs. These pairs are used to extract templates and rules for $T_a$ and $T_{pv}$, while those for $T_{pi}$ are extended manually. To increase diversity, we change the word orders according to the part of speech, including number, adjective, adjective phrase, noun, noun with adjective, noun with clause, noun with verb, noun with prepositional phrase, verb, verb with adverb, adverb, prepositional and prepositional phrase. In Fig. 4, the top left shows an example of templates for $T_a$ and $T_{pv}$ derived from extraction, while the top right shows the corresponding templates for $T_a$ and $T_{pi}$ derived from manual completion.

**Template-guided Generation for $T_a$**. We use LLMs to generate anchor sentences by filling template slots based on prior knowledge and maximum likelihood estimation. In Fig. 4, the bottom middle sentence $T_a$ is generated using the template for $T_a$ as a guide.

**Rule-guided Permutation for $T_{pv}$**. $T_{pv}$ is generated by swapping or rearranging words in $T_a$ based on predefined rules, ensuring that $T_{pv}$ introduces semantic variation. This method avoids a random generation or a semantically equivalent passive structure to $T_a$, which a common pitfall in autonomous generation by LLMs. By following these rules, $T_{pv}$ includes many rare combinations not commonly found in existing NLP corpora. In Fig. 4, $T_{pv}$ is generated by swapping [Noun1] and [Noun3] in $T_a$ (shown in the top left).

**Paraphrasing-guided Permutation for $T_{pi}$**. $T_{pi}$ can be generated by following rules, such as exchanging phrases connected by coordinating conjunctions. However, not all sentences contain coordinating conjunctions, so we also allow other synonymous transformations, including passive voice and slight rephrasing. Both $T_{pi}$ examples in Fig. 4 are acceptable.

## 3.3 DATA ANNOTATION AND STATISTICS

**LLM and Human Annotation**. We establish 14 specific criteria to define what constitutes a "valid" input sample. LLMs check each sample against these criteria, labeling them as "yes" or "no" with confidence scores. Samples labeled "no" with confidence scores above 0.8 are removed. Then, 15 annotators and 3 experts manually verify the remaining samples. Each sample is independently reviewed by two annotators, with an expert resolving any disagreements. This process produced 11,454 valid samples, from which 684 challenging cases are selected for testing based on thresholds and voting. Details are in Appendix C.2.

**Scale and Split**. SemVarBench comprises 11,454 samples of $(T_a, T_{pv}, T_{pi})$, divided into a training set and a test set. The training set contains 10,806 samples, while the test set consists of 648 samples. All evaluations are on the test set.

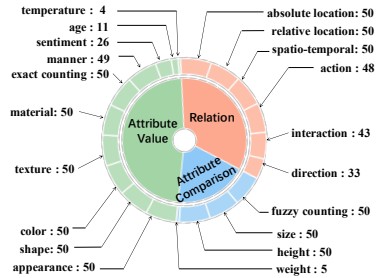

Figure 5: Distribution of semantic variations by category in the semVar-Bench test set.

**Category**. In SemVarBench, samples are divided into 20 categories based on their types of semantic variation. These categories are further classified into three aspects: Relation, Attribute Comparison, and Attribute Values. Fig. 5 shows the distribution of the test set in SemVarBench.

| Model | Abbr. | Type | #DIM | Text Encoder | #TEP | Image Generator | #IGP | Image Decoder | #IDP | #ToP |
|---|---|---|---|---|---|---|---|---|---|---|
| **Open-source Models** | | | | | | | | | | |
| Stable Diffusion v1.5 (Rombach et al., 2022) | SD 1.5 | Diffusion | 768 | CLIP ViT-L | 123.06M | UNet | 859.52M | VAE | 83.65M | 1.07B |
| Stable Diffusion v2.1 (Rombach et al., 2022) | SD 2.1 | Diffusion | 1024 | OpenCLIP ViT-H | 340.39M | UNet | 865.91M | VAE | 83.65M | 1.29B |
| Stable Diffusion XL v1.0 (Podell et al., 2023) | SD XL 1.0 | Diffusion | 2048 | CLIP ViT-L & OpenCLIP ViT-bigG | 123.06M 694.66M | UNet | 4.83B | VAE | 83.65M | 6.51B |
| Stable Cascade (Pernias et al., 2024) | SD CA | Diffusion | 1280 | CLIP ViT-G | 694.66M | UNet | 5.15B | VQGAN | 18.41M | 6.86B |
| DeepFloyd IF XL (Saharia et al., 2022) | DeepFloyd | Diffusion | 4096 | T5-XXL | 4.76B | UNet | 6.02B | VAE | 55.33B | 11.18B |
| PixArt-alpha XL (Chen et al., 2024a) | PixArt | Diffusion | 4096 | Flan-T5-XXL | 4.76B | Transformer | 611.35M | VAE | 83.65M | 5.46B |
| Kolors (Team, 2024) | Kolors | Diffusion | 4096 | ChatGLM3 | 6.24B | UNet | 2.58B | VAE | 83.65M | 8.91B |
| Stable Diffusion 3[medium] (Esser et al., 2024) | SD 3 | Diffusion | 2048 | CLIP ViT-L & OpenCLIP ViT-bigG & T5-XXL | 117.92M 662.48M 4.76B | Transformer | 2.03B | VAE | 83.82M | 7.69B |
| FLUX.1[dev] | FLUX.1 | Diffusion | 768 | CLIP ViT-L & T5-XXL | 123.06M 4.76B | Transformer | 11.90B | VAE | 83.82M | 16.87B |
| **API-based Models** | | | | | | | | | | |
| Midjourney V6 | MidJ V6 | Diffusion | – | – | – | – | – | – | – | – |
| DALL-E 3 (Betker et al., 2023) | DALL-E 3 | Diffusion | – | T5-XXL | 4.76B | UNet | – | VAE | – | – |
| CogView-3-Plus | CogV3-Plus | Diffusion | – | T5-XXL[1] | 4.76B[1] | Transformer | – | VAE[1] | – | – |
| Ideogram 2 | Ideogram 2 | Diffusion | – | – | – | – | – | – | – | – |

[1] The T5-XXL mentioned here is the text encoder of Cogview-3, which is the previous version of Cogview-3-Plus. We have not been able to find specific information about the text encoder and image decoder in the exact materials provided.

Table 1: Mainstream T2I models. #DIM: pooled dimension of text encoders' outputs. #TEP, #IGP, #IDP, #ToP: parameters of text encoders, image generators, image decoders and whole models.

## 4 EXPERIMENTS

### 4.1 EXPERIMENTAL SETUP

**T2I Synthesis Models.** We evaluate 13 mainstream T2I models as shown in Tab. 1. For each sentence, we generate one image, resulting in a total of $684 \times 3 \times 13$ images. Each input prompt is the sentence itself, without any negative prompts or additional details expanded by prompt generators.

**Evaluators.** We use 4 MLLMs as automatic evaluators to calculate text-image alignment scores: Gemini 1.5 Pro, Claude 3.5 Sonnet, GPT-4o and GPT-4 Turbo. Each evaluator is given a sentence and an image and asked to assign two scores: object accuracy (0-50 points) and relation accuracy (0-50 points). The sum of these two scores is taken as the total score, which is then normalized to $[0, 1]$. To validate MLLMs' effectiveness, we conducted human evaluation using three raters on 80 samples (20 each from Midjourney v6, DALL-E 3, CogView3Plus, and Ideogram2) with the same scoring protocol. Then we measure the correlation coefficients of MLLMs with human preferences.

**Metrics.** We use 4 metrics: text-image alignment score ($\bar{S}_{ii}$), our proposed SemVar-Effect ($\kappa$), visual semantic variation under semantic change ($\gamma_{w/}^I$) and maintenance ($\gamma_{w/o}^I$). For each sample, $\bar{S}_{ii} = \frac{1}{|K|} \sum_{i \in K} S(T_i, I_i)$, where $K = \{a, pv, pi\}$. High $\bar{S}_{ii}$, $\gamma_{w/}^I$ and $\kappa$, coupled with low $\gamma_{w/o}^I$, indicate a model's strong causality of input to output. For brevity, we denote $\bar{S}_{ii}$, $\gamma_{w/}^I$, and $\gamma_{w/o}^I$ as $\bar{S}$, $\gamma_w$, and $\gamma_{wo}$.

**Evaluation Dataset.** We evaluate T2I models on the test set in a zero-shot manner. To demonstrate the improvements from fine-tuning, we collected sentences and their generated images from the training set, selecting only those with high quality, high discrimination, and consistent variations as the training data. Details about the selection of the training data are provided in Appendix D.3.

### 4.2 RESULTS

The results of the influence of inputs semantic variations on outputs semantic variations in T2I synthesis are shown in Tab. 2. The scores for $\bar{S}$ range between 0.6 and 0.8. Despite the alignment score $\bar{S}$ reaching up to 0.8, this does not imply a strong grasp of semantics. The following three metrics provide a more comprehensive view of the model's ability to handle semantic variations.

**Visual Semantic Variation with Changed Textual Semantics.** As shown in Tab. 2, the values of $\gamma_w$ are all below 0.52 for all evaluators, significantly lower than the optimal value of 1. This indicates that none of the T2I models perform at an acceptable level. These models are highly insensitive to semantic variations. This finding aligns with the widely accepted notion that T2I models tend to

| Models | Gemini 1.5 Pro | | | | Claude 3.5 Sonnet | | | | GPT-4o | | | | GPT-4 Turbo | | | |
|---|---|---|---|---|---|---|---|---|---|---|---|---|---|---|---|---|
| | $\bar{S}(\uparrow)$ | $\gamma_w(\uparrow)$ | $\gamma_{wo}(\downarrow)$ | $\kappa(\uparrow)$ | $\bar{S}(\uparrow)$ | $\gamma_w(\uparrow)$ | $\gamma_{wo}(\downarrow)$ | $\kappa(\uparrow)$ | $\bar{S}(\uparrow)$ | $\gamma_w(\uparrow)$ | $\gamma_{wo}(\downarrow)$ | $\kappa(\uparrow)$ | $\bar{S}(\uparrow)$ | $\gamma_w(\uparrow)$ | $\gamma_{wo}(\downarrow)$ | $\kappa(\uparrow)$ |
| **Open-source Models** | | | | | | | | | | | | | | | | |
| SD 1.5 | 0.55 | 0.43 | 0.46 | -0.03 | 0.64 | 0.19 | 0.20 | -0.01 | 0.63 | 0.34 | 0.33 | 0.01 | 0.65 | 0.32 | 0.32 | 0.00 |
| SD 2.1 | 0.58 | 0.45 | 0.46 | -0.01 | 0.66 | 0.21 | 0.20 | 0.01 | 0.65 | 0.33 | 0.31 | 0.02 | 0.68 | 0.35 | 0.34 | 0.01 |
| SD XL 1.0 | 0.62 | 0.39 | 0.39 | -0.00 | 0.69 | 0.19 | 0.18 | 0.00 | 0.71 | 0.31 | 0.28 | 0.03 | 0.72 | 0.32 | 0.28 | 0.03 |
| SD CA | 0.59 | 0.42 | 0.41 | 0.01 | 0.69 | 0.19 | 0.18 | 0.01 | 0.67 | 0.31 | 0.31 | -0.00 | 0.69 | 0.32 | 0.31 | 0.01 |
| DeepFloyd | 0.64 | 0.44 | 0.44 | 0.00 | 0.71 | 0.20 | 0.19 | 0.01 | 0.69 | 0.33 | 0.30 | 0.03 | 0.74 | 0.33 | 0.28 | 0.05 |
| PixArt | 0.60 | 0.35 | 0.32 | 0.02 | 0.69 | 0.17 | **0.15** | 0.02 | 0.70 | 0.29 | 0.26 | 0.03 | 0.69 | 0.29 | 0.27 | 0.02 |
| Kolors | 0.60 | 0.41 | 0.42 | -0.01 | 0.69 | 0.22 | 0.22 | -0.01 | 0.69 | 0.31 | 0.30 | 0.01 | 0.69 | 0.33 | 0.30 | 0.02 |
| SD 3 | 0.67 | 0.45 | 0.40 | 0.05 | 0.76 | 0.23 | 0.19 | 0.04 | 0.75 | 0.36 | 0.29 | 0.07 | 0.76 | 0.33 | 0.28 | 0.05 |
| FLUX.1 | 0.72 | 0.43 | 0.35 | 0.08 | 0.75 | 0.23 | 0.17 | 0.06 | 0.72 | 0.42 | 0.33 | 0.10 | 0.75 | 0.40 | 0.30 | 0.10 |
| **API-based Models** | | | | | | | | | | | | | | | | |
| MidJ V6 | 0.68 | 0.46 | 0.39 | 0.07 | 0.73 | 0.24 | 0.21 | 0.03 | 0.72 | 0.40 | 0.33 | 0.07 | 0.73 | 0.38 | 0.32 | 0.06 |
| DALL-E 3 | 0.75 | 0.46 | 0.33 | 0.14 | **0.80** | 0.25 | 0.18 | 0.06 | **0.82** | 0.36 | **0.22** | 0.13 | **0.83** | 0.35 | 0.30 | 0.10 |
| CogV3-Plus | 0.79 | **0.52** | 0.35 | 0.17 | **0.80** | **0.28** | 0.18 | **0.10** | 0.81 | **0.49** | 0.28 | **0.20** | 0.82 | **0.43** | 0.26 | **0.17** |
| Ideogram 2 | **0.80** | 0.47 | **0.29** | **0.18** | 0.79 | 0.26 | 0.17 | 0.09 | 0.81 | 0.46 | 0.27 | **0.20** | 0.81 | 0.40 | **0.24** | 0.15 |

Table 2: Evaluations on T2I models (left column) using semantic variations under multiple MLLM scores (top row). $\bar{S}_{ii}(\uparrow)$: text-image alignment score. $\kappa(\uparrow)$: SemVar-Effect, measuring input-to-output variation contribution. $\gamma_w^I(\uparrow)$ and $\gamma_{wo}^I(\downarrow)$: visual semantic variation under semantic change and maintenance. **Bold** and underline indicate 1st and 2nd optimal cases. Blue and green indicates average SemVarEffect scores between 0.05 and 0.10, and above 0.10, respectively.

treat input text as a collection of isolated words , leading them to interpret sentences with minor changes in word order as having the same meaning.

**Visual Semantic Variation with Unchanged Textual Semantics**. The values of $\gamma_{wo}$ in Tab. 2 are unexpectedly much higher than the optimal value of 0. Only the models highlighted in blue and green demonstrate slightly better performance, with $\gamma_{wo}$ scores consistently lower than $\gamma_w$. These T2I models illustrate potential semantic variations caused by word order through images, yet struggle to differentiate between meaning-variant and meaning-invariant inputs. These models primarily understand language based on word order rather than the underlying semantics.

**Influence of Textual Semantics on Visual Semantic Variations**. In Tab. 2, the $\kappa$ values for all evaluators are below 0.20, indicating considerable room for improvement in T2I models' understanding of semantic variations. Models with higher alignment scores are more sensitive to semantic variations caused by word orders. However, models highlighted in blue overreact to permutations maintaining the meanings, resulting in higher $\gamma_{wo}$ values and subsequently lower $\kappa$ values. These models excel at capturing common alignments but struggles to handle semantic variations.

**Human Evaluation**. We observe consistent performance trends between human raters and the four MLLMs across all evaluated models. Correlation analysis on CogView3-Plus reveals moderate (up to 0.54) correlation coefficients between machine and human scores, suggesting our selected MLLMs can serve as a reliable proxy for human evaluation. Detais are shown in Appendix E.2.

**Comparison of Metrics**. SemVarEffect $\kappa$ focuses on the consistency of variations, while the alignment score $\bar{S}$ focuses on the overall output quality. Thus, a model may score high on $\kappa$ even if it consistently generates incorrect attributes (such as believing oranges are blue and bananas are green). SemVarEffect validates the reliability of high alignment scores: high $\bar{S}$ with low $\kappa$ indicates limited semantic understanding, while high $\bar{S}$ with high $\kappa$ suggests effective semantic understanding.

## 4.3 ANALYSIS

**Is a superior text encoder the exclusive solution for T2I models to grasp semantic variations?** Given differences in text encoders' ability to discriminate semantic variations, we examine whether two metrics, alignment scores $\bar{S}$ and SemVarEffect $\kappa$, offer insights beyond what text encoders capture. We use text similarity[2] to measure the text encoder's ability, and use visual semantic variation scores $\gamma_w$ and $\gamma_{wo}$ to measure SemVarEffect, as illustrated in Fig. 6. PixArt and Kolors, utilizing T5 and ChatGLM as text encoders, fail to transfer the results of distinguishing semantic variations to image generators, as shown by permutation-variance (indicated by squares). However, FLUX.1, utilizing weaker CLIP-T5 hybrid models as text encoders, achieves higher $\bar{S}$ and greater differentiation in $\gamma_w$ and $\gamma_{wo}$ , despite showing minimal changes in text similarity. It indicates that a model's

---

[2]Sentences for changed textual semantics unexpectedly show higher text similarity than those for unchanged textual semantics, likely due to the edit distance between our sentences. For further analysis, see Appendix F.

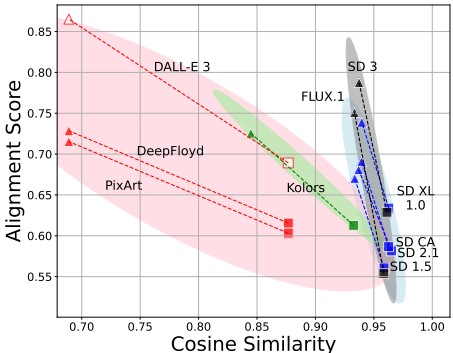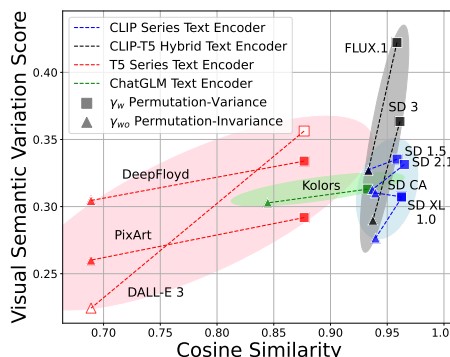

Figure 6: Illustration of the text embedding similarity between the anchor text and the permuted text. Squares represent permutation-variance results (with changed textual semantics), while triangles represent permutation-invariance results (with unchanged textual semantics). The evaluator is GPT-4o. (a) The alignment score between the anchor image $I_a$ and a permutation $T_{p*}$ decreases as the text similarity between $T_a$ and $T_{p*}$ increases. (b) The semantic variation score $\gamma$ increases as the text similarity between $T_a$ and $T_{p*}$ increases. The cosine similarity for DALL-E 3, an API-driven model, is deduced using T5-XXL, indicated by hollow shapes.

ability to distinguish semantic variations is not only dependent on text encoders, and further efforts are needed in cross-modal alignment to effectively transfer these differences to the image generators.

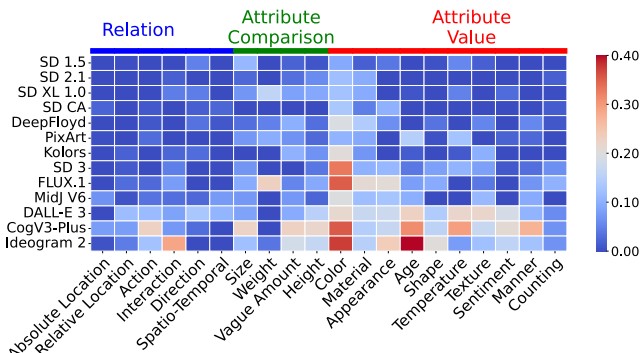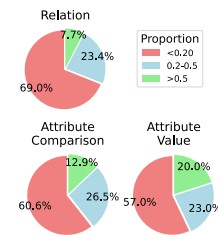

Figure 7: The distribution of categories across different T2I models based on SemVarEffect scores $\kappa$. The evaluator is GPT-4 Turbo.

Figure 8: The distribution of SemVarEffect scores for the SOTA model Ideogram 2 across different aspects of the samples. The evaluator is GPT-4 Turbo.

**Does the influence of input semantic variations on output semantic variations vary by category?** As shown in Fig. 7, the SemVarEffect scores in Color consistently exceed 0.4 in many models, while those in other categories are mostly below 0.1. This suggests that T2I models understand semantic variations well only in the case of Color. We found that the SemVarEffect scores of Ideogram 2 in Relation, Attribute Comparison, and Attribute Value are 0.07, 0.13, and 0.19. To compare the distribution of SemVarEffect scores across different aspects, we set 0.2 and 0.5 as thresholds. As shown in Fig. 8, the proportion of high scores in Attribute Values is significantly higher than those in Relation and Attribute Comparison. T2I models lack the capability to discriminate semantic variations, particularly in aspects emphasizing relations and comparisons. Fig. 9 shows failed examples in Relation and Comparison. Although T2I models can generate correct images for common relations, they tend to rigidly adhere to these common relations even when semantic variations occur, leading to incorrect images. More examples are provided in Appendix G.

**Does fine-tuning improve T2I model performance on semantic variations?** We examine improvements from fine-tuning text encoders and image generators. We use samples in the training set to generated images and select text-images pairs with high alignment scores and high discriminability as training data, details shown in Appendix D.3. As shown in Tab. 3, for categories with sufficient high-quality data, such as Color, supervised fine-tuning (SFT) enhanced the performance

of the T2I model. However, in categories with insufficient high-quality data, such as Direction, SFT led to a decline in performance. Additionally, direct preference optimization (DPO) resulted in performance drops due to failures in permutation-invariance, as evidenced by the increased $r_{wo}$.

| Category | Models | GPT-4o | | | |
|---|---|---|---|---|---|
| | | $\bar{S}(\uparrow)$ | $\gamma_w(\uparrow)$ | $\gamma_{wo}(\downarrow)$ | $\kappa(\uparrow)$ |
| Color | SD XL | 0.73 | 0.33 | 0.25 | 0.08 |
| | + sft-unet | **0.78**(↑) | 0.38(↑) | **0.20**(↓) | **0.18**(↑) |
| | + sft-text | 0.73(−) | 0.40(↑) | 0.27(↑) | 0.13(↑) |
| | + dpo-unet | 0.69(↓) | 0.43(↑) | 0.27(↑) | 0.17(↑) |
| | + dpo-text | 0.68(↓) | **0.47**(↑) | 0.29(↑) | **0.18**(↑) |
| Absolute Location | SD XL | 0.64 | 0.29 | 0.34 | -0.05 |
| | + sft-unet | **0.65**(↑) | **0.34**(↑) | **0.32**(↓) | **0.02**(↑) |
| | + sft-text | 0.64(−) | 0.31(↑) | 0.36(↑) | -0.05(−) |
| | + dpo-unet | 0.60(↓) | 0.29(↑) | **0.31**(↓) | -0.02(↑) |
| | + dpo-text | 0.57(↓) | 0.33(↑) | 0.39(↑) | -0.07(↓) |

| Category | Models | GPT-4o | | | |
|---|---|---|---|---|---|
| | | $\bar{S}(\uparrow)$ | $\gamma_w(\uparrow)$ | $\gamma_{wo}(\downarrow)$ | $\kappa(\uparrow)$ |
| Height | SD XL | **0.77** | 0.34 | **0.23** | **0.10** |
| | + sft-unet | **0.77**(−) | 0.33(↓) | 0.24(↑) | 0.09(↓) |
| | + sft-text | 0.73(↓) | **0.39**(↑) | 0.34(↑) | 0.05(↓) |
| | + dpo-unet | 0.71(↓) | 0.34(−) | 0.33(↑) | 0.02(↓) |
| | + dpo-text | 0.66(↓) | **0.40**(↑) | 0.53(↑) | -0.13(↓) |
| Direction | SD XL | **0.79** | 0.20 | **0.15** | **0.05** |
| | + sft-unet | 0.77(↓) | 0.24(↑) | 0.23(↑) | 0.01(↓) |
| | + sft-text | 0.77(↓) | 0.23(↑) | 0.21(↑) | 0.02(↑) |
| | + dpo-unet | 0.65(↓) | 0.23(↑) | 0.26(↑) | -0.03(↓) |
| | + dpo-text | 0.70(↓) | **0.29**(↑) | 0.27(↑) | 0.01(↓) |

Table 3: Performance of fine-tuned models measured by the SemVarEffect score. Training candidates for Color, Absolute Location, Height, and Direction are 4.4k, 1.7k, 0.2k, and 0.3k.

It is crucial to find a balance between sensitivity and robustness to semantic changes, as this determines whether performance can be enhanced. However, fine-tuning tends to improve sensitivity at the expense of robustness. While T2I models become more sensitive to permutations with different meanings, this discrimination is quickly disrupted by over-sensitivity to permutations with similar meanings, weakening the model's ability to discern differences. This phenomenon may be attributed to two potential limitations in Diffusion-based T2I Models. First, its cross-attention mechanism only maps tokens to visual regions without capturing inter-token relationships. Second, the training process lacks semantic-level supervision. Fine-tuning improves token-region correspondence in Tab. 4, but cannot enhance the understanding of semantic relationships between tokens. Thus, these permutations, which differ only in word order but contain identical tokens, confuse the models and lead to performance declines, especially during DPO. More validation is in Appendix F.4.

| Category | Model | Token Accuracy | | |
|---|---|---|---|---|
| | | $T_a$ | $T_{pv}$ | $T_{pi}$ |
| Absolute Location | SDXL | 0.709 | 0.640 | **0.716** |
| | +sft-unet | **0.718** | **0.654** | **0.716** |
| Height | SDXL | 0.881 | 0.660 | 0.861 |
| | +sft-unet | **0.886** | **0.662** | **0.866** |

Table 4: Performance of fine-tuned models measured by token accuracy on Absolute_Location and Height. Token accuracy refers to the ratio of tokens successfully generated that correspond to images. We filter out meaningful tokens and verify their visual representation in the image using GPT-4o.

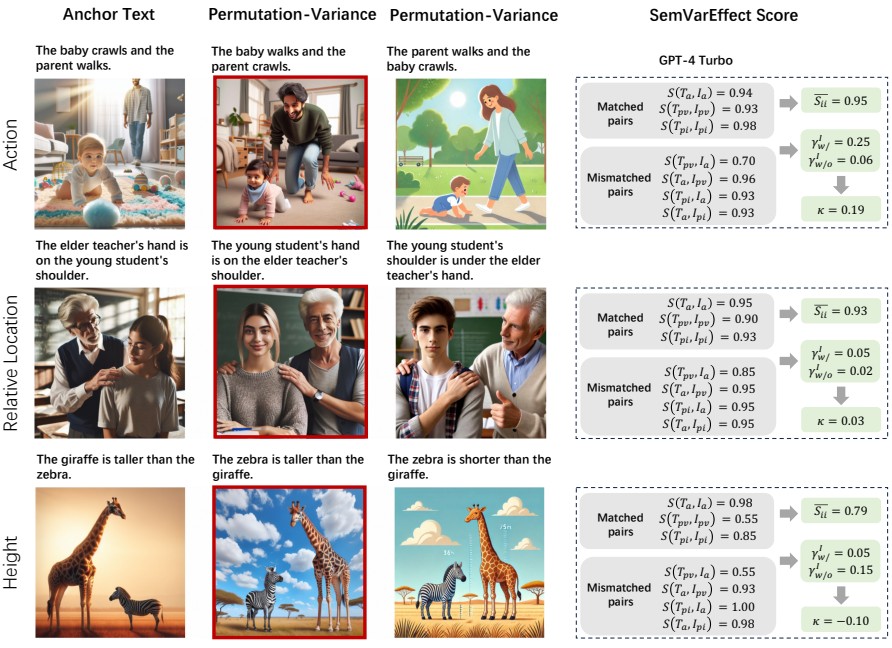

Figure 9: Failed examples of DALL-E 3 on Relation and Attribute Comparison.

**Do T2I models' struggles with semantic relationships stem from training data imbalance?** We conducted experiments testing performance of fine-tuned SDXL trained with balanced training data. We used a human-crafted balanced dataset of cat↔dog chasing interactions (80 images per direction) for fine-tuning, and tested on two unseen prompt pairs involving the same "chasing" relationship between common objects. The model consistently showed poor generalization: accuracy remained low for both anti-commonsense scenarios (mouse↔cat, 3-3/30) and plausible scenarios (bull↔man, 4-5/30). Failure analysis revealed three main categories: (1) Relationship Understanding Failures, where objects appear either without interaction or with incorrect interactions, indicating the model's inability to comprehend the "chasing" concept; (2) Reversed Roles, where the model fails to properly assign who chases whom; and (3) Missing Objects, where the model fails to generate all required objects. Even among the generated images, correct relationships occurred less frequently than these failure cases, suggesting random performance rather than true understanding. Thus, the model's struggles persist even with perfectly balanced training data, suggesting the core issue lies in relationship understanding rather than data imbalance. Details are in Appendix F.3.4.

## 5 RELATED WORK

**Evaluation of T2I synthesis**. Benchmarks of T2I synthesis primarily focus on general alignment (Saharia et al., 2022; Yu et al., 2022; Cho et al., 2023a), composition (Park et al., 2021; Feng et al., 2023a; Park et al., 2021; Hu et al., 2023; Cho et al., 2023b; Li et al., 2024), bias and fairness (Lee et al., 2023b; Luo et al., 2024b;a), common sense (Fu et al., 2024) and creativity (Lee et al., 2023b). In these evaluations, the quality of images is measured by detection-based or alignment-based metrics. Recent research on T2I synthesis has explored samples involving semantic variations caused by word orders, typically using them to evaluate reasoning abilities with alignment-based metrics (Marcus et al., 2022; Lee et al., 2023b; Li et al., 2024). However, a significant gap in this research is the underexplored area of whether the generated images consistently represent fundamental semantic variations within the input text.

**Semantic Variation Evaluation in VLMs.** In VLMs, semantic variations caused by word order has been evaluated by benchmarks like Winoground (Thrush et al., 2022) and its expansion in specific domain (Burapacheep et al., 2024). Winoground is designed to challenge models with visio-linguistic compositional reasoning. It requires models to accurately match two images to their respective captions, where the two captions are different permutations of the same words, resulting in different meanings. To enhance performance on Winoground, studies have focused on expanding training datasets with negative samples and optimizing training strategies to handle the resulting semantic variations (Yüksekgönül et al., 2023; Hsieh et al., 2023; Burapacheep et al., 2024).

The application of Winoground to T2I synthesis faces several limitations due to the variety and quantity of its permutations. First, the dataset, with 400 sentence pairs, provides only 171 suitable for text-image composition analysis (Diwan et al., 2022), where samples are classified into three categories: object, relation, and both. This limited variety is insufficient for a comprehensive evaluation of T2I models. Second, the suitability of certain samples for T2I model evaluation is problematic. Winoground primarily focuses on semantic distinctiveness for cross-modal retrieval (Yüksekgönül et al., 2023; Ma et al., 2023; Cascante-Bonilla et al., 2023). It overlooks the criteria essential for T2I synthesis, such as sentence completeness, clarity of expression, unambiguity, and specificity in referencing image elements. All of these factors have been carefully considered in the quality control of our benchmark annotations.

## 6 CONCLUSION

We comprehensively study the challenge of semantic variations in T2I synthesis, specifically focusing on causality between semantic variations of inputs and outputs. We propose a new metric, SemVarEffect, to quantify the influence of input semantic variations on model outputs, and a novel benchmark, SemVarBench, designed to examine T2I models' understanding of semantic variations. Our experiments reveal that SOTA T2I models struggle with semantic variations, scoring below 0.2 on our benchmark. Fine-tuning shows limited improvement, improving sensitivity but at the cost of robustness. These findings underscore the need for better cross-modal understanding of relation in semantics, particularly for capturing inter-token dependencies in T2I synthesis.

ACKNOWLEDGMENTS

This work is supported by Alibaba Group through Alibaba Innovative Research Program, the Post-doctoral Fellowship Program of CPSF (GZC20232292), the Suzhou Key Laboratory of Artificial Intelligence and Social Governance Technologies (SZS2023007) and the Smart Social Governance Technology and Innovative Application Platform (YZCXPT2023101). We also thank Yunpeng Liu, Yuanyi Xu, and Kejun Xue for their assistance in supplementing the human evaluation experiments during the rebuttal phase.

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

- Section B describes the data requirements, as well as the properties of the text-image alignment function and the SemVarEffect score.
- Section C details the construction process of the benchmark.
- Section D presents the implementation details of the evaluation.
- Section E presents more experimental results.
- Section F provides further analysis of the results.
- Section G visualizes more successful and failed examples in the evaluation.
- Section H discusses the limitations of our evaluation and benchmark.

## A   FOUR TYPES OF SEMANTIC VARIATIONS RESULTS IN T2I SYNTHESIS

The results of semantic variations in T2I synthesis, both in text and images, can be divided into four types, as shown in Fig. 10.

- **Image Changing Semantics with Text Changing Semantics**: The consistency between the input and output in the first quadrant suggests that the model tends to understand the different meanings introduced by linguistic permutations. In this case, the value of semantic variation ($\gamma_{w/}$) tends to approach 1.
- **Image Maintaining Semantics with Text Changing Semantics**: The inconsistency between the input and output in the fourth quadrant suggests that the model does not understand the different meanings introduced by linguistic permutations. In this case, the value of semantic variation ($\gamma_{w/}$) tends to approach 0.
- **Image Changing Semantics with Text Maintaining Semantics**: The consistency between the input and output in the third quadrant suggests that the model tends to understand the similar meanings introduced by linguistic permutations. In this case, the value of semantic variation ($\gamma_{w/o}$) tends to approach 0.
- **Image Maintaining Semantics with Text Maintaining Semantics**: The inconsistency between the input and output in the second quadrant suggests that the model does not understand the similar meanings introduced by linguistic permutations. In this case, the value of semantic variation ($\gamma_{w/o}$) tends to approach 1.

## B   PROPERTIES OF SEMVAREFFECT

### B.1   PRELIMINARY

A T2I generation model $f$ consists of one or more text encoders, image generators, and an image decoder. The T2I generation model $f$ generates images $I \in \mathcal{I}$ for each input textual prompt $T \in \mathcal{T}$. $\mathcal{T}$ represents the textual space, and $\mathcal{I}$ represents the visual space. $S(T, I)$ denotes the alignment score between $T$ and $I$.

Let $T_a$ be an anchor textual prompt. Let $T_{p*}$ represent a permutation of $T_a$, where $T_{pv}$ is a permutation with a meaning different from $T_a$, and $T_{pi}$ is a permutation with the same meaning as $T_a$. Let $I_a$, $I_{pv}$, and $I_{pi}$ be the resulting images generated by a T2I model from $T_a$, $T_{pv}$, and $T_{pi}$, respectively. We expect that $I_{p*}$ will be a rearrangement of the objects or relations found within $I_a$.

### B.2   TEXTUAL VS. VISUAL SEMANTIC VARIATIONS

The measurement of semantic variations in the transition from $(T_a, I_a)$ to $(T_{p*}, I_{p*})$ can be defined from two perspectives: (1) textual semantic variations $r^T$, which refer to the semantic changes in

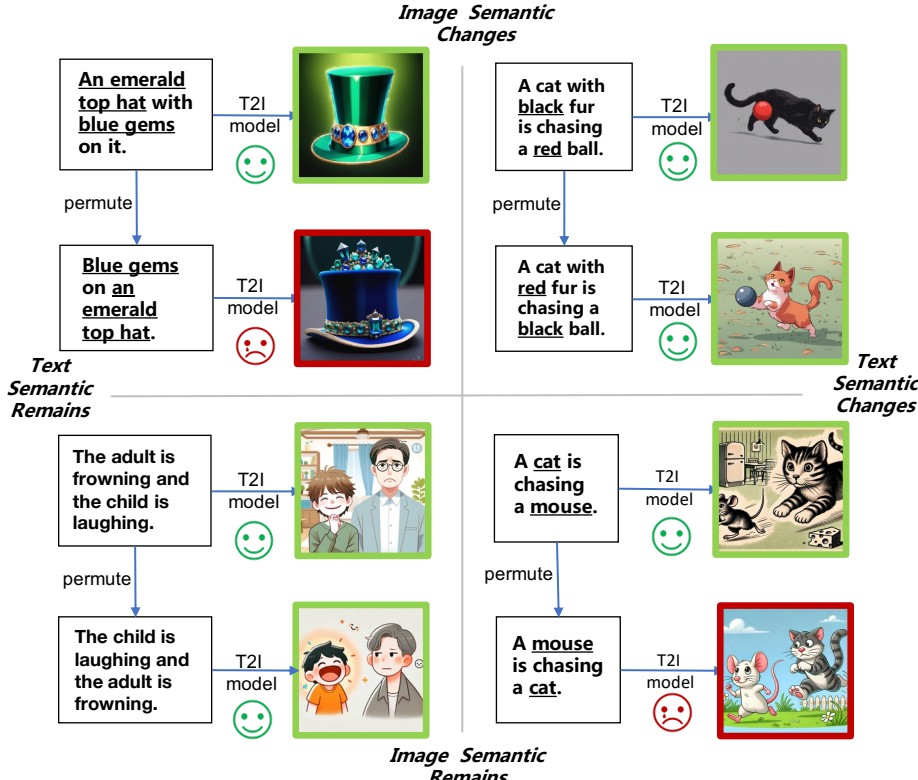

Figure 10: Four types of semantic variation results in T2I synthesis. The images with red borders represent incorrect outputs, while those with green borders represent correct outputs.

texts, observed through images $I_a$ and $I_{p*}$, and (2) visual semantic variations $r^I$, which refer to the semantic changes in images, observed through texts $T_a$ and $T_{p*}$.

Specifically, we define the textual semantic variations observed in a single image $I$. We decompose complex semantic variation into minimal discrete steps, called localized changes. For each sentence $T$, the textual semantic variation of a single minimal discrete step $T + \Delta T$, denoted as $\mu_T(T, I)$, is the difference in alignment scores: $\mu_T(T, I) = S(T + \Delta T, I) - S(T, I)$. When the anchor sentence $T_a$ transforms to a permutation sentence $T_{p*}$ through these minimal discrete steps, the integrated textual semantic variation is: $\sum_{T_a}^{T_{p*}} \mu_T(T, I) = S(T_{p*}, I) - S(T_a, I)$. Therefore, the summation of textual semantic variations is defined as $\gamma^T = \sum_{I \in \{I_a, I_{p*}\}} \left| \sum_{T_a}^{T_{p*}} \mu(T, I) \right|$.

Similarly, we define the visual semantic variation observed in a single sentence $T$. For each image $I$, the visual semantic variation of a single minimal discrete step $I + \Delta I$, denoted as $\mu_I(T, I)$, is the difference in alignment scores: $\mu_I(T, I) = S(T, I + \Delta I) - S(T, I)$. When the anchor image $I_a$ transforms to a permutation image $I_{p*}$ through these minimal discrete steps, the integrated visual semantic variation is: $\sum_{I_a}^{I_{p*}} \mu_I(T, I) = S(T, I_{p*}) - S(T, I_a)$. Therefore, the summation of visual semantic variations is defined as $\gamma^I = \sum_{T \in \{T_a, T_{p*}\}} \left| \sum_{I_a}^{I_{p*}} \mu(T, I) \right|$.

We have not evaluated the influence by measuring the synchronicity of semantic changes between images and text, which has been applied in VLM (Wang et al., 2023). This is because semantic variations introduce a unique challenge in evaluating T2I synthesis: images are not independent; they are influenced by both the input textual prompt and the inherent characteristics of the model, complicating the independent assessment of semantic changes.

Therefore, we conduct the evaluation by measuring the influence of external interventions on semantic variations within the corresponding generated images from T2I synthesis models, while avoiding the direct imposition of interventions on the images.

## B.3 Properties of Alignment Scores $S$

**Definition of Text-Image Alignment Score**. To facilitate semantic analysis, we structured these permutations by objects and triples. Changing the word order affects the arrangement of objects or relations and leads to changes in syntactic dependencies and semantics. Let $T_{p*}$ represent any permutation of $T_a$. $T_a$ and $T_{p*}$ share the same set of objects, denoted as $V$, and the same set of relations, denoted as $R$. The triple set $E$ in $T_a$ is a subset of $V \times R \times V$. Some triples in $T_{p*}$ may differ from those in $T_a$, but they have the same number of triples. For example, the initial triple set of $T_a$ contains (apple, on, box), (girl, touch, apple) and (girl, NULL, box). After swapping box and apple, the triple set of $T_{p*}$ contains (box, on, apple), (girl, touch, box) and (girl, NULL, apple).

We define the alignment score $S$ between $T$ and $I$ as the sum of fine-grained alignment scores for objects and triples:

$$S(T, I) = \sum_{i=1}^{|V|} S_{obj_i}(T, I) + \sum_{j=1}^{|E|} S_{tri_j}(T, I), \tag{5}$$

where $|V|$ is the number of objects in $T$ and $|E|$ is the number of triples in $T$. The components of the alignment score are defined as piecewise functions:

$$S_{obj_i}(T, I) = \begin{cases} w_{v_i} & \text{if the } i\text{-th object matches,} \\ 0 & \text{if the } i\text{-th object does not match,} \end{cases}$$

$$\tag{6}$$

$$S_{tri_j}(T, I) = \begin{cases} w_{e_j} & \text{if the } j\text{-th triple matches,} \\ 0 & \text{if the } j\text{-th triple does not match,} \end{cases}$$

where $w_{v_i}$ and $w_{e_j}$ are the weighted matching scores for the $i$-th object and $j$-th triple, respectively. We obtain the alignment function $S$ that satisfies the constraints in Eq. 10. Consequently, the alignment score of a matched text-image pair is calculated as follows:

$$S(T_{p*}, I_{p*}) = S(T_a, I_a) = \sum_{v_i \in V_{MA}} w_{v_i} + \sum_{e_j \in E_{MA}} w_{e_j}, \tag{7}$$

where $V_{MA}$ and $E_{MA}$ represent the exactly matched objects and triples between a text prompt and its generated image, with $|V_{MA}| = |V|$ and $|E_{MA}| = |E|$, respectively. The alignment score for a mismatched text-image pair is calculated as follows:

$$S(T_{p*}, I_a) = S(T_a, I_{p*}) = \sum_{v_i \in V_{MI}} w_{v_i} + \sum_{e_j \in E_{MI}} w_{e_j}, \tag{8}$$

where $V_{MI}$ and $E_{MI}$ represent the partially matched objects and triples between a text prompt and a mismatched image, with $|V_{MI}| = |V|$ and $0 \leq |E_{MI}| \leq |E|$.

**Range of $S$**. If $f$ accurately depicts the text through images and $S$ faithfully measures the semantic changes between text space and the image space, then any alignment score $S(T, I)$ is bounded by:

$$\sum_{v_i \in V} w_{v_i} \leq S(T, I) \leq \sum_{v_i \in V} w_{v_i} + \sum_{e_j \in E} w_{e_j}. \tag{9}$$

In our implementation, we set the value of $S(T, I)$ as an integer between 0 and 100, where the object accuracy ranges from 0 and 50 and the triple accuracy ranges from 0 and 50. We then normalize it into a real number within the range $[0, 1]$. Based on the assumption of $f$ mentioned above, we have $0.5 \leq S(T, I) \leq 1$.

However, limitations in the capabilities of the model $f(\cdot)$ and the alignment function $S(\cdot)$, often prevent the alignment score values from achieving the property outlined in Eq. 10. For example, if a model $f$ generates a low-quality image $I$, it may fail to accurately depict all target objects ($V$), leading to $|V_{MA}| < |V|$ and $|V_{MI}| < |V|$. This results in object accuracy below 0.5 (as illustrated in the bottom case of Fig. 25) and inconsistent relation accuracy (see cases in Fig. 20 and Fig.

21). Furthermore, inaccuracies in the scoring approach $S(\cdot)$ may incorrectly assess the similarity between text prompts and generated images, causing unpredictable fluctuations in semantic variation measurements (as illustrated in Fig. 27).

**Identity Relation for $S$.** Under ideal conditions where $f$ accurately transforms all semantic variations from text space to image space, the alignment scores would satisfy the following constraints:

$$S(T_a, I_a) \equiv S(T_{p*}, I_{p*}) \text{ and } S(T_{p*}, I_a) \equiv S(T_a, I_{p*}). \tag{10}$$

Eq. 10 is also demonstrated under the assumption that the alignment function is an equivariant map in the continuous textual feature space $\mathcal{T}$ and visual feature space $\mathcal{I}$, as detailed in (Wang et al., 2023). This assumption ensures that the alignment scores vary consistently with the semantic changes in images or text. This property is crucial for determining the characteristics of the data in SemVarBench and for designing the alignment functions.

However, the low-quality image $I$, resulting from a poorly performing model $f$, and the inaccuracies in the scoring approach $S$, often prevent the alignment scores from meeting the criteria in Eq. 10. These limitations make the direct comparison of textual and visual semantic variation scores unreliable in T2I synthesis. This approach was previously explored in (Wang et al., 2023).

## B.4 PROPERTY OF VISUAL SEMANTIC VARIATIONS $\gamma^I$

We analyze the theoretical relationship between visual semantic variations and model performance. According to Eqs. 1, 7 and 8, we derive the visual semantic variations as follows:

$$\gamma^I = 2 \left| \sum_{v_i \in \{V_{MA} - V_{MI}\}} w_{v_i} + \sum_{e_j \in \{E_{MA} - E_{MI}\}} w_{e_j} \right|. \tag{11}$$

For both permutation-variance and permutation-invariance, the value of $\sum_{v_i \in \{V_{MA} - V_{MI}\}} w_{v_i}$ remains constant, which we denote as $C_1$. As a result, the visual semantic variation can be simplified to $\gamma^I = 2 \left| C_1 + \sum_{e_j \in \{E_{MA} - E_{MI}\}} w_{e_j} \right|$, primarily depending on the size of the set $E_{MA} - E_{MI}$. However, the size of the set varies dramatically between the two settings:

- In permutation-variance settings $(T_a, T_{pv})$, the optimal value for the set $E_{MA} - E_{MI}$ is its maximum set $E$, resulting in an obvious positive correlation between visual semantic variation $\gamma^I_{w/}$ and model performance.

- In permutation-invariance settings $(T_a, T_{pi})$, the optimal value for the set $E_{MA} - E_{MI}$ is its minimum set $\emptyset$, resulting in an obvious negative correlation between visual semantic variation $\gamma^I_{w/o}$ and model performance.

Therefore, we conclude that visual semantic variations in permutation-variance and permutation-invariance settings differ significantly.

- A higher $\gamma_{w/}$ value indicates that the model effectively captures and reflects the intended semantic variations in the input text.

- A lower $\gamma_{w/o}$ value indicates that the model maintains semantic consistency in the images despite variations in the input text.

## B.5 PROPERTY OF SEMVAREFFECT SCORE $\kappa$

The SemVarEffect score $\kappa$, is defined as the difference between two types of visual semantic variations score $\gamma^I_{w/}$ and $\gamma^I_{w/o}$. It quantifies the model's ability to discriminate between significant and negligible semantic changes in the text. Under ideal conditions, $\kappa$ ranges from 0 to 1. $\kappa$ is maximized when $\gamma^I_{w/}$ reaches its upper bound of 1, which occurs in extreme cases where no relation between objects are identical, and $\gamma^I_{w/o}$ reaches its optimal value of 0.

- If $\kappa$ is large, it suggests that the model is sensitive to semantic changes, recognizing variations in meaning. However, this does not necessarily imply strong alignment. The model might detect changes in semantics but still struggle to fully capture all objects and relationships described in the text, indicating a gap between sensitivity and complete alignment.

- If $\kappa$ is small or close to zero, it suggests that the model either fails to reflect meaningful semantic changes or overreacts to minor text variations. Regardless of the overall alignment score, the model may generate similar images even in the presence of significant semantic differences in the input text.

# C  CONSTRUCTION DETAILS

## C.1  DATA COLLECTION

| Seed Sentence Pairs from Winoground | Templates & Rule |
|---|---|
| *caption_0*: a bird eats a snake
*caption_1*: a snake eats a bird | $T_a$: [Noun1] [Verb (vt)] [Noun2]
$T_{pv}$: [Noun2] [Verb (vt)] [Noun1]
$T_a \rightarrow T_{pv}$: [Noun1] $\leftrightarrow$ [Noun2] |
| *caption_0*: a person is in a helicopter which is in a car
*caption_1*: a person is in a car which is in a helicopter | $T_a$: [Noun1] [Verb (vi)] [Prepositional Phrase1 (location)] which is in [Prepositional Phrase2 (location)]
$T_{pv}$: [Noun1] [Verb (vi)] [Prepositional Phrase2 (location)] which is in [Prepositional Phrase1 (location)]
$T_a \rightarrow T_{pv}$: [Prepositional Phrase1 (location)] $\leftrightarrow$ [Prepositional Phrase2 (location)] |
| *caption_0*: there are some pineapples in boxes, and far more pineapples than boxes
*caption_1*: there are some boxes containing pineapples, and far more boxes than pineapples | $T_a$: ([Prepositional Phrase1 (location)], )(There be)[Noun1] [locate in] [Noun2], and far more [Noun1] than [Noun2]
$T_{pv}$: ([Prepositional Phrase1 (location)], )(There be)[Noun2] [contain] [Noun1], and far more [Noun2] than [Noun1]
$T_a \rightarrow T_{pv}$: [Noun1] $\leftrightarrow$ [Noun2] |
| *caption_0*: the person sitting down is supporting the person standing up
*caption_1*: the person standing up is supporting the person sitting down | $T_a$: [Noun1] (which) [Verb1 (vi)] [Verb (vt)] [Noun2] (which) [Verb2 (vi)]
$T_{pv}$: [Noun1] (which) [Verb2 (vi)] [Verb (vt)] [Noun2] (which) [Verb1 (vi)]
$T_a \rightarrow T_{pv}$: [Verb1 (vi)] $\leftrightarrow$ [Verb2 (vi)] |
| *caption_0*: the person with green legs is running quite slowly and the red legged one runs faster
*caption_1*: the person with green legs is running faster and the red legged one runs quite slowly | $T_a$: [Noun1] [Prepositional Phrase1/Relative Clause1 (appearance)] [Verb1 (vi)] slowly and [Noun2] [Prepositional Phrase2/Relative Clause2 (appearance)] [Verb2 (vi)] faster
$T_{pv}$: [Noun1] [Prepositional Phrase1/Relative Clause1 (appearance)] [Verb1 (vi)] faster and [Noun2] [Prepositional Phrase2/Relative Clause2 (appearance)] [Verb2 (vi)] slowly
$T_a \rightarrow T_{pv}$: slowly $\leftrightarrow$ faster |

Table 5: Examples of extracted templates and transformation rules between templates of $(T_a, T_{pv})$.

**Template Acquisition** We designate 171 compositional cases in Winoground (Thrush et al., 2022), labeled as "no-tag" in subsequent research (Diwan et al., 2022), and refer to them as SEED$_0$ and SEED$_1$. The template of $T_{pv}$, the permutation with semantic changes, is extracted from each pair of seeds by human annotators. Then, we define the rule of $T_{pi}$, which is the permutation without semantic changes, as the original template of $T_{pi}$. An example is illustrated as follows.

$T_a$: [Noun1] [Verb] [Noun2] and [Noun1] behind [Noun3]
$T_{pv}$: [Noun3] [Verb] [Noun2] and [Noun3] behind [Noun1]
$T_{pi}$: [Noun1] behind [Noun3] and [Noun1] [Verb] [Noun2]

$T_a \rightarrow T_{pv}$: [Noun1] $\leftrightarrow$ [Noun3]
$T_a \rightarrow T_{pi}$: [Noun1] [Verb] [Noun2] $\leftrightarrow$ [Noun1] behind [Noun3]

If there is no coordinating conjunction such as *and* or *while* in the template of $T_{pi}$, the template can be set to *NULL*. In this case, the permutation $T_{pi}$ will be generated based on the LLM according to other solutions.

$T_a$: [Noun1] [Verb1 (vi)] [Verb (vt)] [Noun2] [Verb2 (vi)]
$T_{pv}$: [Noun1] [Verb2 (vi)] [Verb (vt)] [Noun2] [Verb1 (vi)]
$T_{pi}$: NULL

$T_a \rightarrow T_{pv}$: [Verb1 (vi)] $\leftrightarrow$ [Verb2 (vi)]
$T_a \rightarrow T_{pi}$: NULL

**Template-guided Generation for** $T_a$. The prompt for generating $T_a$, which is guided by the templates and seed pairs, is as follows:

---

Assuming you are a linguist, you have the ability to create a similar sentence following the structure of given sentences.

The given two sentences are {SEED$_0$} and {SEED$_1$}. The structure of them are both "{Template $T_a$}". Please create a similar "{Template $T_a$}" sentence as "TEXT0", and diversify your sentence as much as possible by using different themes, scenes, objects, predicate, verbs, and modifiers.

Output a list containing {NUM} json objects that contain the following keys: TEXT0. Use double quotes instead of single quotes for key and value. Now, let's start. The output json object list:

---

**Rule-guided Permutation for** $T_{pv}$. The prompt for generating $T_{pv}$ based on $T_a$ and its corresponding rule, is:

---

Assuming you are a linguist, you have the ability to judge the structure of existing sentences and imitate more new sentences with similar structure but varied content.

Step 1: Input some sentences structured by {Template $T_a$} and {Template $T_{pv}$}. We call each sentence as "TEXT0".
Step 2: For each "TEXT0", perform the change which is "{RULE of $T_a \rightarrow T_{pv}$}" and keep the other words unchanged as "TEXT1".

For example, TEXT0={TEXT0}. Only swap/move {RULE of $T_a \rightarrow T_{pv}$} and keep the other words unchanged to generate TEXT1={TEXT1}.

Output a list containing {NUM} json objects that contain the following keys: TEXT0, TEXT1. Use double quotes instead of single quotes for key and value. Now, let's start. The input is: {TEXT0}. The output json object list:

---

**Paraphrasing-guided Permutation for** $T_{pi}$. The prompt for generating $T_{pi}$ based on $T_a$ and its corresponding rule, is:

---

[Instruction]
Please generate a sentence that has a similar length and meaning in the following six ways:
1. Change the word order: For example, "a red and yellow dog" can be changed to "a yellow and red dog." In some languages, adjusting the order of words in a sentence can create a new sentence form without changing the meaning. For instance, "I like you" can be adjusted to "You are the person I like".
2. Passive voice: For example, "a kid is flying a yellow kite" can be changed to "a yellow kite is being flown by a kid."
3. Change the description: For example, "a boy is playing with a girl" can be changed by paraphrasing and altering the sentence structure to "a boy is playing. He is near a girl."
4. Use synonyms: Replace words in the sentence with their synonyms. For example, "happy" can be replaced with "joyful".
5. Use infinitive or gerund forms: For example, "He likes to run" can be changed to "He enjoys running".
6. Simplify or expand: You can either simplify the sentence structure or add additional information to create a new sentence. For example, "The quick, brown fox jumps over the lazy dog" can be simplified to "The fox jumps over the dog", or expanded to "The fox, which is quick and brown, jumps over the lazy dog".

Now, please generate a similar sentence for input prompt given at the end. Provide one sentence for each of the six methods. If a sentence cannot be generated using a particular method, please output "None".
Add the results as a list of JSON objects, containing 6 JSON objects. Each object should include the keys: number, modification method, and sentence.

[Prompt]
"{TEXT0}"

---

## C.2   DATA ANNOTATION

**Criteria for Valid Samples**. The primary challenge in annotation lies in defining the criteria for what qualifies as "valid". For T2I synthesis models, we define "valid" input text based on 14 specific criteria. First, we illustrate these criteria through examples of $T_a$ and $T_{pv}$. Second, for $T_{pi}$, we require it to apply one of the six synonymous transformations defined in the prompt for generating

| Type | Valid Criteria | Example | ✓/✗ |
|---|---|---|---|
| Basic | Complete Expression | $T_a$: Swinging on the swing and off the metal chains. | ✗ |
| | | $T_{pv}$: Swinging off the swing and on the metal chains. | ✗ |
| | | $T_{pi}$: Swinging off the metal chains and on the swing. | ✗ |
| | Clear and Concrete Objects | $T_a$: A brighter sun is shining on a dimmer object. | ✗ |
| | | $T_{pv}$: A dimmer sun is shining on a brighter object. | ✗ |
| | | $T_{pi}$: A dimmer object is shined on by a brighter sun. | ✗ |
| | Reasonable Semantics | $T_a$: An engineer builds a bridge. | ✓ |
| | | $T_{pv}$: A bridge builds an engineer. | ✗ |
| | | $T_{pi}$: A bridge is built by an engineer. | ✓ |
| Visualizable | Visually Depicted Elements | $T_a$: There are more salads than burgers on the menu. | ✗ |
| | | $T_{pv}$: There are more burgers than salads on the menu. | ✗ |
| | | $T_{pi}$: There are less burgers than salads on the menu. | ✗ |
| | Static Scene or Multiple Exposure Scene | $T_a$: The wave is moving faster and the fish is swimming slowly. | ✗ |
| | | $T_{pv}$: The fish is swimming faster and the wave is moving slowly. | ✗ |
| | | $T_{pi}$: The fish is swimming slowly and the wave is moving faster. | ✗ |
| | Moderate Details | $T_a$: In the library, there are a stack of books and some more magazines. | ✗ |
| | | $T_{pv}$: In the library, there are a stack of magazine and some more books. | ✗ |
| | | $T_{pi}$: In the library, there are some more magazines and a stack of books. | ✗ |
| | Quantifiable Comparison | $T_a$: There are more ants than bees in the garden. | ✗ |
| | | $T_{pv}$: There are more bees than ants in the garden. | ✗ |
| | | $T_{pi}$: There are less bees than ants in the garden. | ✗ |
| Discriminative | Modification Rules | $T_a$: A sharp knife is on a dull cutting board. | ✗ |
| | | $T_{pv}$: A dull cutting board is under a sharp knife. | ✗ |
| | | $T_{pi}$: A dull cutting board is under a sharp knife. | ✗ |
| | Distinct Textual Semantics | $T_a$: The boat is on the dock and the fisherman is on the pier. | ✗ |
| | | $T_{pv}$: The boat is on the pier and the fisherman is on the dock. | ✗ |
| | | $T_{pi}$: The fisherman is on the pier and the boat is on the dock. | ✗ |
| | Visually Distinguishable | $T_a$: There's a delicious chocolate cake with a bitter coffee frosting. | ✗ |
| | | $T_{pv}$: There's a bitter chocolate cake with a delicious coffee frosting. | ✗ |
| | | $T_{pi}$: There's a bitter coffee frosting with a delicious chocolate cake. | ✗ |
| Recognizable | Item-Specific Scene | $T_a$: There are more books than shelves in this library. | ✓ |
| | | $T_{pv}$: There are more shelves than books in this library. | ✗ |
| | | $T_{pi}$: There are less shelves than books in this library. | ✓ |
| | Item-Specific Character | $T_a$: A photographer wearing a camera strap with his lens in the air and a videographer wearing a tripod. | ✗ |
| | | $T_{pv}$: A photographer wearing a tripod with his lens in the air and a videographer wearing a camera strap. | ✗ |
| | | $T_{pi}$: A videographer wearing a tripod and a photographer wearing a camera strap with his lens in the air. | ✗ |
| | Attire-based Character | $T_a$: The soldier in the barracks is cleaning equipment and the officer in the office is reviewing reports. | ✗ |
| | | $T_{pv}$: The soldier in the barracks is reviewing reports and the officer in the office is cleaning equipment. | ✗ |
| | | $T_{pi}$: The officer in the office is reviewing reports and the soldier in the barracks is cleaning equipment. | ✗ |
| | Action-based Character | $T_a$: The businessman is wearing navy suit and red tie. | ✗ |
| | | $T_{pv}$: The businessman is wearing red suit and navy tie. | ✗ |
| | | $T_{pi}$: The businessman is wearing red tie and navy suit. | ✗ |

Table 6: Error Examples of LLM-generated permutation-based sentences ($T_a$, $T_{pv}$, $T_{pi}$) and the criteria they violate.

$T_{pi}$. From a semantic perspective, $T_{pi}$ must be strictly consistent with $T_a$, ensuring the consistency and accuracy of the entire dataset. We list these criteria and examples in the following.

- Basic
  - Complete Expression: Both sentences should be complete and free of obvious linguistic errors.
  - Clear and Concrete Objects: Both sentences must be clear and unambiguous, either contextually or inherently, and specifically describe tangible objects, avoiding abstract concepts.
  - Meaningful Sentence: Both sentences must maintain logical coherence within their respective contexts. The reasonable definition includes real-world plausibility or scenarios typically seen as implausible in virtual or imaginative settings (like children's literature, animations, or science fiction), like flying pigs or dinosaurs piloting planes. For example, "a shorter person can reach a higher shelf while a taller one cannot" is not reasonable in any world.

- Visualizable

- Visually Depicted Element: Both sentences must convey visual elements, including objects, scenes, actions, and attributes, ensuring that the text prompts are visually depictable and that the image content is identifiable during evaluation.
- Static Scene or Multiple Exposure Scene: Both sentences should be visually representable through images alone, without the need for video, audio, or other sensory inputs such as touch and smell. Temporal aspects, procedures, and comparisons in test cases must be conveyable within the scope of a single image.
- A Moderate Level of Detail: Sentences should maintain a moderate level of detail with similar scales for objects and scenes. Excessive or mismatched scales can result in sentences that are challenging to depict. For example, comparing the quantity of books and magazines "in a library" is less suitable than comparing them "on a table".
- Quantifiable Comparison: Comparisons in both sentences should be quantifiable, using measures like counts, areas, or volumes. For example, "There are more students in the classroom than words on the blackboard" is difficult to compare quantitatively.

- Discriminative
  - Following Permutation Rules: Generated samples $T_{pv}$ must strictly follow the designated manual template, including word swapping and movement.
  - Distinct Textual Semantics: Two sentences must have distinct textual semantics. Otherwise, the pairs are considered invalid.
  - Visually Distinguishable: Two sentences should be visually distinct, with a clear differentiation in the visual characteristics of the objects or scenes described. Subtle differences that require very close observation are not considered distinct visual differences.

- Recognizable
  - Item-Specific Scenes: Scenes in sentences should be identifiable, with key elements maintained for recognition. Otherwise, identification may be challenging. For instance, a sentence describing a `library` where "bookshelves outnumber books" might be unrecognizable, as we typically expect a library to contain many books.
  - Item-Specific Characters: When a sentence depicts a character through associations with specific items, these items or behaviors should remain consistent for easy identification. Otherwise, the character may be hard to recognize. For instance, `chefs` are usually associated with "chef's attire, cooking utensils, and kitchens".
  - Attire-Based Characters: When a sentence presents characters identifiable by their attire, such as `firefighters`, `police officers`, `soldiers`, `doctors`, and `nurses`, their clothing should remain consistent for clear recognition. Changes in attire could obscure their identities.
  - Action-Based Characters: When a sentence features characters defined by specific actions or interactions, such as `bartenders` (mixing drinks), `businessmen` (negotiating), `journalists` (interviewing), `divers` (deep-sea diving), their typical activities should remain consistent. Altering distinctive features or placing characters in unusual scenarios may obscure their identities.

**Automatic Annotation**. We employ a machine-human hybrid verification process to filter out invalid samples that violate any characteristic. We use LLMs to judge whether each sample violates any of the specific criteria, labeling them "yes" or "no" and providing confidence scores. The samples whose confidence exceeds a threshold of 0.8 are removed from the dataset. We initially collected 48K samples, each including 3 sentences. The automatic filtering helped eliminate over 42% of them, resulting in a final corpus of 27K samples.

**Human Annotation**. We use 15 annotators and 3 experienced experts to manually verify the samples. All annotators have linguistic knowledge and are provided with detailed annotation guidelines. Each sample is independently annotated by two annotators. Then, an experienced expert reviews the controversial annotations and makes the final decision. After annotation, we randomly sampled 100 valid samples to assess annotation accuracy. Two experts evaluated that 99% of the samples were valid. Finally, we obtained 11,479 valid, non-duplicated samples.

**Hard Samples Selection**. To effectively evaluate T2I models, it is crucial to select challenging samples rather than simple ones. Initially, we generate images using SOTA models like DALL-E 3,

| Category | Train | Test | Total |
|---|---|---|---|
| **Relation** | | | |
| Absolute Location | 1,716 | 50 | 1,766 |
| Relative Location | 1,111 | 50 | 1,161 |
| Action | 216 | 48 | 264 |
| Interaction | 153 | 43 | 196 |
| Direction | 342 | 33 | 375 |
| Spatio-temporal | 234 | 50 | 284 |
| **Attribute Comparison** | | | |
| Vague amount | 1,839 | 50 | 1,889 |
| Size | 2,168 | 50 | 3,118 |
| Height | 253 | 50 | 303 |
| Weight | 5 | 5 | 10 |
| **Attribute Value** | | | |
| Color | 4,451 | 50 | 4,501 |
| Appearance | 1,972 | 50 | 2,022 |
| Texture | 542 | 50 | 592 |
| Shape | 190 | 50 | 240 |
| Size | 516 | 50 | 566 |
| Material | 227 | 50 | 277 |
| Manner | 194 | 49 | 243 |
| Sentiment | 88 | 26 | 114 |
| Age | 22 | 11 | 33 |
| Temperature | 14 | 4 | 18 |
| Counting | 614 | 50 | 664 |
| **Total** | **15,518** | **819** | **14,699** |
| **Total(deduplication)** | **11,454** | **684** | **10,770** |

Table 7: Statistics of SemVarBench.

and flagging those with alignment scores below 0.7. Then, we aggregate the votes from these models to determine the most representative candidates, and select those with the highest votes for further filtering. To ensure diversity, we categorize these samples based on permutation types, as shown in Fig. 5, with a maximum limit of 50 samples per category. Finally, 684 samples were included in our benchmark.

## C.3 DATA STATISTICS

**Category.** The samples in SemVarBench are divided into 20 categories based on their permutation types. Furthermore, these categories are further classified into three aspects based on triple types, as illustrated in Tab. 24. These aspects are *Relation*, *Attribute Comparison* and *Attribute Value*. Specifically, *Relation* aspect includes six categories: *Action, Interaction, Absolute Location, Relative Location, Spatial-Temporal, Direction*. *Attribute Contrast* includes four categories: *Size, Height, Weight, Vague Amount*. *Attribute Value* includes ten categories: *Color, Counting, Texture, Material, Shape, Age, Sentiment, Temperature, Manner*, and *Appearance*.

**Scale and Split**. SemVarBench comprises 11,454 valid samples of $(T_a, T_{pv}, T_{pi})$, totaling 34,362 sentences. We divide it into a training set and a test set. The training set contains 10,806 samples, while the test set consists of 648 challenging samples for effective evaluation, as shown in Tab. 7. All our evaluations are conducted on the test set.

**Distribution**. Since some permutations contain multiple words, they may fall into more than one category. In the training set, 51.06% of the permutations involve only one category, 35.12% involve two categories, 7.35% involve three categories, and 0.5% involve more than four categories. In the test set, 82.75% of permutations involve only one category, 14.77% involve two categories, and 2.49% involve three categories. As a result, the total count of categorized samples exceeds the actual number of unique samples.

**SemVarBench vs. Other benchmarks**. Compared with existing benchmarks, SemVarBench focuses on the understanding of semantic variations for text-to-image synthesis, which includes two types of permutations: permutation-variance and permutation-invariance. Other comparisons, such as source, scale, annotation, and split, are illustrated in Tab. 8.

| Benchmark | Concentration | Data Source | #Prompts | Annotation | Split |
|-----------|---------------|-------------|----------|------------|-------|
| DrawBench (Saharia et al., 2022) | General | Human | 200 | Human | Test |
| PartiPromps (Yu et al., 2022) | General | Human | 1600 | Human | Test |
| PaintSkills (Cho et al., 2023a) | General | Template | 73.3K | – | Train/Test |
| HRS-Bench (Bakr et al., 2023) | General | Template & LLM | 45.0K | Human | Test |
| SR$_{2D}$ (Gokhale et al., 2022) | Compositional | Dataset | 25.3K | – | Test |
| ABC-6K (Feng et al., 2023a) | Compositional | Dataset | 6.4K | – | Test |
| CC-500 (Feng et al., 2023a) | Compositional | Template | 500 | – | Test |
| TIFA v1.0 (Hu et al., 2023) | Compositional | Dataset | 4.1K | – | Test |
| VPEval-skill (Cho et al., 2023b) | Compositional | Dataset | 3.8K | – | Test |
| DSG-1K (Cho et al., 2024) | Compositional | Dataset | 1.1K | – | Test |
| T2I-CompBench (Huang et al., 2023) | Compositional | Template & LLM | 6.0K | – | Train/Test |
| Winoground (Thrush et al., 2022) | Permutation-Variance | Human | 800 | Human | Test |
| Winoground-T2I (Zhu et al., 2023) | Permutation-Variance | Template & LLM | 22K | LLM & Human | Test |
| **SemVarBench(ours)** | Permutation-Variance Permutation-Invariance | Template & LLM | 34K | LLM & Human | Train/Test |

Table 8: Comparison between SemVarBench and other T2I synthesis benchmarks.

# D DETAILS OF EXPERIMENT SETTING

## D.1 T2I SYNTHESIS MODELS

We evaluate 13 mainstream T2I models as shown in Tab. 1. For each sentence, we generate one image, resulting in a total of $684 \times 3 \times 13$ images. These T2I models are Stable Diffusion v1.5[3] (denoted as SD 1.5), Stable Diffusion v2.1[4] (denoted as SD 2.1), Stable Diffusion XL v1.0[5] (denoted as SD XL 1.0), Stable Cascade[6] (denoted as SD CA), DeepFloyd IF XL[7] (denoted as DeepFloyd), PixArt-alpha XL[8](denoted as PixArt), Kolors, Stable Diffusion 3 [medium][9](denoted as SD 3), FLUX.1 [dev][10] (denoted as FLUX.1), Midjourney V6[11] (denoted as MidJ V6), DALL-E 3[12], CogView3-Plus[13] (denoted as CogV3-Plus), and Ideogram 2[14]. The schedulers for SD 1.5 and SD 2.1 are set to DPM-Solver++, while all other settings are left as default.

## D.2 EVALUATOR

We use four advanced MLLMs as evaluators to demonstrate the general applicability of our proposed evaluation metrics: Gemini 1.5 Pro, Claude 3.5 Sonnet, GPT-4o, and GPT-4 Turbo. GPT-4o and GPT-4 Turbo have been shown to achieve near-human performance in evaluating alignment in T2I synthesis models (Zhang et al., 2023; Chen et al., 2024b).Claude 3.5 Sonnet outperforms GPT-4o and Gemini 1.5 Pro (Anthropic, 2024). The versions of these MLLMs used are as follows: Gemini 1.5 Pro (`gemini-1.5-pro-001`), Claude 3.5 Sonnet (`claude-3-5-sonnet-20240620`), GPT-4o (`gpt-4o-2024-05-13`), and GPT-4 Turbo (`gpt-4-turbo-2024-04-09`). The alignment score components follow the division outlined in (Zhang et al., 2023), with the exception of the aesthetic score component, which has been omitted. The complete prompt is as follows.

---

[3]The model used is `ruwnayml/stable-diffusion-v1-5`, which is now deprecated. A mirror is available at: https://huggingface.co/stable-diffusion-v1-5/stable-diffusion-v1-5

[4]https://huggingface.co/stabilityai/stable-diffusion-2-1

[5]https://huggingface.co/stabilityai/stable-diffusion-xl-base-1.0;https://huggingface.co/stabilityai/stable-diffusion-xl-refiner-1.0

[6]https://huggingface.co/stabilityai/stable-cascade-prior;https://huggingface.co/stabilityai/stable-cascade

[7]https://huggingface.co/DeepFloyd/IF-I-XL-v1.0;https://huggingface.co/DeepFloyd/IF-II-L-v1.0;https://huggingface.co/stabilityai/stable-diffusion-x4-upscaler

[8]https://huggingface.co/PixArt-alpha/PixArt-XL-2-1024-MS

[9]https://huggingface.co/stabilityai/stable-diffusion-3-medium

[10]https://huggingface.co/black-forest-labs/FLUX.1-dev

[11]https://www.midjourney.com/home

[12]https://openai.com/index/dall-e-3/

[13]https://www.bigmodel.cn/dev/api/image-model/cogview

[14]https://about.ideogram.ai/2.0

Does the generated image align with the given prompt?

[Instruction] Carefully assess the generated image in terms of relevance to the prompt and object accuracy. Notice that the image is digitally created or artificially generated, and I hope you help feedback on the quality of a generated image rather than discussing the content of a real photograph.

Use the following criteria to guide your evaluation: with Relevance (0-50 points), Object Accuracy (0-50 points). After providing your explanation, you must rate the generated image by strictly following this format: "[[rating]]", for example: "Relevance (0-50 points): [[35]], Object Accuracy (0-50 points): [[30]]".

[Prompt]
{prompt}

After receiving outputs from LLMs, we use regular expressions to extract scores. In our experiments, the outputs from the four evaluators mentioned above consistently followed the specified format as defined in the prompt. We also tested Qwen-VL-Chat, Qwen-VL-Plus, Qwen-VL-Max, and LLAVA-1.6, which exhibited poor adherence to the specified format and required a more complex extraction process. To simplify the evaluation process, we decided to adopt results exclusively from Gemini 1.5 Pro, Claude 3.5 Sonnet, GPT-4o, and GPT-4 Turbo.

### D.3 TRAINING SETTING

**Training Data Selection**. The training set of SemVarBench comprises 10,806 samples. We investigate the improvement from fine-tuning the T2I model Stable Diffusion XL v1.0. We select the generated images whose alignment scores meet the requirements. These constraints are as follows.

First, the generated image should be approximately aligned with its corresponding text prompt.

$$
\begin{cases}
S(T_a, I_a) & > C_2, \\
S(T_{pv}, I_{pv}) & > C_2,
\end{cases}
\tag{12}
$$

where $C_2$ is a threshold.

Second, the alignment scores between matched text-image pairs should be higher than those between mismatched text-image pairs.

$$
\begin{cases}
S(T_a, I_a) & > S(T_a, I_{pv}), \\
S(T_a, I_a) & > S(T_{pv}, I_a), \\
S(T_{pv}, I_{pv}) & > S(T_a, I_{pv}), \\
S(T_{pv}, I_{pv}) & > S(T_{pv}, I_a),
\end{cases}
\tag{13}
$$

Third, the visual semantic variations observed from different text prompts should be the same when the initial image and the final image are the same.

$$
S(T_a, I_a) - S(T_a, I_{pv}) \approx S(T_{pv}, I_{pv}) - S(T_{pv}, I_a),
\tag{14}
$$

Similarly, the textual semantic variations observed from different images should be the same when the initial text prompt and the final text prompt are the same.

$$
S(T_a, I_a) - S(T_{pv}, I_a) \approx S(T_{pv}, I_{pv}) - S(T_a, I_{pv}),
\tag{15}
$$

By utilizing this approximate equality relationship in Eq. 14 and Eq. 15, we constrain the alignment score using the following inequality:

$$
\begin{cases}
|(S(T_a, I_a) - S(T_a, I_{pv})) - (S(T_{pv}, I_{pv}) - S(T_{pv}, I_a))| < C_3, \\
|(S(T_a, I_a) - S(T_{pv}, I_a)) - (S(T_{pv}, I_{pv}) - S(T_a, I_{pv}))| < C_3,
\end{cases}
\tag{16}
$$

In our experiments, we utilized Stable Diffusion XL v1.0 to generate an image for each text prompt within the training set. To select the training data, we designated $C_2 = 0.8$ and $C_3 = 0.1$. Ultimately, we selected 327 samples, resulting in 981 sentences.

**Supervised Fine-Tuning (SFT)**. All text-image pairs $(T_i, I_i)$ are incorporated into the training set. Each sample $(T_a, T_{pv}, T_{pi})$ results in three text-image pairs: $(T_a, I_a)$, $(T_{pv}, I_{pv})$, and $(T_{pi}, I_{pi})$,

which leads to a total of 981 diverse pairs. The selected set of samples is denoted as $D_s$. The loss function for SFT remains unchanged (Kingma et al., 2021; Song et al., 2021), which is defined as

$$\mathcal{L}(\theta) = \mathbb{E}_{(x,y)\in\mathcal{D}_s}\left[\|\epsilon - \epsilon_\theta(z_t, t, y)\|_2^2\right], \tag{17}$$

where $x$, $y$, $t$, $z_t$ are the representations of the image $I_i$, text prompt $T_i$, timestamp $t$, and the latent representation of the image at timestamp $t$, respectively. We conducted two separate fine-tuning processes using the diffusers library[15]: We only fine-tuned the LoRA model either on the UNet or the text encoder for 5,000 steps, with a training batch size of 1. Our computational resources included an NVIDIA GeForce RTX 4090 with 25.2 GB of VRAM and a 16-core AMD EPYC 9354 processor, with 60.1 GB of system memory available. We trained the LoRA model with a rank of 4 on UNet or text encoders, and the training process took approximately 0.5 hours.

**Direct Policy Optimization (DPO)**. We added text-image tuples of the form $(T_i, I_i, I_j)$ to the training set, where the semantic content of $T_i$ does not match $T_j$. For each input $T_i$, $I_i$ represents the chosen image and $I_j$ the rejected one. Each sample $(T_a, T_{pv}, T_{pi})$ results in four text-image tuples: $(T_a, I_a, I_{pv})$, $(T_{pv}, I_{pv}, I_a)$, $(T_{pv}, I_{pv}, I_{pi})$, and $(T_{pi}, I_{pi}, I_{pv})$, which leads to a total of 1,308 tuples. The loss function for DPO remains unchanged (Wallace et al., 2024), which is defined as

$$\mathcal{L}(\theta) = -\mathbb{E}_{(x^w, x^l, y)\sim\mathcal{D}_s, z_t^w\sim q(z_t^w|x^w), z_t^l\sim q(z_t^l|x^l)}\log\sigma\big($$
$$-\beta(\|\epsilon^w - \epsilon_\theta(z_t^w, t, y)\|_2^2 - \|\epsilon^w - \epsilon_{ref}(z_t^w, t, y)\|_2^2 - \tag{18}$$
$$(\|\epsilon^l - \epsilon_\theta(z_t^l, t, y)\|_2^2 - \|\epsilon^l - \epsilon_{ref}(z_t^l, t, y)\|_2^2))),$$

where $x^w$, $x^l$, $y$, $t$, $z_t^w$, $z_t^l$, $\sigma$ are the representations of the chosen image $I_i$, the rejected image $I_j$, text prompt $T_i$, timestamp $t$, the latent representation of the chosen image at timestamp $t$, the latent representation of the rejected image at timestamp $t$, and the sigmoid function. We executed two separate fine-tuning processes using the DiffusionDPO[16]: We only fine-tuned the LoRA model either on the UNet or the text encoder for 5,000 steps, with a training batch size of 1. Our computational resources included a Tesla V100-SXM2 with 32 GB of VRAM and an 11-core Intel(R) Xeon(R) Platinum 8163 processor, with 88.0 GB of system memory available. We trained the LoRA model with a rank of 4 on the UNet or text encoders, and the training process took approximately 4.5 hours.

# E MORE EXPERIMENT RESULTS

## E.1 RESULTS ON CATEGORIES

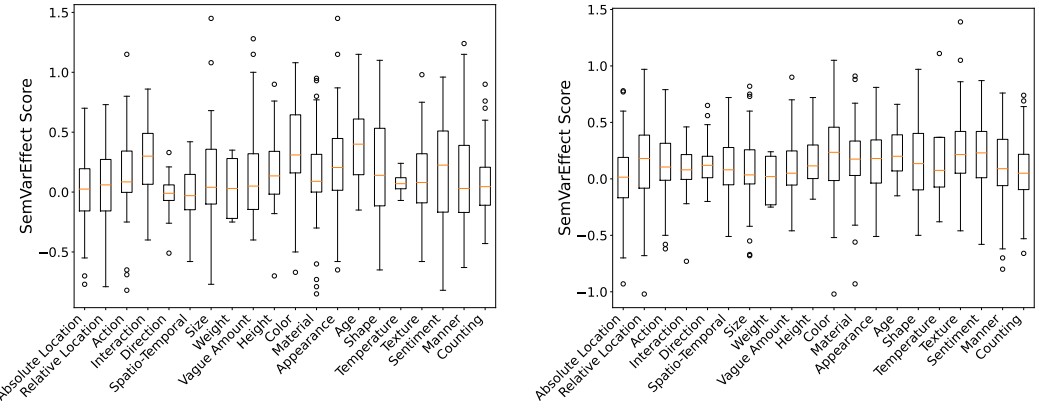

Figure 11: The distribution of SemVarEffect scores across various categories for the Ideogram 2 and the DALL-E 3 model, as evaluated by GPT-4 Turbo. Left: Ideogram 2. Right: DALL-E 3.

---

[15]The diffusers library supports fine-tuning of both the Unet and the Unet + text encoder. We made minor modifications to support fine-tuning only the text encoder. The url of scripts provided by diffusers is: https://github.com/huggingface/diffusers/tree/main/examples/text_to_image/

[16]We used the code provided by the diffusers library, which supports fine-tuning of the Unet. We made minor modifications to support fine-tuning only the text encoder. The url of scripts provided by diffusers is: https://github.com/huggingface/diffusers/tree/main/examples/research_projects/diffusion_dpo/

| Models | Relation | | | | | | Attribute Comparison | | | |
|---|---|---|---|---|---|---|---|---|---|---|
| | Absolute Location | Relative Location | Action | Interaction | Direction | Spatial-Temporal | Size | Weight | Vague Amount | Height |
| **Open Source Models** | | | | | | | | | | |
| Stable Diffusion v1.5 | -0.01 | 0.00 | -0.06 | -0.11 | 0.05 | -0.01 | 0.11 | -0.01 | -0.07 | 0.01 |
| Stable Diffusion v2.1 | -0.01 | -0.06 | -0.08 | 0.02 | -0.02 | -0.00 | 0.03 | -0.10 | 0.02 | 0.06 |
| Stable Diffusion XL v1.0 | -0.02 | -0.08 | -0.01 | 0.05 | 0.05 | -0.05 | 0.07 | 0.16 | 0.03 | 0.09 |
| Stable Cascade | 0.02 | -0.03 | 0.01 | -0.03 | 0.02 | 0.02 | -0.02 | -0.09 | 0.08 | -0.01 |
| DeepFloyd IF XL | -0.01 | -0.00 | 0.01 | -0.01 | 0.04 | 0.01 | 0.03 | 0.05 | -0.04 | 0.03 |
| PixArt-alpha XL | 0.00 | -0.01 | 0.03 | 0.00 | -0.04 | -0.03 | 0.07 | 0.10 | 0.10 | 0.03 |
| Kolors | -0.03 | 0.02 | -0.07 | 0.02 | 0.03 | -0.02 | -0.06 | -0.10 | 0.07 | 0.07 |
| Stable Diffusion 3 | -0.03 | 0.01 | -0.02 | 0.05 | -0.08 | -0.04 | 0.07 | -0.02 | 0.10 | 0.04 |
| FlUX.1 | -0.03 | 0.03 | 0.03 | 0.08 | -0.04 | -0.00 | 0.09 | **0.23** | 0.05 | 0.09 |
| **API-based Models** | | | | | | | | | | |
| Midjourney V6 | 0.07 | 0.01 | 0.04 | 0.03 | 0.03 | 0.08 | 0.07 | -0.12 | 0.07 | 0.02 |
| DALL-E 3 | -0.00 | **0.12** | 0.11 | 0.08 | **0.13** | **0.11** | 0.08 | -0.00 | 0.09 | 0.15 |
| CogView3-Plus | **0.08** | 0.08 | **0.23** | 0.07 | 0.03 | -0.01 | **0.23** | -0.03 | **0.23** | **0.22** |
| Ideogram 2 | 0.01 | 0.04 | 0.13 | **0.29** | -0.02 | -0.02 | 0.12 | 0.04 | 0.17 | 0.17 |

Table 9: The results of SemVarEffect scores $\kappa$ on aspects *Relation* and *Attribute Comparison*. The evaluator is GPT-4 Turbo.

| Models | Attribute Value | | | | | | | | | | AVG |
|---|---|---|---|---|---|---|---|---|---|---|---|
| | Color | Material | Appearance | Age | Shape | Temperature | Texture | Sentiment | Manner | Counting | |
| **Open Source Models** | | | | | | | | | | | |
| Stable Diffusion v1.5 | 0.09 | 0.02 | 0.04 | -0.20 | 0.01 | 0.06 | 0.02 | -0.06 | -0.05 | -0.07 | -0.01 |
| Stable Diffusion v2.1 | 0.12 | 0.10 | -0.03 | -0.09 | -0.00 | -0.06 | -0.03 | -0.10 | -0.01 | 0.02 | 0.00 |
| Stable Diffusion XL v1.0 | 0.13 | 0.09 | -0.01 | 0.01 | -0.01 | 0.05 | -0.00 | 0.03 | -0.00 | 0.00 | 0.02 |
| Stable Cascade | 0.14 | 0.05 | 0.10 | -0.03 | -0.02 | -0.15 | -0.03 | -0.05 | 0.06 | 0.01 | 0.04 |
| DeepFloyd IF XL | 0.19 | 0.14 | 0.06 | -0.19 | 0.04 | -0.02 | 0.05 | -0.05 | 0.06 | 0.01 | 0.04 |
| PixArt-alpha XL | 0.11 | 0.09 | 0.00 | 0.15 | 0.01 | 0.13 | -0.02 | 0.02 | 0.03 | -0.00 | 0.02 |
| Kolors | 0.21 | 0.07 | -0.01 | 0.01 | -0.09 | 0.01 | 0.10 | -0.01 | -0.04 | 0.00 | 0.01 |
| Stable Diffusion 3 | 0.33 | 0.10 | 0.11 | 0.04 | 0.08 | 0.11 | 0.08 | 0.01 | 0.03 | 0.06 | 0.06 |
| FlUX.1 | 0.35 | **0.21** | 0.21 | 0.08 | 0.09 | -0.13 | 0.04 | -0.06 | 0.07 | **0.10** | 0.09 |
| **API-based Models** | | | | | | | | | | | |
| Midjourney V6 | 0.20 | 0.12 | 0.13 | 0.10 | -0.03 | -0.21 | 0.11 | -0.05 | 0.09 | -0.02 | 0.05 |
| DALL-E 3 | 0.22 | 0.17 | 0.17 | 0.23 | 0.14 | 0.22 | **0.22** | 0.19 | 0.11 | 0.06 | 0.12 |
| CogView3-Plus | 0.35 | 0.15 | 0.17 | 0.31 | 0.16 | **0.30** | 0.17 | **0.21** | **0.27** | 0.07 | **0.15** |
| Ideogram 2 | **0.37** | 0.15 | **0.24** | **0.42** | **0.20** | 0.08 | 0.12 | 0.16 | 0.13 | 0.07 | 0.13 |

Table 10: The results of SemVarEffect scores $\kappa$ on aspects *Attribute Value*. AVG represents the average effect score of all samples on aspect *Relation*, *Attribute Comparison* and *Attribute Value*. The evaluator is GPT-4 Turbo.

**Effects of Semantic Variations on Different Categories**. The impact of semantic variations is not uniform across different semantic classes, as shown in Fig. 7, with exact scores listed in Tabs. 9 and Tab. 10. For *Relation*, most models consistently show low scores, as indicated by the dark blue shading in Fig. 7. This suggests that models handle samples involving *Relations*—such as *Absolute Location*, *Relative Location*, and *Actions*—with limited accuracy. For *Attribute Value*, models such as Ideogram2 perform significantly better at capturing attributes such as *Color*, as shown by the prominent red shading in Fig. 7. These models demonstrate a clear advantage in both generating and recognizing these attributes. In contrast, models such as DALL-E 3 and CogV3-Plus display a more balanced but average performance across most categories (shaded in light orange and light blue). For *Attribute Comparison* (e.g., *Size*, *Weight*, *Height*), most models score lower, indicating their weaker ability to handle complex attribute comparisons.

Although most T2I models struggle with capturing semantic variations in many categories, some categories, such as *Color* and *Age*, demonstrate slightly better performance, as indicated by higher median values. Fig. 11 illustrates the distribution of SemVarEffect scores across various categories for the Ideogram 2 model, while Fig. 12 shows the scores for different T2I models in the *Color* and *Direction* categories. Most categories have medians (marked by the orange line) close to zero, indicating that T2I models generally struggle to capture the semantic variations introduced by word order changes, particularly in the *Direction* category. However, some categories, such as *Weight* and *Color*, show slightly higher median values, indicating that semantic variation caused by word order changes may have a minor positive effect in these instances. Categories such as *Absolute Location* and *Counting* show greater variability in model responses, while categories such as *Sentiment* and *Texture* show more consistent effects with narrower distributions.

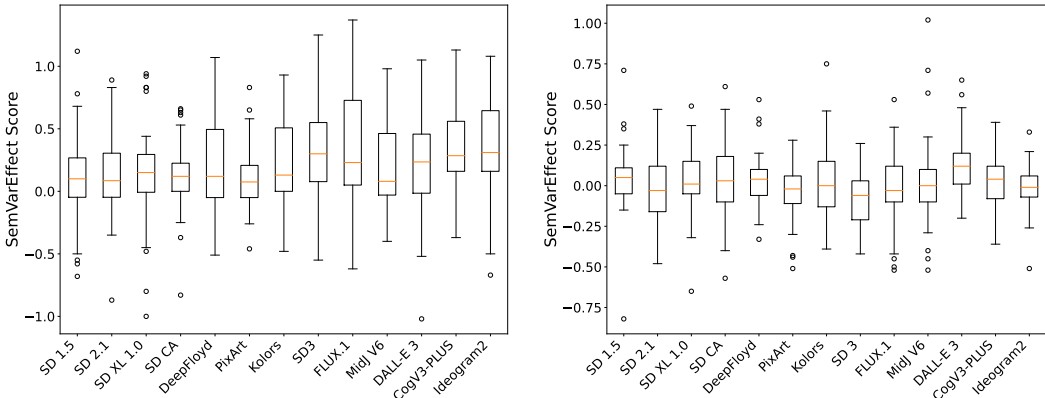

Figure 12: The distribution of SemVarEffect scores across various T2I models within the Color and Direction categories, as evaluated by GPT-4 Turbo. Top: Color. Bottom: Direction.

### E.2 RESULTS ON HUMAN EVALUATION

**Human Evaluation**. To validate the effectiveness of MLLMs we used, we conducted human evaluation with three raters on 80 samples (20 from each of the SOTA models: Midjourney v6, DALL-E 3, CogView3-Plus, and Ideogram2) through stratified sampling (one for each category). Following the same scoring protocol as our automatic evaluation, each rater scores the semantic alignment of matched or mismatched image-text pairs, and we use their mean scores to calculate SemVarEffect. The results demonstrate the reliability of our MLLM-based evaluation approach. First, we observe consistent performance trends between human raters and the four MLLMs across all evaluated models (Tab. 11). Second, our correlation analysis on CogView3-Plus reveals moderate Pearson's $\rho$, Spearman's $\phi$, and Cohen's Kappa $\kappa_{cohen}$ coefficients between machine and human scores (Tab. 12), suggesting that our selected MLLMs can serve as a reliable proxy for human evaluation. This validates our MLLM-based evaluation while confirming the current limitations of T2I models. The interfaces and criteria for human evaluation are shown in Fig. 13 and Tab. 13.

| Models | $\bar{S}(\uparrow)$ | $\gamma_w(\uparrow)$ | $\gamma_{wo}(\downarrow)$ | $\kappa(\uparrow)$ |
|---|---|---|---|---|
| Midjourney V6 | 0.59 | 0.37 | 0.49 | -0.12 |
| DALL-E 3 | 0.63 | 0.53 | 0.50 | 0.03 |
| CogView3-Plus | 0.69 | **0.52** | 0.33 | **0.19** |
| Ideogram 2 | **0.74** | 0.50 | **0.31** | **0.19** |

Table 11: Human evaluation results of different T2I models in understanding semantic variations.

| Models | $\rho(\uparrow)$ | $\phi(\uparrow)$ | $\kappa_{cohen}(\uparrow)$ |
|---|---|---|---|
| GPT-4o | **0.54** | **0.53** | 0.53 |
| GPT-4v | 0.50 | 0.51 | **0.54** |
| Claude-3.5-Sonnet | 0.42 | 0.37 | 0.37 |
| Gemini-Pro-1.5 | 0.27 | 0.11 | 0.23 |

Table 12: Correlation coefficients between GPT-4o, GPT-4v, Claude 3.5 Sonnet, Gemini Pro 1.5 and human evaluations of the SemVar-Effect for CogView3-Plus.

## F MORE ANALYSIS

### F.1 TEXT ENCODER

**Do different text encoders themselves distinguish semantic variations caused by linguistic permutations in the text space?** We explore the efficacy of diverse text encoders in discerning such nuances. Fig. 14 compares the text similarity between $T_a$ and $T_{pv}$ across models utilizing different text encoder models. SD 1.5, SD 2.1, SD XL v1.0, and SC utilize CLIP series models as text encoders, while DeepFloyd, PixArt, and DALL-E 3 utilize T5 series models. The similarity metric is depicted as $1 - \cos(T_a, T_{pv})$, with higher values indicating a stronger ability of the text encoder to differentiate between the semantics of the two sentences. This indicates that the choice of text encoder significantly influences the model's semantic discrimination capabilities.

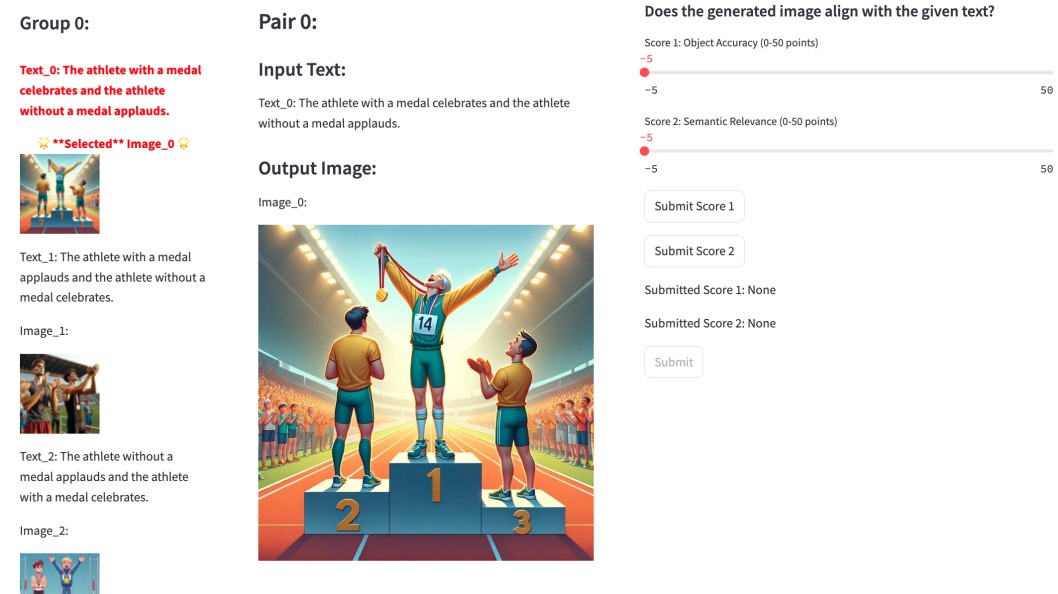

Figure 13: Interface for the human evaluation

| Score 1: Object Accuracy (0-50 points): The existence and attributes of objects. | |
|---|---|
| 50 | Perfectly Accurate (All objects exist, and their attributes are described correctly). |
| 40 | Highly Accurate (All objects exist, but some minor details may differ slightly). |
| 30 | Partially Accurate (Most objects exist, but there are noticeable errors in appearance, color, or shape). |
| 20 | Mostly Inaccurate (Significant deviation from the description, with multiple attribute errors). |
| 10 | Severely Inaccurate (Most object details contradict the description). |
| 0 | Completely Inaccurate (Objects described are missing or entirely replaced by different objects). |
| **Score 2: Relevance (0-50 points)**: Relationships, positions, and interactions between objects. | |
| 50 | Perfect Match (Object positions and interactions fully align with the description). |
| 40 | Highly Matched (Most object positions and relationships align, with minor mismatches that do not affect overall understanding). |
| 30 | Partially Matched (Key objects exist, but there are significant mismatches in position or relationships, leading to a slightly different overall context). |
| 20 | Mostly Mismatched (Object relationships do not align with the description). |
| 10 | Severely Misaligned (Most object relationships are chaotic). |
| 0 | Completely Mismatched (All Object relationships are completely different from the description). |

Table 13: Criteria for the human evaluation

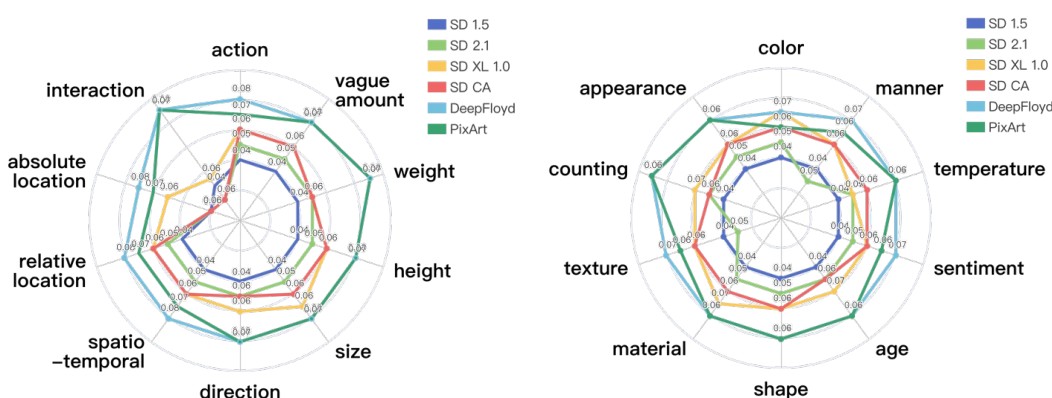

Figure 14: The semantic discrimination capabilities of different text encoders measured by $1 - \text{cosine}(T_a, T_{pv})$.

| Category | $(T_a, T_{pv})$ | $(T_a, T_{pi})$ | $(T_a, T_{random})$ |
|---|---|---|---|
| **Relation** | | | |
| Absolute Location | 11.94 | 27.24 | 52.40 |
| Relative Location | 12.26 | 28.62 | 46.98 |
| Action | 13.94 | 31.85 | 48.35 |
| Interaction | 13.56 | 29.58 | 44.26 |
| Direction | 12.03 | 27.18 | 50.21 |
| Spatio-temporal | 17.40 | 42.74 | 59.94 |
| **Attribute Comparison** | | | |
| Vague amount | 19.38 | 36.56 | 50.38 |
| Size | 11.38 | 26.00 | 46.82 |
| Height | 13.04 | 23.00 | 31.22 |
| Weight | 10.00 | 26.20 | 22.20 |
| **Attribute Value** | | | |
| Color | 11.80 | 30.86 | 46.90 |
| Material | 12.40 | 28.42 | 41.92 |
| Appearance | 13.86 | 44.14 | 59.14 |
| Age | 14.73 | 34.73 | 46.64 |
| Shape | 13.34 | 33.48 | 40.98 |
| Temperature | 11.00 | 27.50 | 38.50 |
| Texture | 11.74 | 31.20 | 54.22 |
| Sentiment | 11.96 | 33.15 | 48.65 |
| Manner | 13.37 | 33.71 | 52.90 |
| Counting | 8.44 | 29.06 | 44.18 |
| **Average** | **13.12** | **31.65** | **54.30** |

Table 14: The average edit distance between sentences in different categories.

**Why do permutations without semantic changes exhibit higher text similarity scores compared to those with semantic changes?** This phenomenon is closely related to our dataset's construction methodology, where $T_{pi}$ is generated by swapping two long phrases located on either side of a coordinating conjunction or a predicate, such as the *and* in Fig. 4. We observed that permutations with semantic changes in our benchmark have significantly smaller edit distances from the anchor sentence compared to synonymous sentences, as shown in Tab. 14. The average edit distances between $(T_a, T_{pv})$, $(T_a, T_{pi})$ and $(T_a, T_{random})$ are 13, 32, and 53. As our analysis does not rely on the similarity scores of synonymous sentences, this does not affect our previous findings.

## F.2 EVALUATION METRICS

### F.2.1 ALIAGNMENT SCORE & SEMVAREFFECT SCORE

**Detailed Analysis on Text-image Alignment Scores vs. SemVarEffect Score** Fig. 15 illustrates that although the distributions of the SemVarEffect score and the alignment score are similar, the SemVarEffect score demonstrates a higher degree of differentiation, especially when distinguishing between FLUX.1 and SD 3. Based on the alignment score, it could be concluded that FLUX.1, SD 3, and SD XL 1.0 have comparable performance levels and may be grouped into the same cluster. However, based on the SemVarEffect score, it becomes evident that FLUX.1 and SD 3 differ distinctly from SD XL 1.0. SD XL 1.0 responds more similarly to semantic variations caused by word order changes in a manner similar to SD 1.5, SD 2.1, and SD CA. Correspondingly, we observe that when using the T5-XXL series model as the text encoder, the difference between DALL-E 3 and other models, such as PixArt and DeepFloyd, becomes more pronounced when assessed by the SemVarEffect score.

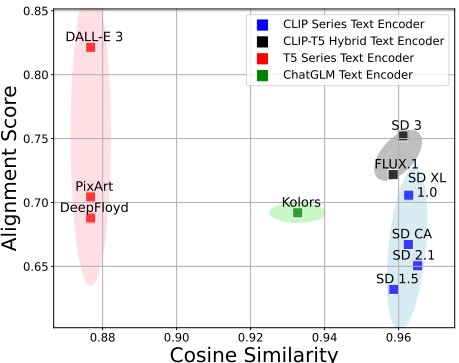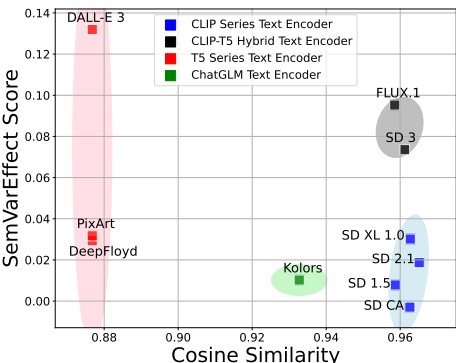

Figure 15: A comparison of alignment scores and the SemVarEffect score under the same conditions of text similarity. The squares are results of permutations of permutation-variance. The evaluator is GPT-4 Turbo.

It should be noted that SemVarEffect is not intended to challenge alignment evaluation, but rather to serve as a complementary metric to alignment score $\bar{S}$. This metric focuses only on the consistency of variations rather than the absolute quality of generations. Thus, a model could still score well on SemVarEffect even if it consistently generates incorrect attributes (such as believing oranges are blue and bananas are green), as long as it maintains consistent semantic variations. SemVarEffect helps validate the reliability of high alignment scores $\bar{S}$: high $\bar{S}$ with low $\kappa$ indicates limited semantic understanding, while high $\bar{S}$ with high $\kappa$ suggests effective semantic understanding.

### F.3 TRAINING DATA

#### F.3.1 DATA FILTER CHOICES

**How do the filtering standards of training data using MMLM potentially affect the evaluation results, particularly for categories with limited samples?** We examine filtering standards across data-rich and data-limited categories to study the effects of data scale and data quality. The filtering criteria are as follows:

- Alignment Score Thresholds: Values of 0.8 and 0.6 for $C_2$ (defined in Eq. 12)
- Existence of Strict Filtering on Semantic Variations: Clear distinction between matched and unmatched text-image pairs through strict filtering criteria (defined in Eq. 13-16).

– High filtering. High filtering standards can reduce the available data in categories with limited data, leading to decreased alignment scores and SemVarEffect scores. Given the same initial data size, *Absolute_location* has lower alignment scores ($> 0.6$) than *Height/Direction* ($> 0.7$) due to stricter filtering effects, as shown in Tab. 3. However, due to its large initial data pool (1.7k candidates), *Absolute_location* retains sufficient high-quality samples for effective fine-tuning under the "0.8+strict" setting, despite the low retention rate, as shown in Tabl. 15. This comparison between categories with different data scales demonstrates that initial data volume is crucial for achieving improvements.

– Relaxing filtering. Results show that relaxing filtering criteria to allow more training data leads to worse performance compared to the strictest standards. Removing "0.8+strict filtering" leads to decreased performance across all categories, even those with sufficient data like *Color* and *Absolute_location*, as shown in Tab. 15. For categories with limited data, "0.8+strict filtering" minimizes the decrease in both alignment scores and semantic variation effects, as shown in Tab. 16. This demonstrates that even with limited data, lowering filtering standards is not a viable solution.

#### F.3.2 COMMONSENSE BIASES

**Is the model's insensitivity to word order variations caused by commonsense biases in training data?** To investigate this question, we conducted controlled experiments with balanced training data containing both commonsense and anti-commonsense examples, and evaluated the model's generalization to novel object pairs. For training, we created a balanced dataset consisting of 40 plausible images for "cat chasing mouse" (human-selected from DALL-E 3 generations) and 40 anti-commonsense images for "mouse chasing cat". The anti-commonsense images were manually created, where 31 images were generated via DALL-E 3 and professionally edited by designers, and

| Filter Constrains | $\bar{S}(\uparrow)$ | $\gamma_w(\uparrow)$ | $\gamma_{wo}(\downarrow)$ | $\kappa(\uparrow)$ |
|---|---|---|---|---|
| zeroshot | 0.73 | 0.33 | 0.25 | 0.08 |
| 0.8 + strict filtering | **0.78**($\uparrow$) | **0.44**($\uparrow$) | 0.24($\downarrow$) | 0.20($\uparrow$) |
| 0.8 + no filtering | 0.70($\downarrow$) | 0.42($\uparrow$) | 0.29($\uparrow$) | 0.14($\uparrow$) |
| 0.6 + strict filtering | 0.73(-) | **0.44**($\uparrow$) | **0.23**($\downarrow$) | **0.22**($\uparrow$) |
| 0.6 + no filtering | 0.72($\downarrow$) | **0.44**($\uparrow$) | 0.27($\uparrow$) | 0.19($\uparrow$) |

| Filter Constrains | $\bar{S}(\uparrow)$ | $\gamma_w(\uparrow)$ | $\gamma_{wo}(\downarrow)$ | $\kappa(\uparrow)$ |
|---|---|---|---|---|
| zeroshot | 0.64 | 0.29 | 0.34 | -0.05 |
| 0.8 + strict filtering | **0.65**($\uparrow$) | **0.42**($\uparrow$) | 0.35($\uparrow$) | **0.07**($\uparrow$) |
| 0.8 + no filtering | 0.54($\downarrow$) | 0.25($\downarrow$) | **0.31**($\downarrow$) | -0.05(-) |
| 0.6 + strict filtering | 0.56($\downarrow$) | 0.36($\uparrow$) | 0.38($\uparrow$) | -0.02($\uparrow$) |
| 0.6 + no filtering | 0.57($\downarrow$) | 0.31($\uparrow$) | 0.40($\uparrow$) | -0.09($\downarrow$) |

Table 15: Fine-tuning Results on *Color* (left) and *Absolute_Location* (right) with different data filtering standard.

| Filter Constrains | $\bar{S}(\uparrow)$ | $\gamma_w(\uparrow)$ | $\gamma_{wo}(\downarrow)$ | $\kappa(\uparrow)$ |
|---|---|---|---|---|
| zeroshot | **0.77** | **0.34** | 0.23 | **0.10** |
| 0.8 + strict filtering | **0.77**(-) | 0.27($\downarrow$) | **0.09**($\downarrow$) | 0.07($\downarrow$) |
| 0.8 + no filtering | 0.74($\downarrow$) | 0.32($\downarrow$) | 0.28($\uparrow$) | 0.04($\downarrow$) |
| 0.6 + strict filtering | 0.72($\downarrow$) | 0.29($\downarrow$) | 0.38($\uparrow$) | 0.05($\downarrow$) |
| 0.6 + no filtering | 0.73($\downarrow$) | 0.33($\downarrow$) | 0.29($\uparrow$) | 0.04($\downarrow$) |

| Filter Constrains | $\bar{S}(\uparrow)$ | $\gamma_w(\uparrow)$ | $\gamma_{wo}(\downarrow)$ | $\kappa(\uparrow)$ |
|---|---|---|---|---|
| zeroshot | **0.79** | 0.20 | **0.15** | **0.05** |
| 0.8 + strict filtering | 0.77($\downarrow$) | **0.25**($\uparrow$) | 0.24($\uparrow$) | 0.02($\downarrow$) |
| 0.8 + no filtering | 0.75($\downarrow$) | 0.24($\uparrow$) | 0.24($\uparrow$) | 0.00($\downarrow$) |
| 0.6 + strict filtering | 0.75($\downarrow$) | 0.19($\downarrow$) | 0.28($\uparrow$) | -0.09($\downarrow$) |
| 0.8 + no filtering | 0.74($\downarrow$) | 0.20(-) | 0.27($\uparrow$) | -0.07($\downarrow$) |

Table 16: Fine-tuning Results on *Height* (left) and *Direction* (right) with different data filtering standard.

9 images were created using vector graphics compositing, as shown in Figure 16. To evaluate the model's performance, we categorized generation errors into five types:

- Missing Objects: generated images lacking one or more required objects.
- No Interaction: all objects present but without any interaction.
- Wrong Interaction: objects interacting but not performing the required chasing action.
- Wrong Direction: objects running but not in a chasing formation (e.g., running in opposite directions).
- Reversed Roles: correct chasing action but with reversed subject-object roles (e.g., mouse chasing cat when cat chasing mouse was required).

Our training performance results showed that while generation quality improved (the number of right cases increased from 0-2 to 10-14), Reversed Role errors also increased, suggesting that the model learned to generate chase scenes but struggled with directional semantics, as shown in Tab. 17. When testing on novel object pairs with the same "chasing" relationship (both plausible pairs like "hippo↔elephant" and "bull↔man"), we observed consistently poor performance across all new pairs, with similar increasing trends in No Interaction errors and Reversed Roles, as shown in Tab. 18 and Tab. 19. Notably, there was no significant difference between plausible and anti-commonsense scenarios. These findings suggest that the core challenge isn't commonsense bias, but rather a fundamental limitation in processing directional relationships. The model struggles equally with both plausible and anti-commonsense scenarios, indicating an inability to establish proper subject-object relationships regardless of semantic plausibility.

### F.3.3 PLAUSIBLE SCENARIOS

**Can current T2I models effectively distinguish different semantic relations in plausible scenarios?** We investigate the ability of advanced commercial T2I models to handle semantic variations through two plausible scenarios: "A cat chasing a dog" and "A dog chasing a cat". Both scenarios are

| Class | Reasons | SD XL | | FT SD XL (trained on mouse↔cat) | |
|---|---|---|---|---|---|
| | | mouse→cat | cat→mouse | mouse→cat | cat→mouse |
| Wrong | Missing Objects | 12 | 14 | 4 | 2 |
| | No Interaction | 3 | 1 | **7** | **7** |
| | Wrong Interaction | 4 | 5 | 3 | 2 |
| | Wrong Direction | 7 | 8 | 0 | 5 |
| | Reversed Role | 2 | 0 | **6** | 0 |
| Right | Partial/Full Match | 0 | 2 | **10** | **14** |

Table 17: Training performance of SD XL before and after fine-tuning on the balanced mouse↔cat data.

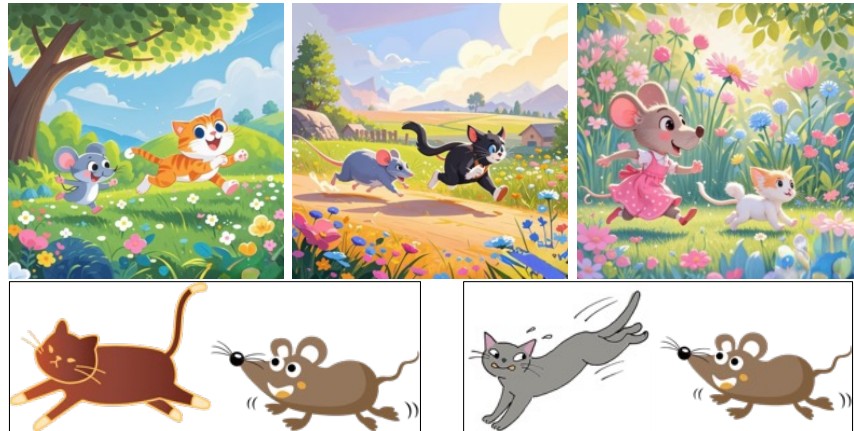

Figure 16: Examples of an anti-commonsense scenario "a mouse chasing a cat". Top: DALL-E 3 generated images with professional designer editing in Photoshop. Bottom: Vector graphics compositions.

| Class | Reasons | SD XL | | FT (trained on mouse↔cat) | |
|---|---|---|---|---|---|
| | | bull→man | man→bull | bull→man | man→bull |
| Wrong | Missing Objects | 0 | 0 | 0 | 0 |
| | No Interaction | 6 | 12 | **14** | **15** |
| | Wrong Interaction | 5 | 2 | 4 | **3** |
| | Wrong Direction | 16 | 12 | 8 | 3 |
| | Reversed Role | 2 | 2 | 2 | **6** |
| Right | Partial/Full Match | 1 | 2 | **2** | **3** |

Table 18: Testing performance on unseen bull↔man pairs after fine-tuning SD XL on the balanced mouse↔cat data.

possible in real life. Using DALL-E 3 and CogView3-Plus, we generated 30 images for each prompt and evaluated them based on strict criteria: (a) both animals, (b) running, (c) in the same direction, (d) clear spatial relationships (chaser behind chased), and (e) proper chase interaction. Results show significant performance differences, as shown in Tab. 20. For "A dog chasing a cat", DALL-E 3 achieved 70% accuracy (21/30). However, for "A cat chasing a dog", performance dropped dramatically. DALL-E 3 achieved only 13.3% accuracy while CogView3-Plus failed completely. Failed cases either involved No Interaction or Reversed Role errors. Even though these scenarios are more plausible than "a mouse chasing a cat", particularly "a cat chasing a dog", current advanced T2I models still struggle with semantic role reversals.

### F.3.4 DATA IMBALANCE

**Do T2I models' struggles with semantic relationships stem from training data imbalance?** We conducted experiments to test the performance of fine-tuned SDXL trained with balanced training data. We used a human-filtered balanced dataset of cat↔dog chasing interactions for training, with 80 images for "a dog chasing a cat" and 80 images for "a cat chasing a dog", selected from

| Class | Reasons | SD XL | | FT (trained on mouse↔cat) | |
|---|---|---|---|---|---|
| | | hippo→elephant | elephant→hippo | hippo→elephant | elephant→hippo |
| Wrong | Missing Objects | 11 | 16 | 4 | 9 |
| | No Interaction | 11 | 10 | **16** | **14** |
| | Wrong Interaction | 7 | 3 | 4 | 2 |
| | Wrong Direction | 0 | 1 | 0 | 0 |
| | Reversed Role | 1 | 0 | **4** | **1** |
| Right | Partial/Full Match | 0 | 0 | **2** | **4** |

Table 19: Testing performance on unseen hippo↔elephant pairs after fine-tuning SD XL on the balanced mouse↔cat data.

| Class | Reasons | DALL-E 3 cat→dog | DALL-E 3 dog→cat | Cogview3-Plus cat→dog | Cogview3-Plus dog→cat |
|---|---|---|---|---|---|
| Wrong | Miss Objects | 0 | 0 | 0 | 0 |
| | No Interaction | 17 | 5 | 12 | **22** |
| | Wrong Interaction | 0 | 0 | 0 | 0 |
| | Wrong Direction | 0 | 3 | 0 | 0 |
| | Reversed Role | 9 | 1 | **18** | 0 |
| Right | Partial/Full Match | 4 | 21 | 0 | 8 |

Table 20: Error analysis of advanced commercial text-to-image models (DALL-E 3 and Cogview3-Plus) in generating directional relationships.

| Class | Reasons | SD XL mouse→cat | SD XL cat→mouse | FT SD XL (trained on cat↔dog) mouse→cat | FT SD XL (trained on cat↔dog) cat→mouse |
|---|---|---|---|---|---|
| Wrong | Missing Objects | 12 | 14 | 12 | **21** |
| | No Interaction | 4 | 5 | 2 | 1 |
| | Wrong Interaction | 3 | 1 | **4** | **2** |
| | Wrong Direction | 7 | 8 | 4 | 1 |
| | Reversed Role | 2 | 0 | **5** | **2** |
| Right | Partial/Full Match | 0 | 2 | **3** | **3** |

Table 21: Testing performance on unseen mouse↔cat pairs after fine-tuning SD XL on the balanced cat↔dog data.

**DALL-E 3 generations.** We used two unseen prompt pairs involving the same "chasing" relationship between common objects for testing. The experiment results are shown in Tab. 21 and Tab. 22. Despite balanced training, the model showed consistently poor generalization: accuracy remained low for both anti-commonsense scenarios (mouse↔cat, 3-3/30) and plausible scenarios (bull↔man, 4-5/30). Failure analysis revealed three main categories: (1) Missing Objects, where the model fails to generate all required objects; (2) Relationship Understanding Failures, where objects appear either without interaction or with incorrect interactions (e.g. No Interaction, Wrong Interaction, and Wrong Direction), indicating the model's inability to comprehend the "chasing" concept; and (3) Reversed Roles, where the model fails to properly assign who chases whom. Even among generated images, correct relationships occurred less frequently than these failure cases, suggesting random performance rather than true understanding. Thus, the models' struggles persist even with perfectly balanced training data, suggesting the core issue lies in relationship understanding rather than data imbalance.

## F.4 TRAINING MECHANISM

### F.4.1 TOKEN-LEVEL IMPROVEMENT VS. SEMANTIC-LEVEL IMPROVEMENT

**Does fine-tuning enhance token-level or semantic-level understanding?** The semantic performance of fine-tuned SD XL is shown in Tab. 3. To distinguish between improvements in token-level or semantic level, we additionally conducted experiments at token level. We instructed GPT-4 to verify whether "words with specific meaning" (including nouns, verbs, adjectives, adverbs, and relationship-describing prepositions) from the input text are reflected in the generated image. For example, in "a dog chasing a cat", GPT-4 would identify three content words ("dog", "chasing", "cat") and verify their presence independently. Our analysis of the *Absolute_location* and *Height*

| Class | Reasons | SD XL bull→man | SD XL man→bull | FT SD XL (trained on cat↔dog) bull→man | FT SD XL (trained on cat↔dog) man→bull |
|---|---|---|---|---|---|
| Wrong | Missing Objects | 0 | 0 | 0 | 0 |
| | No Interaction | 6 | 12 | 6 | **13** |
| | Wrong Interaction | 5 | 2 | 0 | 0 |
| | Wrong Direction | 16 | 12 | 11 | 9 |
| | Reversed Role | 2 | 2 | **9** | **3** |
| Right | Partial/Full Match | 1 | 2 | **4** | **5** |

Table 22: Testing performance on unseen bull↔man pairs after fine-tuning SD XL on the balanced cat↔dog data.

categories revealed improved token-level accuracy after fine-tuning, as shown in Tab. 4. While fine-tuning improves token accuracy for the *Height* category, it actually leads to a degradation in semantic understanding. This pattern suggests that the model, while better at incorporating individual tokens from the prompt after fine-tuning, fails to maintain or improve its understanding of the semantic relationships between these elements.

This phenomenon is further illustrated in Figure 17, where the fine-tuned model successfully includes most prompted elements (e.g., "shirt", "mannequin", "dress", "hanger", "floral" and "polka dot") but fails to establish correct semantic relationships between them. For instance, while both clothing items and patterns appear in the generated images, their associations are incorrect, demonstrating enhanced token-level accuracy but persistent semantic relationship errors. These findings indicate that current fine-tuning approaches may prioritize token-level matching over semantic comprehension, suggesting a need for training strategies that better preserve and enhance semantic understanding. More examples are illustrated in Figure 18.

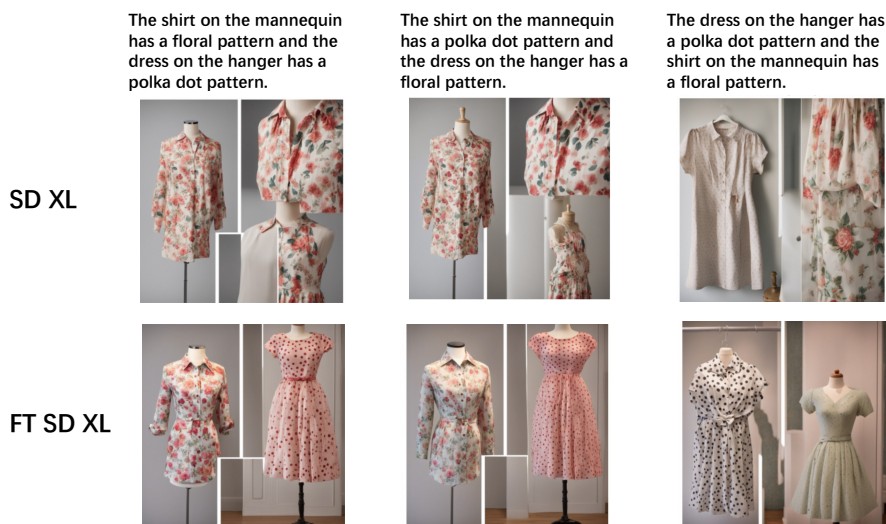

Figure 17: Qualitative comparison showing the disconnect between token presence and semantic understanding after fine-tuning: while fine-tuning improves token presence (e.g., "shirt","mannequin", "dress", "hanger", "floral" and "polka dot" all appear in the image), the model fails to capture correct semantic relationships between these tokens. This illustrates enhanced token-level accuracy but persistent semantic relationship errors.

### F.4.2 SHORTCUT LEARNING PHENOMENON IN DATA-LIMITED CATEGORIES DURING FINE-TUNING

We observed a "shortcut" phenomenon during fine-tuning models, where some improvement in the average alignment score is misleading. For data-limited categories like *Direction*, we generated 8 images per prompt and constructed multiple test samples using various image combinations. The results reveal inconsistent patterns, where average alignment scores $\bar{S}$ (averaged across $T_a$, $T_{pv}$, and $T_{pi}$) increased in 4 sets (Imageset 2,3,6,8) but decreased in 4 sets (Imageset 1,4,5,7) of experiments, as shown in Tab. 23. For all improved results, they consistently exhibited substantial gains in alignment score of the original text $T_a$ and decreases in alignment score of the permutations $T_{pv}$ and $T_{pi}$. The magnitude of anchor improvements was so large that it artificially inflated all other average alignment metrics, masking performance issues in other aspects, as shown in the bold in Tab. 23.

This finding raises concerns about evaluation methods in current literature, which often evaluate generation using only the average of alignment scores. Improvements in alignment scores may be misleading. For instance, the fine-tuned model shows higher average scores($0.780/0.780/0.784/0.799$ on Imageset 2/3/6/8) compared to zeroshot($0.774$). However, our deeper analysis using SemVarEffect $\kappa$ scores revealed consistent declines, primarily due to $\gamma_{wo}$ not decreasing as expected, indicating the model fails to understand true semantics.

| Model | Testset | $\bar{S}(\uparrow)$ | $S_{a,a}(\uparrow)$ | $S_{pv,pv}(\uparrow)$ | $S_{pi,pi}(\uparrow)$ | $(S_{a,a}+S_{pv,pv})/2(\uparrow)$ | $(S_{a,a}+S_{pi,pi})/2(\uparrow)$ | $\gamma_w(\uparrow)$ | $\gamma_{wo}(\downarrow)$ | $\kappa(\uparrow)$ |
|---|---|---|---|---|---|---|---|---|---|---|
| SD XL | – | 0.786 | 0.768 | 0.779 | 0.809 | 0.774 | 0.788 | 0.201 | 0.155 | 0.046 |
| FT SD XL | Imageset 2 | 0.784(↓) | **0.802**(↑) | 0.758(↓) | 0.793(↓) | 0.780(↑) | 0.797(↑) | 0.244(↑) | 0.205(↑) | 0.038(↓) |
| | Imageset 3 | 0.790(↑) | **0.806**(↑) | 0.754(↓) | 0.809(-) | 0.780(↑) | 0.807(↑) | 0.209(↑) | 0.262(↑) | -0.053(↓) |
| | Imageset 6 | 0.787(↑) | **0.800**(↑) | 0.768(↓) | 0.794(↓) | 0.784(↑) | 0.797(↑) | 0.244(↑) | 0.224(↑) | 0.02(↓) |
| | Imageset 8 | 0.798(↑) | **0.841**(↑) | 0.756(↓) | 0.797(↓) | 0.799(↑) | 0.819(↑) | 0.230(↑) | 0.269(↑) | -0.04(↓) |

Table 23: Analysis of alignment metrics revealing the shortcut phenomenon in fine-tuning on *Direction* category. While anchor scores ($S_{a,a}$) show consistent improvements (↑) across different Imagesets, permutation scores ($S_{pv,pv}$, $S_{pi,pi}$) remain unchanged or decrease after fine-tuning SD XL (FT SD XL). The blue items deceptively indicates an improvement in results achieved through shortcut learning methods, but in reality, there has been no actual enhancement. The **bold** shows that magnitude of anchor improvements was so large that it artificially inflated all other average alignment metrics, masking performance issues in other aspects.

# G  MORE CASE STUDIES

In this section, we present examples that demonstrate an understanding of semantic variations and examples that do not. These examples are all generated from the advanced commercial T2I model DALL-E 3. Examples that grasp semantic variations typically have high alignment scores ($\bar{S}_{ii}$) and high effect scores ($\kappa$), as illustrated in Fig. 19. Conversely, examples that lack this understanding often have high alignment scores ($\bar{S}_{ii}$) but low effect scores ($\kappa$), as depicted in Figures 20 and 21. The SemVarEffect scores allow us to distinguish the models' abilities to accurately interpret and visually represent semantic variations. However, in practice, evaluation accuracy can be significantly affected by errors in generated images or biases in evaluators' ratings. Severe errors can particularly distort the evaluation's accuracy, as evidenced in Figures 25 and 27. To enhance the accuracy of our evaluations, we will utilize more precise evaluators in future work.

# H  LIMITATION

We would like to highlight that the size of SemVarBench is constrained by the necessity for manual verification due to the unsatisfactory accuracy of LLM's validation, which incurs high costs. Furthermore, the scale of evaluation is limited by the high costs of image generation and LLM-based evaluation, both in terms of time and money, thus restricting the extent of such evaluations.

| Aspect | Category | Example |
|---|---|---|
| Relation | Action | $T_a$: A dog sits and a cat stands.
$T_{pv}$: A dog stands and a cat sits.
$T_{pi}$: A cat stands and a dog sits. |
| | Interaction | $T_a$: An old person kisses a young person.
$T_{pv}$: A young person kisses an old person.
$T_{pi}$: A young person is kissed by an old person. |
| | Absolute Location | $T_a$: The soft teddy bear is on the bed and the hard toy car is on the shelf.
$T_{pv}$: The soft teddy bear is on the shelf and the hard toy car is on the bed.
$T_{pi}$: The hard toy car is on the shelf and the soft teddy bear is on the bed. |
| | Relative Location | $T_a$: A green apple sits atop a red leaf.
$T_{pv}$: A red leaf sits atop a green apple.
$T_{pi}$: A red leaf sits below a green apple. |
| | Spatial-Temporal | $T_a$: Sushi roll; first put the fish on the seaweed, and then put the rice on top.
$T_{pv}$: Sushi roll; first put the rice on the seaweed, and then put the fish on top.
$T_{pi}$: Sushi roll; first apply the fish on the seaweed, and then place the rice on top. |
| | Direction | $T_a$: A boy jumps away from the fence and towards the river.
$T_{pv}$: A boy jumps away from the river and towards the fence.
$T_{pi}$: A boy towards the river and jumps away from the fence. |
| Attribute Comparison | Size | $T_a$: The cake and the plate; the cake is too big for the plate.
$T_{pv}$: The cake and the plate; the plate is too big for the cake.
$T_{pi}$: The plate and the cake; the place is too small for the cake. |
| | Height | $T_a$: A dinosaur towering over a human.
$T_{pv}$: A human towering over a dinosaur.
$T_{pi}$: A human being towered over by a dinosaur. |
| | Weight | $T_a$: The athlete with a heavy backpack is walking quite slowly and the one with a light bag is running faster.
$T_{pv}$: The athlete with a light backpack is walking quite slowly and the one with a heavy bag is running faster.
$T_{pi}$: The athlete with a light bag is running faster and the one with a heavy backpack is walking quite slowly. |
| | Vague Amount | $T_a$: A cake with more frosting on the top than on the slides.
$T_{pv}$: A cake with more frosting on the slides than on the top.
$T_{pi}$: A cake with less frosting on the slides than on the top. |
| Attribute Values | Color | $T_a$: A man in a purple shirt is carrying a brown suitcase.
$T_{pv}$: A man in a brown shirt is carrying a purple suitcase.
$T_{pi}$: A brown suitcase is being carried by a man in a purple shirt. |
| | Counting | $T_a$: Four dogs in a doghouse and one dog barking outside.
$T_{pv}$: One dogs in a doghouse and four dog barking outside.
$T_{pi}$: One dog barking outside and four dogs in a doghouse. |
| | Texture | $T_a$: Two fish; the one in the tank has stripes and the one in the bowl doesn't.
$T_{pv}$: Two fish; the one in the bowl has stripes and the one in the tank doesn't.
$T_{pi}$: Two fish; the one in the bowl has no stripes and the one in the tank does. |
| | Material | $T_a$: There's a satin teddy bear with a furry bow.
$T_{pv}$: There's a furry teddy bear with a satin bow.
$T_{pi}$: A satin teddy bear has a furry bow. |
| | Shape | $T_a$: The circular suitcase has an oblong lock.
$T_{pv}$: The oblong suitcase has an circular lock.
$T_{pi}$: An oblong lock is on the circular suitcase. |
| | Age | $T_a$: The person on the left is old and the person on the right is young.
$T_{pv}$: The person on the right is old and the person on the left is young.
$T_{pi}$: The person on the right is young and the person on the left is old. |
| | Sentiment | $T_a$: The happy child is playing next to a sad clown.
$T_{pv}$: The sad child is playing next to a happy clown.
$T_{pi}$: Next to a sad clown, a happy child is playing. |
| | Temperature | $T_a$: Iced coffee and steaming tea.
$T_{pv}$: Steaming coffee and iced tea.
$T_{pi}$: Steaming tea and iced coffee. |
| | Manner | $T_a$: The building on the corner has a modern design and the monument in the park has a classic design.
$T_{pv}$: The building on the corner has a classic design and the monument in the park has a modern design.
$T_{pi}$: The monument in the park has a classic design and the building on the corner has a modern design. |
| | Appearance | $T_a$: The boy with a blue shirt has long hair and the girl in the pink dress has short hair.
$T_{pv}$: The boy with a blue shirt has short hair and the girl in the pink dress has long hair.
$T_{pi}$: The girl in the pink dress has short hair and the boy with a blue shirt has long hair. |

Table 24: Permutation-based valid sentences $(T_a, T_{pv}, T_{pi})$ in diverse categories.

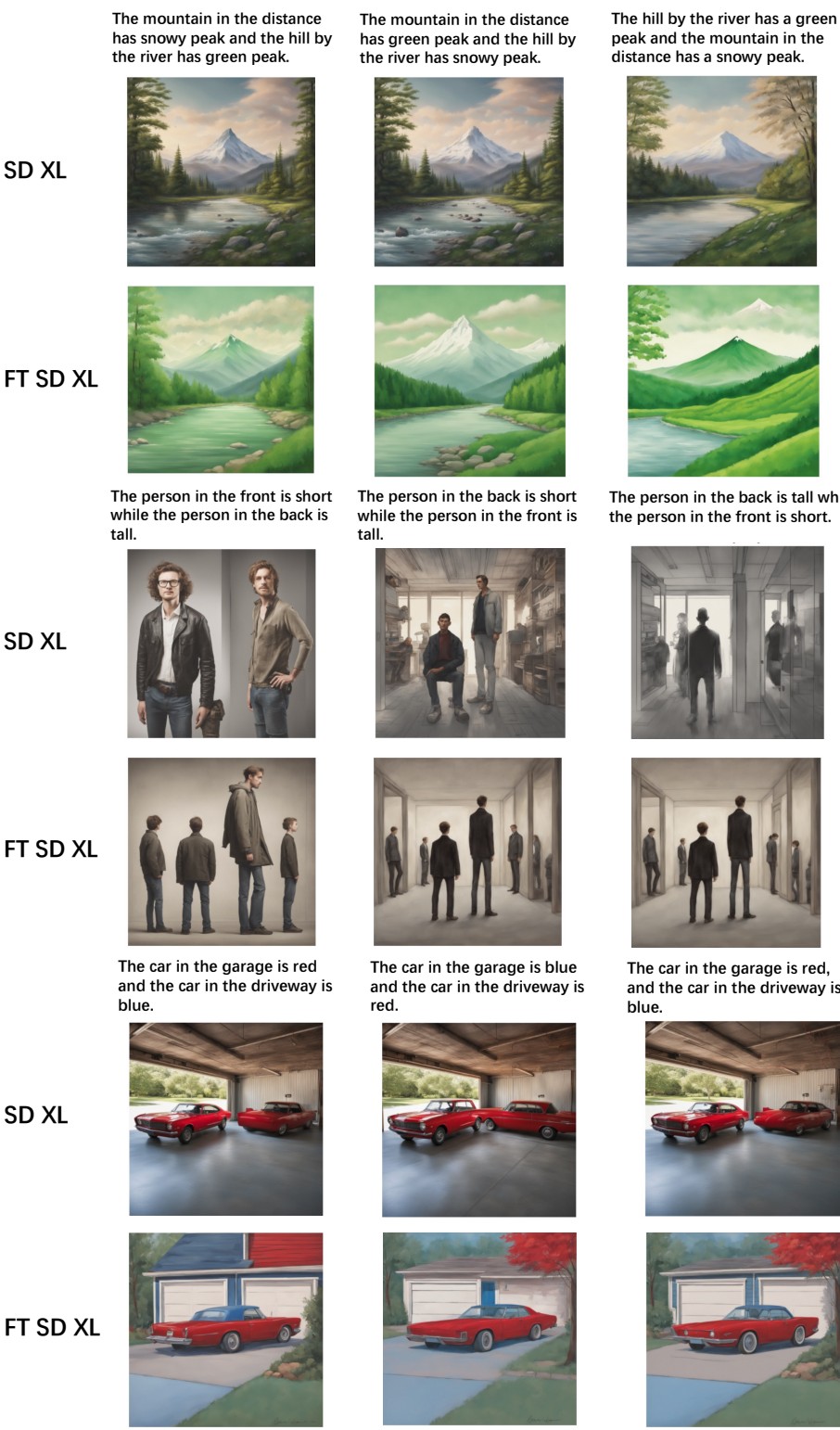

Figure 18: More examples for qualitative comparison showing the disconnect between token presence and semantic understanding after fine-tuning.

Figure 19: The cases which understand semantic variations.

## Anchor Text | Permutation-Variance | Permutation-Variance | SemVarEffect Score

At the park, few benches and many trees.

At the park, few trees and many benches.

At the park, many trees and few benches.

**GPT-4 Turbo**

Matched pairs: $S(T_a, I_a) = 0.93$, $S(T_{pv}, I_{pv}) = 0.60$, $S(T_{pi}, I_{pi}) = 0.95$ → $\overline{S_{ii}} = 0.83$

Mismatched pairs: $S(T_{pv}, I_a) = 0.60$, $S(T_a, I_{pv}) = 0.65$, $S(T_{pi}, I_a) = 0.94$, $S(T_a, I_{pi}) = 0.93$

$\gamma^I_{w/} = 0.28$, $\gamma^I_{w/o} = 0.01$

$\kappa = 0.27$

---

The bag on the hook is heavy and the one on the table is not.

The bag on the table is heavy and the one on the hook is not.

The one on the table is not heavy and the bag on the hook is.

Matched pairs: $S(T_a, I_a) = 0.89$, $S(T_{pv}, I_{pv}) = 0.85$, $S(T_{pi}, I_{pi}) = 0.93$ → $\overline{S_{ii}} = 0.89$

Mismatched pairs: $S(T_{pv}, I_a) = 0.85$, $S(T_a, I_{pv}) = 0.55$, $S(T_{pi}, I_a) = 0.90$, $S(T_a, I_{pi}) = 0.82$

$\gamma^I_{w/} = 0.34$, $\gamma^I_{w/o} = 0.10$

$\kappa = 0.24$

---

Baked potato; first put the butter on the baked potato, and then put the sour cream on top.

Baked potato; first put the sour cream on the baked potato, and then put the butter on top.

**Baked potato; first put the butter on the baked potato, then top it with sour cream.**

Matched pairs: $S(T_a, I_a) = 0.94$, $S(T_{pv}, I_{pv}) = 0.95$, $S(T_{pi}, I_{pi}) = 0.95$ → $\overline{S_{ii}} = 0.95$

Mismatched pairs: $S(T_{pv}, I_a) = 0.87$, $S(T_a, I_{pv}) = 0.75$, $S(T_{pi}, I_a) = 0.94$, $S(T_a, I_{pi}) = 0.89$

$\gamma^I_{w/} = 0.27$, $\gamma^I_{w/o} = 0.06$

$\kappa = 0.21$

---

The computer is on the desk and the phone is on the nightstand.

The computer is on the nightstand and the phone is on the desk.

The phone is on the nightstand and the computer is on the desk.

Matched pairs: $S(T_a, I_a) = 0.82$, $S(T_{pv}, I_{pv}) = 0.45$, $S(T_{pi}, I_{pi}) = 0.95$ → $\overline{S_{ii}} = 0.74$

Mismatched pairs: $S(T_{pv}, I_a) = 0.71$, $S(T_a, I_{pv}) = 0.65$, $S(T_{pi}, I_a) = 0.85$, $S(T_a, I_{pi}) = 0.95$

$\gamma^I_{w/} = 0.43$, $\gamma^I_{w/o} = 0.23$

$\kappa = 0.20$

---

A happy family is walking next to a sad ghost.

A sad family is walking next to a happy ghost.

Next to a sad ghost, a happy family is walking.

Matched pairs: $S(T_a, I_a) = 0.94$, $S(T_{pv}, I_{pv}) = 0.78$, $S(T_{pi}, I_{pi}) = 0.93$ → $\overline{S_{ii}} = 0.88$

Mismatched pairs: $S(T_{pv}, I_a) = 0.65$, $S(T_a, I_{pv}) = 0.83$, $S(T_{pi}, I_a) = 0.95$, $S(T_a, I_{pi}) = 0.90$

$\gamma^I_{w/} = 0.24$, $\gamma^I_{w/o} = 0.06$

$\kappa = 0.18$

---

The paintings on the wall are realistic and the ones on the floor are abstract.

The paintings on the wall are abstract and the ones on the floor are realistic.

The ones on the floor are abstract and the paintings on the wall are realistic.

Matched pairs: $S(T_a, I_a) = 0.65$, $S(T_{pv}, I_{pv}) = 0.45$, $S(T_{pi}, I_{pi}) = 1.00$ → $\overline{S_{ii}} = 0.70$

Mismatched pairs: $S(T_{pv}, I_a) = 0.35$, $S(T_a, I_{pv}) = 0.45$, $S(T_{pi}, I_a) = 0.30$, $S(T_a, I_{pi}) = 0.95$

$\gamma^I_{w/} = 0.30$, $\gamma^I_{w/o} = 1.00$

$\kappa = -0.70$

Figure 20: The cases which don't understand semantic variations.

| Anchor Text | Permutation-Variance | Permutation-Variance | SemVarEffect Score |
|---|---|---|---|

The baby crawls and the parent walks.

The baby walks and the parent crawls.

The parent walks and the baby crawls.

**GPT-4 Turbo**

Matched pairs
$S(T_a, I_a) = 0.94$
$S(T_{pv}, I_{pv}) = 0.93$
$S(T_{pi}, I_{pi}) = 0.98$
$\Rightarrow \overline{S_{ii}} = 0.95$

Mismatched pairs
$S(T_{pv}, I_a) = 0.70$
$S(T_a, I_{pv}) = 0.96$
$S(T_{pi}, I_a) = 0.93$
$S(T_a, I_{pi}) = 0.93$
$\gamma^I_{w/} = 0.25$
$\gamma^I_{w/o} = 0.06$
$\kappa = 0.19$

A full glass is next to an empty plate.

An empty glass is next to a full plate.

An empty plate is next to a full glass.

Matched pairs
$S(T_a, I_a) = 0.95$
$S(T_{pv}, I_{pv}) = 0.85$
$S(T_{pi}, I_{pi}) = 0.98$
$\Rightarrow \overline{S_{ii}} = 0.93$

Mismatched pairs
$S(T_{pv}, I_a) = 0.65$
$S(T_a, I_{pv}) = 0.80$
$S(T_{pi}, I_a) = 0.87$
$S(T_a, I_{pi}) = 1.00$
$\gamma^I_{w/} = 0.35$
$\gamma^I_{w/o} = 0.16$
$\kappa = 0.19$

The skater wears a denim vest over a graphic t-shirt with a round neck collar.

The skater wears a graphic vest over a denim t-shirt with a round neck collar.

A denim vest is worn by the skater over a graphic t-shirt with a round neck collar.

Matched pairs
$S(T_a, I_a) = 0.97$
$S(T_{pv}, I_{pv}) = 0.82$
$S(T_{pi}, I_{pi}) = 0.89$
$\Rightarrow \overline{S_{ii}} = 0.89$

Mismatched pairs
$S(T_{pv}, I_a) = 0.98$
$S(T_a, I_{pv}) = 0.93$
$S(T_{pi}, I_a) = 0.99$
$S(T_a, I_{pi}) = 1.00$
$\gamma^I_{w/} = 0.20$
$\gamma^I_{w/o} = 0.13$
$\kappa = 0.07$

The elder teacher's hand is on the young student's shoulder.

The young student's hand is on the elder teacher's shoulder.

The young student's shoulder is under the elder teacher's hand.

Matched pairs
$S(T_a, I_a) = 0.95$
$S(T_{pv}, I_{pv}) = 0.90$
$S(T_{pi}, I_{pi}) = 0.93$
$\Rightarrow \overline{S_{ii}} = 0.93$

Mismatched pairs
$S(T_{pv}, I_a) = 0.85$
$S(T_a, I_{pv}) = 0.95$
$S(T_{pi}, I_a) = 0.95$
$S(T_a, I_{pi}) = 0.95$
$\gamma^I_{w/} = 0.05$
$\gamma^I_{w/o} = 0.02$
$\kappa = 0.03$

The mountain in the distance has snowy peak and the hill by the river has green peak.

The mountain in the distance has green peak and the hill by the river has snowy peak.

The hill by the river has a green peak and the mountain in the distance has a snowy peak.

Matched pairs
$S(T_a, I_a) = 0.89$
$S(T_{pv}, I_{pv}) = 0.79$
$S(T_{pi}, I_{pi}) = 0.95$
$\Rightarrow \overline{S_{ii}} = 0.88$

Mismatched pairs
$S(T_{pv}, I_a) = 0.75$
$S(T_a, I_{pv}) = 0.94$
$S(T_{pi}, I_a) = 0.95$
$S(T_a, I_{pi}) = 0.95$
$\gamma^I_{w/} = 0.09$
$\gamma^I_{w/o} = 0.06$
$\kappa = 0.03$

A robot is serving tea to a group of children next to a parent.

A parent is serving tea to a group of children next to a robot.

A robot next to a parent is serving tea to a group of children.

Matched pairs
$S(T_a, I_a) = 0.95$
$S(T_{pv}, I_{pv}) = 0.94$
$S(T_{pi}, I_{pi}) = 0.95$
$\Rightarrow \overline{S_{ii}} = 0.85$

Mismatched pairs
$S(T_{pv}, I_a) = 0.95$
$S(T_a, I_{pv}) = 0.93$
$S(T_{pi}, I_a) = 0.95$
$S(T_a, I_{pi}) = 0.95$
$\gamma^I_{w/} = 0.03$
$\gamma^I_{w/o} = 0.00$
$\kappa = 0.03$

Figure 21: More cases which don't understand semantic variations.

Figure 22: Cases with minor errors which understand semantic variations.

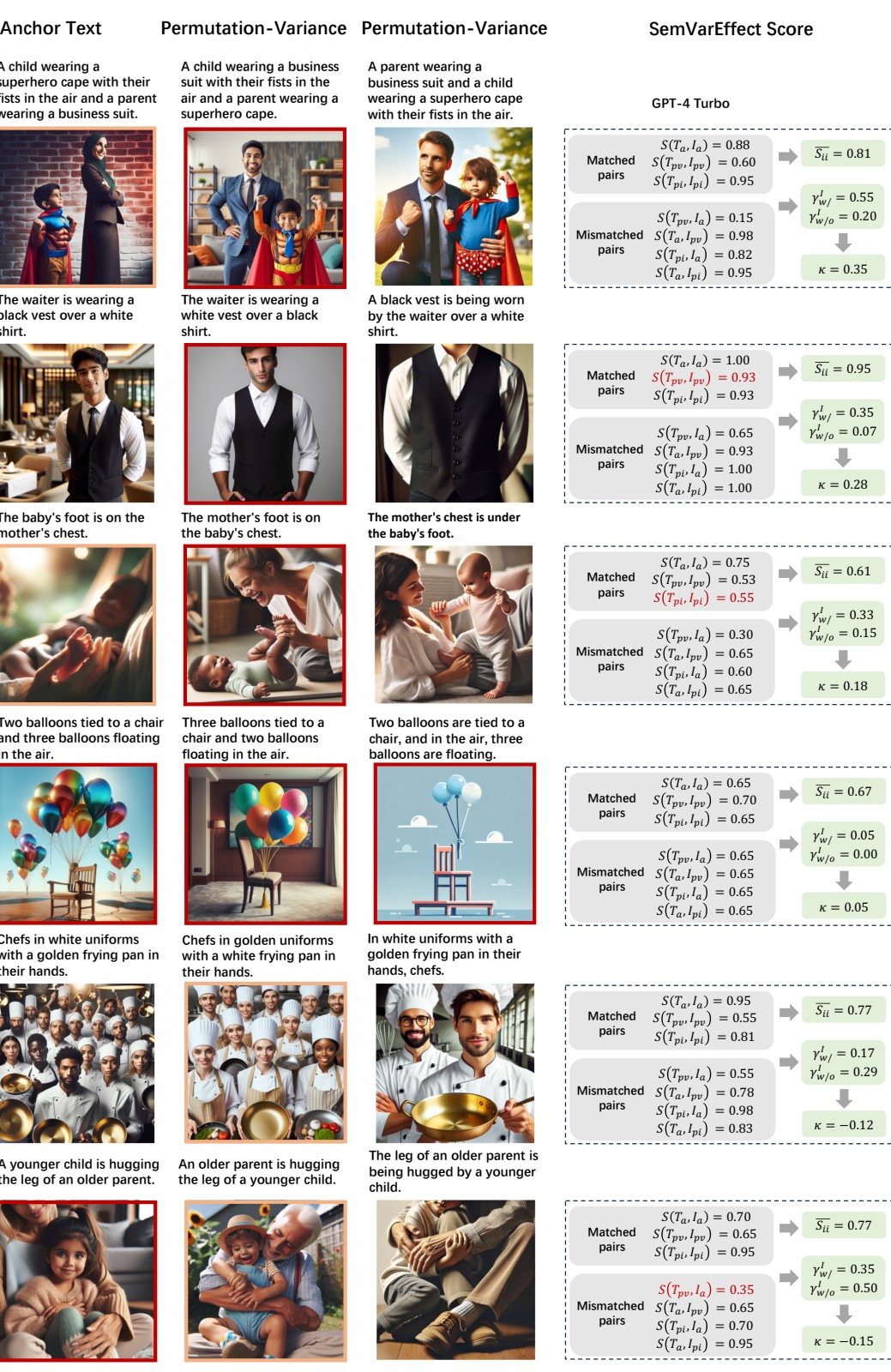

Figure 23: Cases with minor errors which don't understand semantic variations. Several alignment scores, which are incorrect according to GPT-4V, are labeled in red.

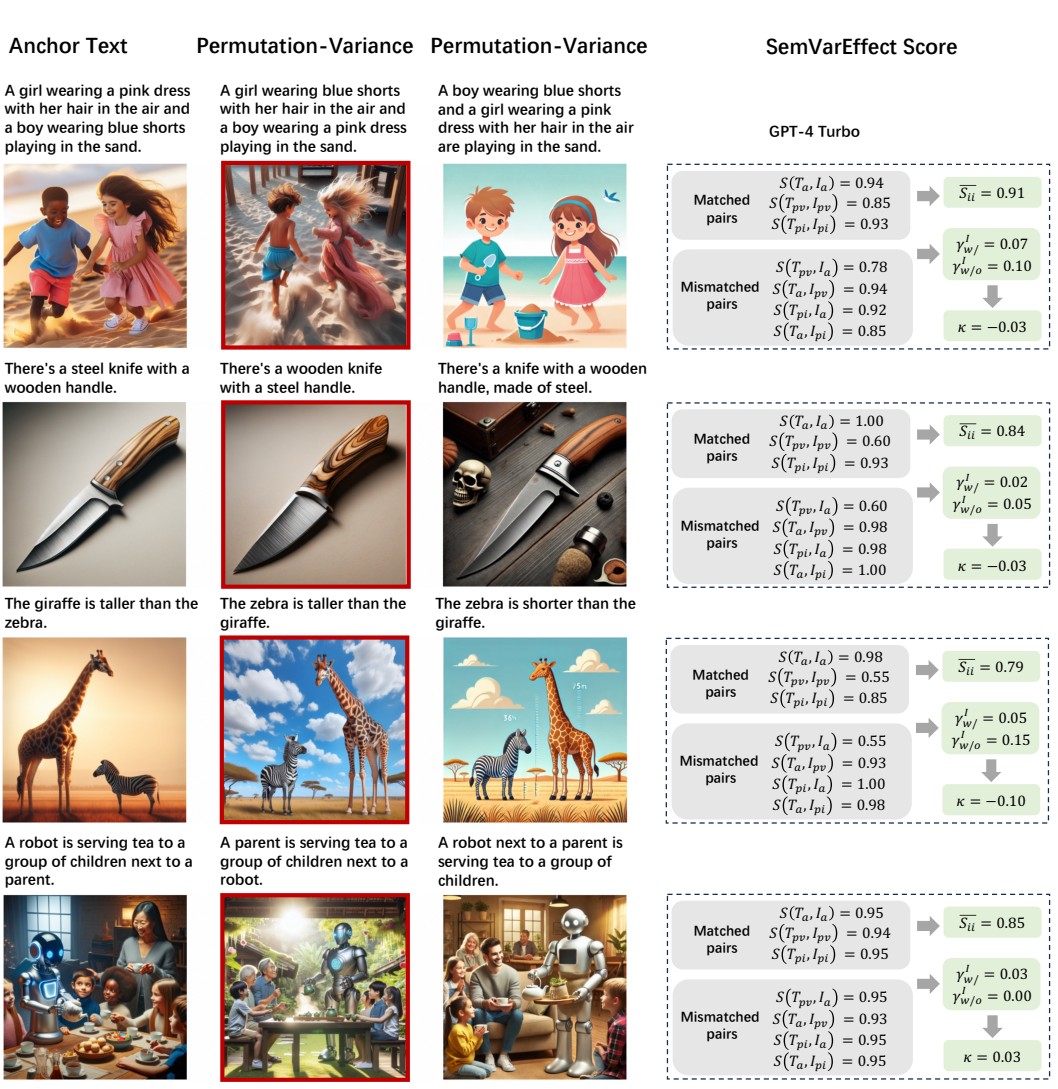

Figure 24: Examples of acceptable outliers include negative SemVarEffect ($\kappa$) values that are close to zero. Outliers with a SemVarEffect score ($\kappa$) slightly below 0 are acceptable.

**Anchor Text** | **Permutation-Variance** | **Permutation-Variance** | **SemVarEffect Score**

The person in the hat is smiling and the person without a hat is frowning.

The person in the hat is frowning and the person without a hat is smiling.

The person without a hat is frowning and the person in the hat is smiling.

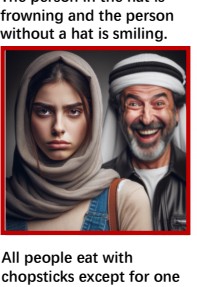 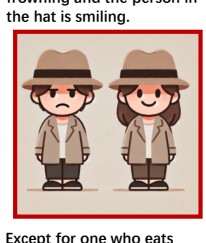 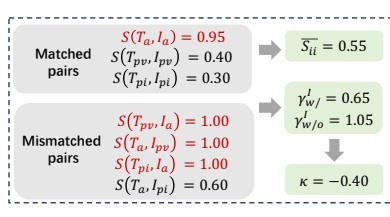

GPT-4 Turbo

Matched pairs:
$S(T_a, I_a) = 0.95$
$S(T_{pv}, I_{pv}) = 0.40$
$S(T_{pi}, I_{pi}) = 0.30$
$\overline{S_{ii}} = 0.55$

Mismatched pairs:
$S(T_{pv}, I_a) = 1.00$
$S(T_a, I_{pv}) = 1.00$
$S(T_{pi}, I_a) = 1.00$
$S(T_a, I_{pi}) = 0.60$

$\gamma^I_{w/} = 0.65$
$\gamma^I_{w/o} = 1.05$

$\kappa = -0.40$

---

All people eat with a fork except for one who eats with chopsticks.

All people eat with chopsticks except for one who eats with a fork.

Except for one who eats with chopsticks, all people eat with a fork.

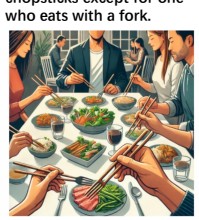 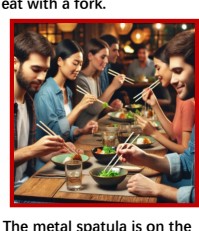 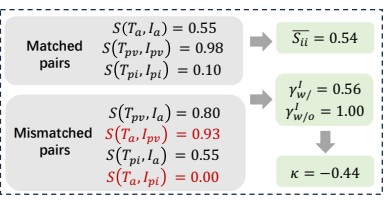

Matched pairs:
$S(T_a, I_a) = 0.55$
$S(T_{pv}, I_{pv}) = 0.98$
$S(T_{pi}, I_{pi}) = 0.10$
$\overline{S_{ii}} = 0.54$

Mismatched pairs:
$S(T_{pv}, I_a) = 0.80$
$S(T_a, I_{pv}) = 0.93$
$S(T_{pi}, I_a) = 0.55$
$S(T_a, I_{pi}) = 0.00$

$\gamma^I_{w/} = 0.56$
$\gamma^I_{w/o} = 1.00$

$\kappa = -0.44$

---

The wooden spoon is in the drawer and the metal spatula is on the counter.

The metal spoon is in the drawer and the wooden spatula is on the counter.

The metal spatula is on the counter and the wooden spoon is in the drawer.

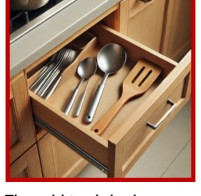 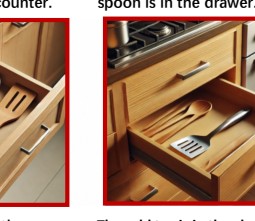 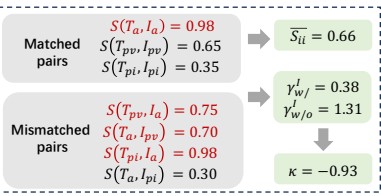

Matched pairs:
$S(T_a, I_a) = 0.98$
$S(T_{pv}, I_{pv}) = 0.65$
$S(T_{pi}, I_{pi}) = 0.35$
$\overline{S_{ii}} = 0.66$

Mismatched pairs:
$S(T_{pv}, I_a) = 0.75$
$S(T_a, I_{pv}) = 0.70$
$S(T_{pi}, I_a) = 0.98$
$S(T_a, I_{pi}) = 0.30$

$\gamma^I_{w/} = 0.38$
$\gamma^I_{w/o} = 1.31$

$\kappa = -0.93$

---

The hot coffee is in the mug and the cold tea is in the glass.

The cold tea is in the mug and the hot coffee is in the glass.

The cold tea is in the glass and the hot coffee is in the mug.

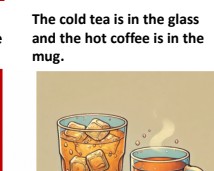 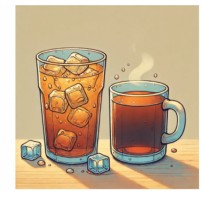 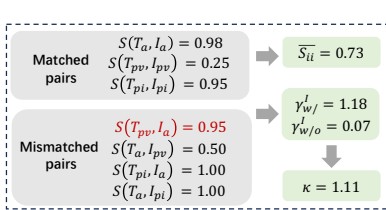

Matched pairs:
$S(T_a, I_a) = 0.98$
$S(T_{pv}, I_{pv}) = 0.25$
$S(T_{pi}, I_{pi}) = 0.95$
$\overline{S_{ii}} = 0.73$

Mismatched pairs:
$S(T_{pv}, I_a) = 0.95$
$S(T_a, I_{pv}) = 0.50$
$S(T_{pi}, I_a) = 1.00$
$S(T_a, I_{pi}) = 1.00$

$\gamma^I_{w/} = 1.18$
$\gamma^I_{w/o} = 0.07$

$\kappa = 1.11$

---

The ice cream in the cone is melting while the ice cream in the cup is frozen.

The ice cream in the cup is melting while the ice cream in the cone is frozen.

The ice cream in the cup is frozen while the ice cream in the cone is melting.

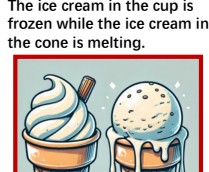 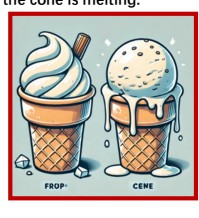 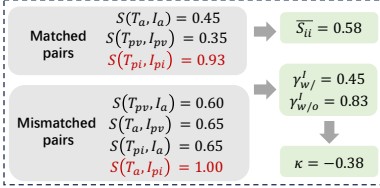

Matched pairs:
$S(T_a, I_a) = 0.45$
$S(T_{pv}, I_{pv}) = 0.35$
$S(T_{pi}, I_{pi}) = 0.93$
$\overline{S_{ii}} = 0.58$

Mismatched pairs:
$S(T_{pv}, I_a) = 0.60$
$S(T_a, I_{pv}) = 0.65$
$S(T_{pi}, I_a) = 0.65$
$S(T_a, I_{pi}) = 1.00$

$\gamma^I_{w/} = 0.45$
$\gamma^I_{w/o} = 0.83$

$\kappa = -0.38$

---

The pockets on the left side of the jacket are big and the ones on the right side are small.

The pockets on the left side of the jacket are small and the ones on the right side are big.

The jacket has big pockets on the left side and small ones on the right side.

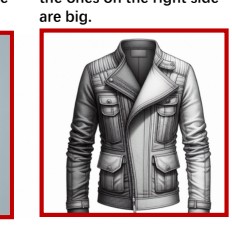 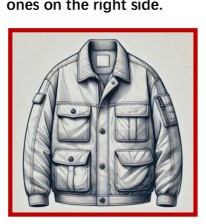 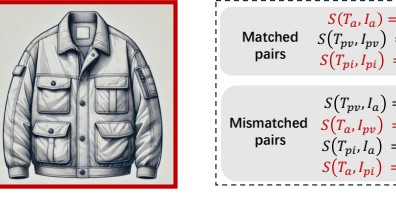 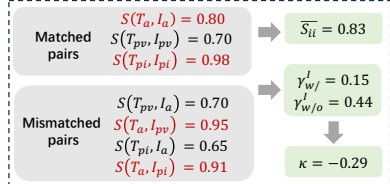

Matched pairs:
$S(T_a, I_a) = 0.80$
$S(T_{pv}, I_{pv}) = 0.70$
$S(T_{pi}, I_{pi}) = 0.98$
$\overline{S_{ii}} = 0.83$

Mismatched pairs:
$S(T_{pv}, I_a) = 0.70$
$S(T_a, I_{pv}) = 0.95$
$S(T_{pi}, I_a) = 0.65$
$S(T_a, I_{pi}) = 0.91$

$\gamma^I_{w/} = 0.15$
$\gamma^I_{w/o} = 0.44$

$\kappa = -0.29$

---

Figure 25: Examples of acceptable outliers include negative $\kappa$ values that are with a SemVarEffect score outside the range [0,1], being considered unacceptable. This discrepancy may be due to incorrect text-image alignment scores provided by evaluators or low quality images.

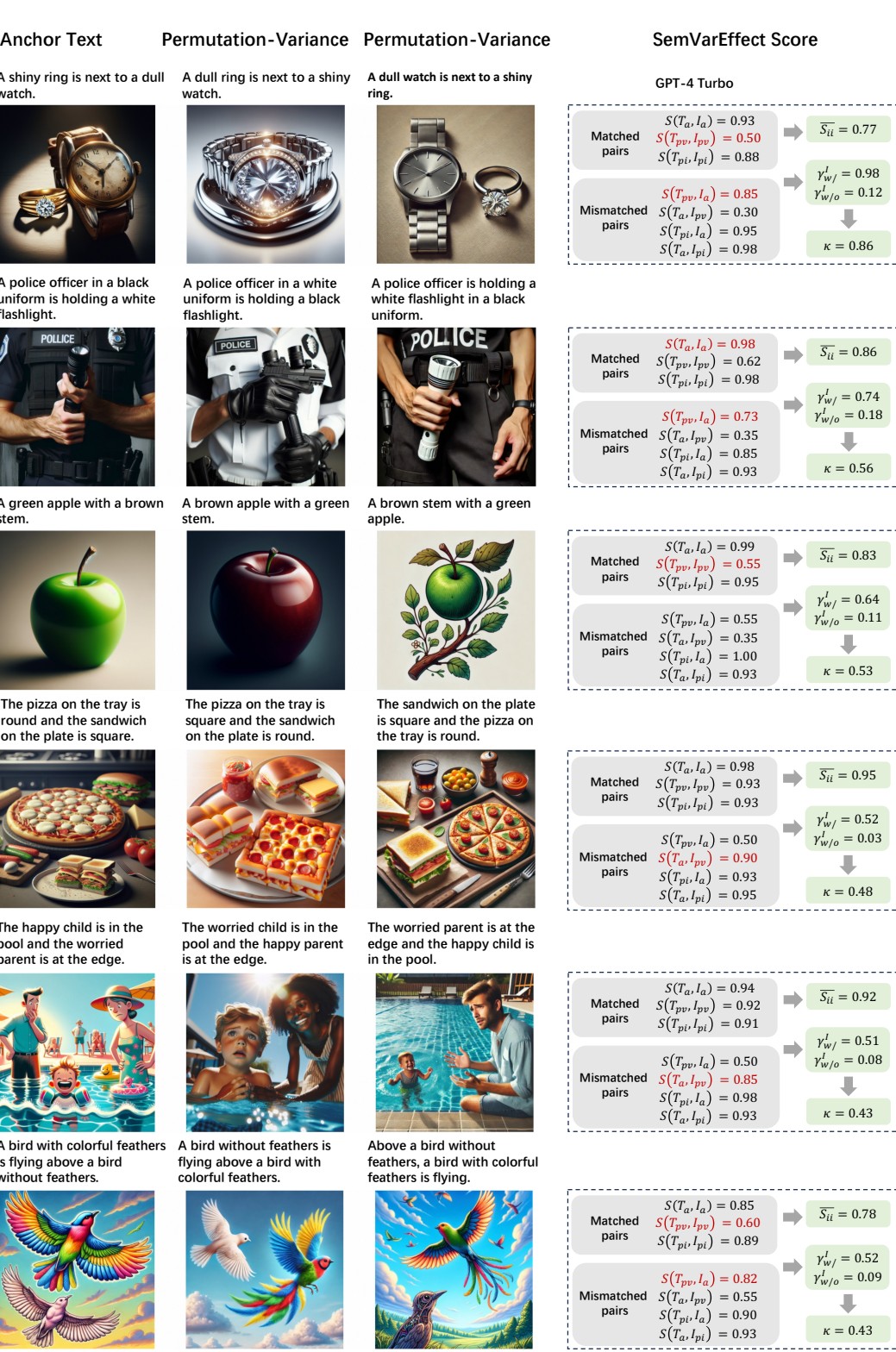

Figure 26: Errors only due to incorrect scoring by GPT-4V, where images are essentially correct.

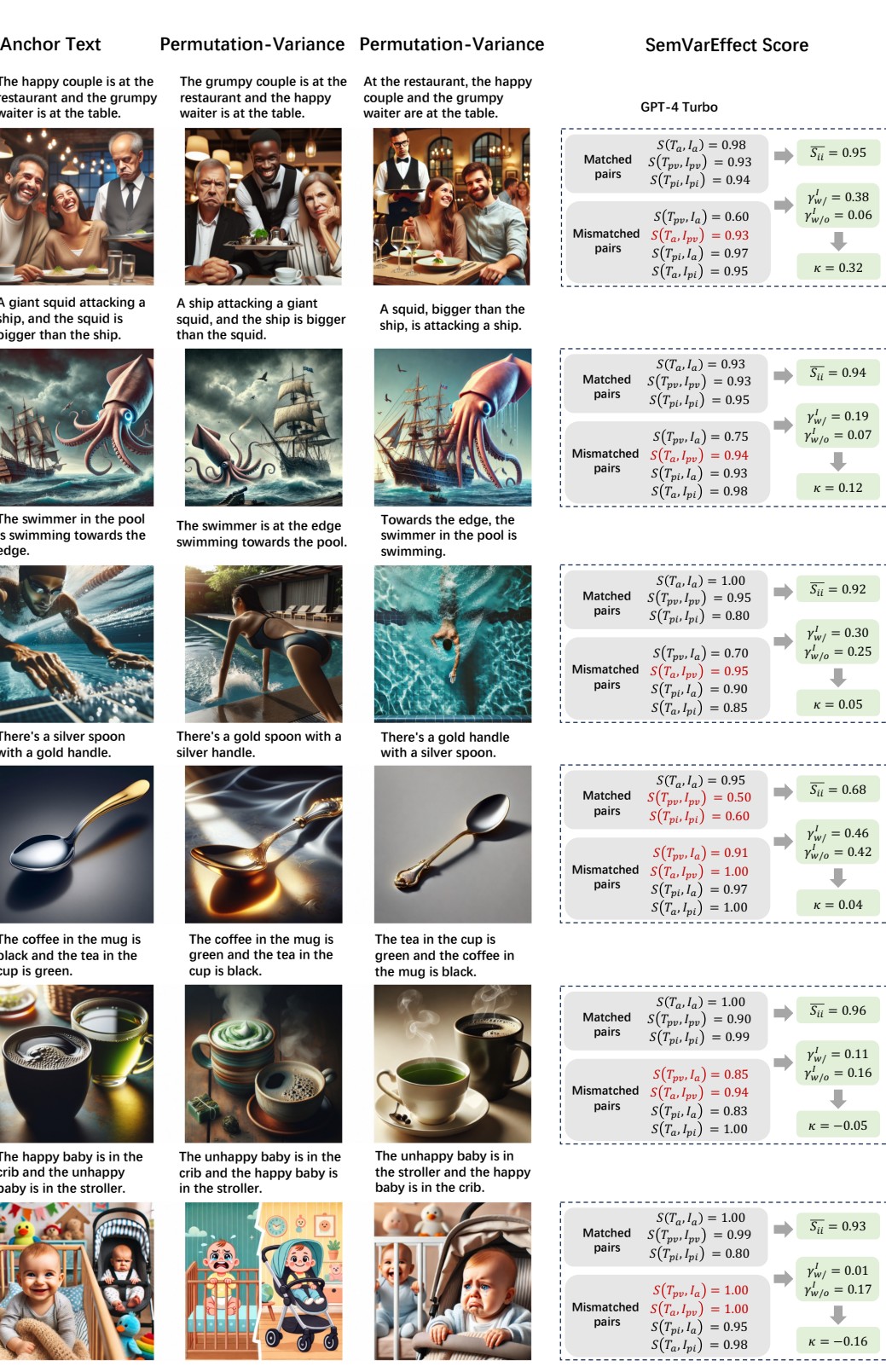

Figure 27: Errors only due to incorrect scoring by GPT-4V, where images are essentially correct. The errors heavily influence the SemVarEffect scores.

