# OpenReview forum: "Evaluating Semantic Variation in Text-to-Image Synthesis: A Causal Perspective"
_ICLR.cc/2025/Conference — ICLR 2025 Poster_

### Official Review · Reviewer_xM6n · 2024-11-02

**Soundness:** 4
**Presentation:** 3
**Contribution:** 4
**Rating:** 8
**Confidence:** 3

**Summary:**

This paper focuses on the performance of Text to Image (T2I) models in handling semantic variations. It identifies that existing T2I models tend to overlook the relationships between words in the input text when faced with similar semantics or semantic substitutions, making it challenging to generate accurate images. At the same time, the paper introduces the SemVarEffect metric and SemVarBench to evaluate this issue.

**Strengths:**

- This paper offers an interesting and novel perspective on the current issues facing T2I models, and provides an effective metric and dataset to measure this issue.
- The paper offers detailed and fair experiments to compare various T2I models.
- The analysis section is thorough and carefully examines the causes of the semantic understanding issues in T2I models, providing a reliable path for future research.

**Weaknesses:**

- I noticed that some of the samples provided in the paper, such as "A mouse chasing a cat," are somewhat unrealistic captions. However, this category is not emphasized in SemVarBench, as seen with sentences like "A cat chasing a dog" and "A dog chasing a cat." These sentences also exhibit semantic variations, but I believe that current T2I models can handle these changes. Therefore, I think further discussion on this issue is needed.
- I believe it is necessary to further discuss the relationship between this phenomenon and the training data in the discussion. In the problematic captions where semantic variations occur, a significant portion of the original captions may be included in the pre-training data. Therefore, similar to the discussion in paper [1], I suggest designing experiments to investigate whether T2I models can perform reverse inference of semantic variations.

[1] Berglund, L., Tong, M., Kaufmann, M., Balesni, M., Stickland, A. C., Korbak, T., \& Evans, O. (2024). *The Reversal Curse: LLMs trained on "A is B" fail to learn "B is A"*. arXiv:2309.12288.

**Questions:**

See Weakness

---

> ### Author Response · Authors · 2024-12-01
>
> I sincerely apologize for this oversight. Though we successfully submitted a revised version several days ago, I failed to verify if the correct version was uploaded. As I cannot locate the revised version in the system now, I have provided an anonymous link [
>  https://anonymous.4open.science/r/submission-41BF/EVALUATING_SEMANTIC_VARIATION_IN_TEXT-TO-IMAGE_SYNTHESIS.pdf ] containing our complete revised manuscript. I deeply appreciate your understanding and time.
>
> **Question 1**: I noticed that some of the samples provided in the paper, such as "A mouse chasing a cat," are somewhat unrealistic captions. However, this category is not emphasized in SemVarBench, as seen with sentences like "A cat chasing a dog" and "A dog chasing a cat." These sentences also exhibit semantic variations, but I believe that current T2I models can handle these changes. Therefore, I think further discussion on this issue is needed.
>
> **Response 1**: We appreciate the reviewer's insightful observation about scenario plausibility. Our experimental findings suggest a different conclusion.
>
> **While "A cat chasing a dog" and "A dog chasing a cat" are both plausible scenarios in real life, our rigorous experiments reveal that current T2I models still struggle significantly with such semantic variations.**
>
> **We updated detailed experiments and analysis in Appendix F.3.3 in the revised version.**
>
> Details are as follows.
>
>   * Experiment Setup:
>     * Using advanced commercial T2I models (DALL-E 3 and CogView3-Plus)
>     * Test data:  "a cat chasing a dog" and "A dog chasing a cat"
>     * Generated 30 images for each plausible prompt
>     * Evaluated based on strict criteria:
>       * Both animals must be present
>       * Running motion
>       * Same direction movement
>       * Clear spatial relationships (chaser behind chased)
>       * Proper chase interaction
>  * Results
>    * The results show striking asymmetric performance:
>      * For "A dog chasing a cat":
>        * DALL-E 3: 70% accuracy (21/30 correct)
>        * CogView3-Plus: 26.7% accuracy (8/30 correct)
>      * For "A cat chasing a dog": Performance dropped dramatically
>        * DALL-E 3: only 13.3% accuracy (4/30 correct)
>        * CogView3-Plus: 0% accuracy (0/30 correct)
>      * The majority of failure cases fall into two types
>        * No Interaction (objects appearing independently)
>        * Reversed Roles (inability to distinguish who chases whom)
>    * Conclusion
>      * This stark performance gap demonstrates that T2I models still struggle with semantic role reversals, even for equally plausible scenarios.
>      * The issue persists regardless of scenario plausibility, indicating the core challenge lies in semantic relationship understanding rather than scenario plausibility.
>
>
> Table 18 (in the revised pdf): Error analysis of advanced commercial text-to-image models (DALL-E 3 and Cogview3-Plus) on directional relationship generation.
> | Class  | Reasons            | DALL-E 3          | DALL-E 3 | Cogview3-Plus | Cogview3-Plus |
> |--------|-------------------|-------------------|----------|---------------|---------------|
> |        |                   | cat→dog           | dog→cat  | cat→dog       | dog→cat       |
> | Wrong  | Miss Objects      | 0                 | 0        | 0             | 0             |
> |        | No Interaction    | 17                | 5        | 12            | 22            |
> |        | Wrong Interaction | 0                 | 0        | 0             | 0             |
> |        | Wrong Direction   | 0                 | 3        | 0             | 0             |
> |        | Reversed Role     | 9                 | 1        | 18            | 0             |
> | Right  | Partial/Full Match| 4                 | 21       | 0             | 8             |

---

> ### Author Response · Authors · 2024-12-01
>
> **Question 2**: I believe it is necessary to further discuss the relationship between this phenomenon and the training data in the discussion. In the problematic captions where semantic variations occur, a significant portion of the original captions may be included in the pre-training data. Therefore, similar to the discussion in paper [1], I suggest designing experiments to investigate whether T2I models can perform reverse inference of semantic variations.
> [1] Berglund, L., Tong, M., Kaufmann, M., Balesni, M., Stickland, A. C., Korbak, T., \& Evans, O. (2024). The Reversal Curse: LLMs trained on "A is B" fail to learn "B is A". arXiv:2309.12288.
>
> **Response 2**: We appreciate the connection to the reversal curse phenomenon.
>
> **We specifically investigated whether the model's struggles stem from training data imbalance (rather than a fundamental limitation in relationship understanding).**
>
> **We updated detailed experiments and analysis in Line 486-498 (details in Appendix F.3.4) in the revised version.**
>
> Details are as follows.
>
> We conducted controlled experiments with carefully balanced training data:
>
>   * Experimental Design:
>     * Fine-tuned SDXL with a human-crafted balanced dataset:
>         * cat$\leftrightarrow$dog interactions (80 images per direction)
>     * Tested generalization on two types of unseen prompts:
>         * Anti-commonsense scenarios (mouse$\leftrightarrow$cat)
>         * Plausible scenarios (bull$\leftrightarrow$man)
>
>   * Results:
>     * Despite balanced training, generalization remained poor:
>         * mouse$\rightarrow$cat vs cat$\rightarrow$mouse: 3/30, 3/30 accuracy
>         * bull$\rightarrow$man vs man$\rightarrow$bull: 4/30, 5/30 accuracy
>     * Failure patterns include:
>         * Relationship Understanding Failures (no/wrong interactions)
>         * Reversed Roles
>         * Missing Objects
>     * Correct relationships occurred less frequently than failure cases
>
>   * Analysis:
>     * Poor performance persists even with:
>         * Perfectly balanced training data
>         * Simple directional relationships
>         * Common objects and plausible scenarios
>     * This suggests the core issue lies in relationship understanding rather than data imbalance
>
>   * Conclusion:
>     * T2I models' struggles with semantic relationships persist even with balanced training, pointing to a more fundamental limitation in relationship-level understanding.
>
> Table 19 (in the revised pdf): Testing performance on novel mouse↔cat pairs after fine-tuning SD XL on balanced cat↔dog data.
> | Class  | Reasons            | SD XL              |     | FT SD XL (trained on cat↔dog)   |  |
> |--------|-------------------|-------------------|-----------|-------------|-------------|
> |        |                   | mouse→cat         | cat→mouse | mouse→cat   | cat→mouse   |
> | Wrong  | Missing Objects   | 12                | 14        | 12          | 21          |
> |        | No Interaction    | 4                 | 5         | 2           | 1           |
> |        | Wrong Interaction | 3                 | 1         | 4           | 2           |
> |        | Wrong Direction   | 7                 | 8         | 4           | 1           |
> |        | Reversed Role     | 2                 | 0         | **5**           | **2**           |
> | Right  | Partial/Full Match| 0                 | 2         | **3**       | **3**       |
>
>
> Table 20 (in the revised pdf): Testing performance on novel bull↔man pairs after fine-tuning SD XL on balanced cat↔dog data.
> | Class  | Reasons            | SD XL              |   | FT SD XL (trained on cat↔dog)   |   |
> |--------|-------------------|-------------------|-----------|-------------|-------------|
> |        |                   | bull→man          | man→bull  | bull→man    | man→bull    |
> | Wrong  | Missing Objects   | 0                 | 0         | 0           | 0           |
> |        | No Interaction    | 6                 | 12        | 6          | 13          |
> |        | Wrong Interaction | 5                 | 2         | 0           | 0           |
> |        | Wrong Direction   | 16                | 12        | 11           | 9           |
> |        | Reversed Role     | 2                 | 2         | **9**           | **3**           |
> | Right  | Partial/Full Match| 1                 | 2         | **4**       | **5**       |

---

> ### Author Response · Authors · 2024-12-03
>
> We sincerely appreciate your time and effort in reviewing this work, especially for bringing up the important reference about reversal curse, which inspired us to investigate similar directional semantic relationship issues in T2I models. As the authors-reviewer discussion phase draws to a close, we would appreciate your feedback on whether our responses have adequately addressed your concerns.
>
> If you find any limitations in our experimental design or areas that need further improvement, we would be very grateful for your suggestions. While some changes may not be feasible during the current review period, your feedback would be invaluable for our future work and continued improvement of this research direction. Thank you for your thoughtful consideration.

---

### Official Review · Reviewer_kcHM · 2024-11-03

**Soundness:** 2
**Presentation:** 3
**Contribution:** 2
**Rating:** 5
**Confidence:** 4

**Summary:**

The paper primarily focuses on the impact of word order variations in text prompt on text-to-image synthesis. It proposes a metric to evaluate a model's semantic sensitivity to changes in word order. The main idea is to calculate the difference in visual semantics of generated images between semantic-variant and semantic-invariant text rearrangements. Additionally, a dataset of approximately 11k samples has been collected as an evaluation benchmark, and popular models have been assessed and analyzed based on this benchmark.

**Strengths:**

- The authors find that existing text-to-image synthesis models struggle to capture semantic shifts caused by word order changes and proposed an evaluation metric and benchmark.
- The SemVarEffect metric is based on the difference between semantic-variant and semantic-invariant text rearrangements, similar to the differential denoising principle in circuits, which I find quite novel.
- The paper is well-organized and easy to understand.

**Weaknesses:**

- The definition of visual semantics: The primary contribution of this work is the metric to measure the impact of word order in text prompts on text-to-image synthesis. The basic principle is to alter the text and observe the resulting visual semantic changes in the generated image. However, within this metric, the “visual semantics” of the generated image is defined through text-image similarity, which may not be entirely accurate. Visual semantics in an image typically refers to the content within the image itself, such as object categories, attributes, or scene information. In contrast, text-image similarity only measures the alignment of high-level features between the two modalities; it does not represent the actual visual semantics. Ideally, visual semantics should be defined independently of text.
- The mechanism for ensuring that the generated data is semantic-variant or semantic-invariant during benchmark creation seems to be insufficiently specified, and why are some data extraction templates and rules created using LLMs while others are crafted manually? What challenges are involved in this?

**Questions:**

- In section 4.1, If text-image similarity is obtained using MLLMs, could biases inherent in the MLLM itself (insensitivity to changes in word order) affect the evaluation results?
- Text-to-image synthesis models have difficulty accurately reflecting semantic changes caused by variations in word order, possibly due to certain commonsense biases inherent in the training data. This could be seen as a form of “built-in bias.” For example, as illustrated in Figure 1 of the paper, “A cat chasing a mouse” is a reasonable depiction, whereas the reordered phrase “A mouse chasing a cat” is less plausible, leading to poorer generation results. It might be helpful to add related experiments to validate this hypothesis.

---

> ### Author Response · Authors · 2024-12-01
>
> I sincerely apologize for this oversight. Though we successfully submitted a revised version several days ago, I failed to verify if the correct version was uploaded. As I cannot locate the revised version in the system now, I have provided an anonymous link [
>  https://anonymous.4open.science/r/submission-41BF/EVALUATING_SEMANTIC_VARIATION_IN_TEXT-TO-IMAGE_SYNTHESIS.pdf ] containing our complete revised manuscript. I deeply appreciate your understanding and time.
>
> **Question 1**: The definition of visual semantics: The primary contribution of this work is the metric to measure the impact of word order in text prompts on text-to-image synthesis. The basic principle is to alter the text and observe the resulting visual semantic changes in the generated image. However, within this metric, the “visual semantics” of the generated image is defined through text-image similarity, which may not be entirely accurate. Visual semantics in an image typically refers to the content within the image itself, such as object categories, attributes, or scene information. In contrast, text-image similarity only measures the alignment of high-level features between the two modalities; it does not represent the actual visual semantics. Ideally, visual semantics should be defined independently of text.
>
> **Response 1**: We appreciate the reviewer's insightful observation about visual semantics definition.
>  * ***In text-to-image (T2I) synthesis, image generation is inherently text-guided rather than a pure visual semantic problem.***
>    * Text-independent visual semantics evaluation works well only for well-defined dimensions, such as:
>      * rotational invariance in image processing
>      * translational invariance in robotics
>    * Text-conditioned visual semantic evaluation has become the mainstream approach across various tasks
>      * Text-guided image editing [1]
>      * Text-image similarity [2]
>  * **In general T2I synthesis, the significance of variations inherently depends on text prompts**
>    * Multiple varying dimensions: style, composition, details
>    * Without text reference, determining meaningful visual variations becomes challenging.
>      * Example: in "a dog chasing a squirrel"
>        * "Three dogs" = semantic error
>        * Different backgrounds ("moon"/"forest") = acceptable variation
>
> [1] Ma Y, Ji J, Ye K, et al. I2EBench: A Comprehensive Benchmark for Instruction-based Image Editing[J]. arXiv preprint arXiv:2408.14180, 2024.
>
> [2] Wang T, Lin K, Li L, et al. Equivariant similarity for vision-language foundation models[C]//ICCV. 2023.
>
> **Question 2**: The mechanism for ensuring that the generated data is semantic-variant or semantic-invariant during benchmark creation seems to be insufficiently specified, and why are some data extraction templates and rules created using LLMs while others are crafted manually? What challenges are involved in this?
>
> **Response 2**: Thank you for this important question. We would like to clarify our approach. **All templates and rules are manually crafted.** The reason for manually extracting templates and rules is **the low LLM accuracy for this task**. That's not a big number. We can handle it manually.
>
> **LLMs are only used for large-scale generation based on these pre-defined templates and rules.** They are not used for extracting templates and rules. The challenges in the sentence generation based on manually extracted templates and rules are:
>
>  * **Challenges**:
>    *  **LLMs tend to produce common patterns over specified uncommon combinations**
>       * $T_a$ and $T_{pi}$ generation is typically accurate
>       * $T_{pv}$ often violates modification rules, resulting in sentence with the same semantics by using passive voice
>    *  **LLMs have difficulty adhering to our varying requirements across different components**
>       *  Varying requirements: passive voice allowed in $T_{pi}$ but forbidden in $T_{pv}$
>
>  * **Solutions**:
>    * **To Follow the Template & Rule: We decomposed the generation process into simpler sub-tasks**:
>      * Template-guided Generation for the raw text $T_a$
>      * Rule-guided Permutation for one permutation with semantic changes $T_{pv}$
>      * Paraphrasing-guided Generation for one permutation with semantic maintenance $T_{pi}$
>    * **To Ensure the Semantic Variations: We verified semantic variations through post-generation verification, consisting of**:
>      * Machine pre-filtering
>      * Human verification

---

> ### Author Response · Authors · 2024-12-01
>
> **Question 3**: In section 4.1, If text-image similarity is obtained using MLLMs, could biases inherent in the MLLM itself (insensitivity to changes in word order) affect the evaluation results?
>
> **Response 3**: We appreciate the reviewer's concern about potential MLLM biases affecting evaluation results.
>
> While we acknowledge that MLLMs may have inherent biases including word order insensitivity, their impact on our study's conclusions is minimal for several reasons:
>
> * **Our methodology is specifically designed to focus on substantial visual differences rather than subtle visual variations**
>   * We evaluate visually apparent semantic relationship failures (e.g., missing objects, wrong attributes, reversed roles)
>   * During data construction and annotation, we only included samples with significant visual distinctions among the 3 sentences (the original text and its two permutations), ensuring clear differences rather than subtle nuances
>
> * **We validate MLLM evaluations with human evaluation.**
>   * Human Evaluation Results
>     * Experiment Setting
>        * Testsets: 80 stratified samples from the testset
>        * Models: 4 advanced T2I models
>        * Evaluators: 3 raters evaluating all samples
>    * Results & Conclusion
>      * Consistent performance trends between human and MLLM evaluations
>      * Moderate correlation coefficients with human judgments (Pearson's, Spearman's, Cohen's Kappa)
>    * Our human evaluation study demonstrates MLLM's reliability
>
> Table 10 (in revised version): Evaluation human rating results of different T2I models in understanding semantic variations.
> | T2I Models      | $\bar{S}(\uparrow)$ | $\gamma_{w}(\uparrow)$ | $\gamma_{wo}(\downarrow)$ | $\kappa(\uparrow)$ |
> |------------|-------|--------|---------|-------|
> | MidJ V6    | 0.59  | 0.37   | 0.49    | -0.12 |
> | DALL-E 3   | 0.63  | 0.53   | 0.50    | 0.03  |
> | CogV3-Plus | 0.69  | 0.52   | 0.33    | 0.19  |
> | Ideogram 2 | 0.74  | 0.50   | 0.31    | 0.19  |
>
>
> Table 11 (in revised version): Correlation coefficients between GPT-4o/v, Claude-3.5-Sonnet, Gemini-Pro-1.5 and human evaluations on SemVarEffect of CogView3-Plus.
> | Evaluators      | $\rho(\uparrow)$ | $\phi(\uparrow)$ | $\kappa_{cohen}(\uparrow)$ |
> |-----------------|-------|-------|-----------|
> | GPT-4o          | 0.54  | 0.53  | 0.53      |
> | GPT-4v          | 0.50  | 0.51  | 0.54      |
> | Claude-3.5-Sonnet| 0.42  | 0.37  | 0.37      |
> | Gemini-Pro-1.5  | 0.27  | 0.11  | 0.23      |

---

> ### Author Response · Authors · 2024-12-01
>
> **Question 4**: Text-to-image synthesis models have difficulty accurately reflecting semantic changes caused by variations in word order, possibly due to certain commonsense biases inherent in the training data. This could be seen as a form of “built-in bias.” For example, as illustrated in Figure 1 of the paper, “A cat chasing a mouse” is a reasonable depiction, whereas the reordered phrase “A mouse chasing a cat” is less plausible, leading to poorer generation results. It might be helpful to add related experiments to validate this hypothesis.
>
> **Response 4**: We appreciate this insightful observation but respectfully disagree that commonsense bias is the primary cause.
>
> **Our experiments demonstrate that even after addressing commonsense bias, models still struggle with semantic relationships**.
>
> **We updated detailed experiments and analysis in Appendix F.3.2 in the revised version.**
>
> --------------------------- Details are as follows. ---------------------------
>
> * If commonsense bias were the main factor:
>   * Introducing anti-commonsense training data should mitigate the issue
>   * Models should perform better on plausible scenarios after balanced training
>
> * Our Experiments:
>   * Training: We fine-tuned SD XL on a carefully balanced dataset with high quality
>     * 40 plausible "cat chasing mouse" images (human selected)
>     * 40 anti-commonsense "mouse chasing cat" images (human-crafted)
>
>   * Results show persistent issues despite balanced training:
>     * Training Performance:
>       * For "mouse chasing cat":
>         * Correct cases improved (0/30 $\rightarrow$ 10/30)
>         * Role reversal errors also increased (2/30 $\rightarrow$ 6/30, more images showing "cat chasing mouse")
>
>     * Testing Performance on Plausible Senarios:
>       * Performance on new pairs remained poor despite marginal improvement
>       * For "bull chasing man" and "man chasing bull"
>         * Correct cases marginally improved:
>           * 1/30 $\rightarrow$ 2/30; 2/30 $\rightarrow$ 3/30
>         * Wrong cases for No Interaction or Reversed Role  increase.
>           * No Interaction: 6-12/30 $\rightarrow$ 14-15/30
>           * Reversed Role: 2-2/30 $\rightarrow$ 2-6/30
>       * Similar failure patterns regardless of scenario plausibility
>
>  * Conclusion:
>    * **These results suggest the core challenge isn't commonsense bias**
>      * Even with balanced anti-commonsense training data, models fail to learn and generalize directional relationships.
>    * **The issue appears to be the models' inability to understand and represent "who does what to whom"**.
>
>
>
> Table 15 (in the revised pdf): Training performance of SD XL before and after fine-tuning
> | Class | Reasons | SD XL |  | FT SD XL (trained on mouse↔cat) |  |
> |--------|----------|---------|------------|------------------------|------------|
> |        |          | mouse→cat | cat→mouse | mouse→cat | cat→mouse |
> | Wrong | Missing Objects | 12 | 14 | 4 | 2 |
> |  | No Interaction | 3 | 1 | 7 | 7 |
> |  | Wrong Interaction | 4 | 5 | 3 | 2 |
> |  | Wrong Direction | 7 | 8 | 0 | 5 |
> |  | Reversed Role | 2 | 0 | 6 | 0 |
> | Right | Partial/Full Match | 0 | 2 | 10 | 14 |
>
> Table 16 (in the revised pdf): Testing fine-tuned models' performance on plausible senario
>
> | Class | Reasons | SD XL |  | FT (trained on mouse↔cat) |  |
> |--------|----------|---------|------------|------------------------|------------|
> |        |          | bull→man | man→bull | bull→man | man→bull |
> | Wrong | Missing Objects | 0 | 0 | 0 | 0 |
> |  | No Interaction | 6 | 12 | 14 | 15 |
> |  | Wrong Interaction | 5 | 2 | 4 | 3 |
> |  | Wrong Direction | 16 | 12 | 8 | 3 |
> |  | Reversed Role | 2 | 2 | 2 | 6 |
> | Right | Partial/Full Match | 1 | 2 | 2 | 3 |

---

> ### Author Response · Authors · 2024-12-03
>
> We sincerely appreciate your insightful questions. We hope our detailed responses with additional experiments and analysis have adequately addressed your concerns. As the authors-reviewer discussion phase draws to a close, we would appreciate your feedback on the following aspects:
>
> * Q1 - The necessity of text-guided visual semantics evaluation in T2I synthesis
> * Q2 - The clarification in manual template extraction and challenges in LLM-based generation process
> * Q3 - The reliability validation through human evaluation
> * Q4 - The new experiments demonstrating that commonsense bias is not the primary factor
>
> If you find any limitations in our experimental design or areas that need further improvement, we would be very grateful for your suggestions. While some changes may not be feasible during the current review period, your feedback would be invaluable for our future work and continued improvement of this research direction. Thank you for your thoughtful consideration.

---

> > ### Comment · Reviewer_kcHM · 2024-12-03
> >
> > Hi,
> > Thank you for your detailed reply, which has resolved some of my doubts. I am very sorry and I will still maintain my original score.
> >
> > Best wishes.

---

> > > ### Author Response · Authors · 2024-12-03
> > >
> > > Thank you for your feedback. While we respect your decision about the score, we would be very grateful if you could share any remaining concerns or specific aspects that still need improvement, as this would be valuable guidance for our future work.
> > >
> > > Thank you again for your time and consideration.

---

### Official Review · Reviewer_nmou · 2024-11-04

**Soundness:** 3
**Presentation:** 3
**Contribution:** 3
**Rating:** 6
**Confidence:** 4

**Summary:**

This paper proposes a novel metric called SemVarEffect and a benchmark called SemVarBench to evaluate text-to-image (T2I) models' ability to capture semantic variations caused by word order changes in input prompts. The key contributions are:
- A causal inference framework to measure how semantic variations in text inputs affect visual semantic variations in generated images.
- A carefully curated dataset of text triplets (anchor, permutation with changed meaning, permutation with unchanged meaning) to test T2I models.
- Comprehensive evaluation of  state-of-the-art T2I models. This also includes fine-grained Analysis of the role of text encoders and cross-modal alignment in handling semantic variations.

**Strengths:**

Strengths:
Novel and important problem formulation: The paper addresses a critical gap in evaluating T2I models' language understanding capabilities.

The causal inference based framework provides a principled way to isolate the effect of semantic variations.

SemVarBench is carefully curated to avoid easily predictable variations and focus on challenging semantic permutations, with comprehensive coverage of different type of variations. LLM generated results were checked.

**Weaknesses:**

The training dataset is generated using the same base SDXL model and filtered by MMLM, which could introduce self-distillation artifacts and bias, particularly affecting performance on categories with less data.

The paper suggests that fine-tuning improves alignment at the word level rather than semantic variants, but doesn't provide sufficient discussion or analysis to support this claim

**Questions:**

How does the choice of filtering training data using MMLM potentially affect the evaluation results, particularly for categories with limited samples?

With the 4 metics you had mentioned, which one is the main metric that would make sense quickly for readers? It is a little hard to follow with different metrics presented at different parts, which I can imagine is a little confusing especially for readers who are looking for a single metric for benchmarking.

---

> ### Author Response · Authors · 2024-12-01
>
> We sincerely apologize for this oversight. Though we successfully submitted a revised version several days ago, we failed to verify if the correct version was uploaded. As we cannot locate the revised version in the system now, we have provided an anonymous link [ https://anonymous.4open.science/r/submission-41BF/EVALUATING_SEMANTIC_VARIATION_IN_TEXT-TO-IMAGE_SYNTHESIS.pdf ] containing our complete revised manuscript. We deeply appreciate your understanding and time.
>
> **Question 1**: The training dataset is generated using the same base SDXL model and filtered by MMLM,
> which could introduce self-distillation artifacts and bias, particularly affecting performance on
> categories with less data.
>
> **Response 1**: Thank you for this important question.
>
> To address the reviewer's concern, we note what the well-known artifacts and bias in self-distillation is:
>  * Models reinforce their incorrect judgments
>  * Tendency to select "easy" samples or features
>  * Results in data distribution shifts and inconsistent annotations
>
> **We want to clarify that our training set construction is NOT a self-distillation process.** Though our training data is generated by SDXL, this differs fundamentally from self-distillation where a model's outputs are directly used to train the same or a similar model (potentially amplifying existing biases). In our approach:
>
> * Uses MLLM as an independent external judge for data filtering
> * Applies strict quality control based on semantic understanding
> * Validates the filtering criteria through expriments
>
> **This external validation and filtering mechanism helps prevent the reinforcement of model biases.**
>
> ------- Details are as follows.-----------
>
> To elaborate on why our approach avoids self-distillation concerns:
> * **Independent External Judge for Data Filtering: Prevention of Self-Distillation Artifacts**
>   * We use MLLM as an independent external judge for filtering, not SDXL itself
>   * MLLM's evaluation criteria are based on semantic understanding, different from SDXL's generation process
>   * This independent filtering mechanism helps break potential self-reinforcement loops
>   * The reliability of MLLMs we used have benn verified by human evaluation. (see Line 355-358)
>  * **Strict Quality Control: Preventing Self-Distillation Artifacts and Bias**
>    * We implement strict quality control to exclude low-quality samples (details shown in Appendix D.3)
>      * Set high thresholds for text-image alignment scores $S > 0.8$
>      * Require large score gaps between unrelated text-image pairs $S_{match} > S_{mismatch}$
>      * Small scoring differences for semantically similar text-image pairs $|S_{raw} - S_{similar}| < 0.1$
>     * Ensuring data source free from artifacts and bias
> * **Validates the Filtering Criteria: Evidence Against Self-Distillation Bias**
>   * If self-distillation artifacts were the main issue, then using stricter MLLM filtering would not help, as the fundamental bias comes from SDXL generation itself.
>   * However, our experiments show that: (Details in Response 3 of Reviwer 2)
>     * stricter MLLM filtering (threshold 0.8) significantly improves performance compared to looser filtering (threshold 0.6)
>     * relax MLLM filtering to expand data by lowering MLLM thresholds
>        * No performance improvement
>      * Best results achieved with strictest filtering criteria
>
>  * Conclusions
>    * The collection of training data is not a process of self-distillation.
>    * Performance decline not due to artifacts/bias
>    * External Judge and Strict filtering ensures data quality
>    * Human Evaluation confirms reliability of evaluation conclusions

---

> ### Author Response · Authors · 2024-12-01
>
> **Question 2**: The paper suggests that fine-tuning improves alignment at the word level rather than
> semantic variants, but doesn’t provide sufficient discussion or analysis to support this claim.
>
> **Response 2**: Thank you for raising this concern. Let us first clarify the concept and then present our comprehensive evidence:
>
> The "alignment" discussed here refers to cross-attention mechanism, specifically how tokens are mapped to spatial regions. Not to be confused with alignment scores. Current mechanism maps tokens to regions without capturing inter-token relationships. **We updated the relative content in Lines 440-446.**
>
> **We also updated the experiments and evidence for token-level correspondance, as shown in Appendix F.4.1 in the revised version.**
>
>
> * **Systematic Methodology to Distinguish Token vs Semantic Understanding**
>    - Used GPT-4 as an independent evaluator
>    - Evaluation criteria:
>      - Token level: Verify presence of specific content words (nouns, verbs, adjectives, etc.)
>      - Semantic level: Assess relationships between elements (already done in Table 3)
>    - Example: For "a dog chasing a cat"
>      - Token check: presence of "dog", "chasing", "cat" independently
>      - Semantic check: correct relationship between these elements
>
> * **Quantitative Evidence**
>    - Table 21: Token-level accuracy improvements
>      - Height category: Token accuracy improved
>    - Table 3 in the paper: Semantic-level degradation
>      - Height category shows:
>        - Overall semantic score (S) decreases
>        - causal effect score decreases
>
> * **Qualitative Analysis**
>    - Figure 16 in the paper: Qualitative analysis
>      - Token success: All elements ("shirt", "mannequin", "dress", "hanger", etc.) present
>      - Semantic failure: Incorrect associations between patterns and clothing items
>    - Additional examples in Figure 17
>
> * **Analysis**
>    - improvement in token presence but degradation in relationships
>
> Table 21 (in the revised version, the right table) : Ratio of Tokens Successfully Generated in Images
> | Model    | T_a    | T_pv   | T_pi   |
> |----------|--------|--------|--------|
> | SDXL     | 0.881  | 0.660  | 0.861  |
> | FT SDXL  | 0.886  | 0.662  | 0.866  |

---

> ### Author Response · Authors · 2024-12-01
>
> **Question 3**: How does the choice of filtering training data using MMLM potentially affect the evaluation results, particularly for categories with limited samples?
>
> **Response 3**: Thank you for this important question. **The impact of MLLM-based data filtering manifests in two aspects: data scale and data quality.**
>
>  * **Data Scale Impact**
>    * High filtering standards can lead to insufficient data, but large initial data pools can still produce enough high-quality samples
>  * **Data Quality Impact**
>    * Lowering filtering standards leads to performance degradation, even with increased training data.
>  * Conclusion
>    * The filtering standards can't be lowered. We can enlarge the initial data pools to relieve the impact of strict filtering standars.
>    * For categories with limited samples, we can generate multiple images for each text to enlarge the initial data pools.
>
> **We updated detailed experiment and analysis in Appendix F.3.1 in the revised version.**
>
> ------------------- More Details are as follows. -------------------------
>    * Data Scale Impact
>      * Analyse Results in Table 3 in the paper
>        * The inital quality of samples from Absolute location category is lower than that from Direction and Height category.
>        * The significantly larger number of samples compensates for this deficiency, leading to the improvement of finetuning.
>      * Conclusion
>        * Sufficient initial data size ensures enough high-quality samples remain after strict filtering
>    * Data Quality Impact
>      * Add Experiment -- Finetune Stable Diffusion XL on the training data filtered by different filtering choices
>        * Filtering Choices
>          * Alignment score thresholds: 0.8 and 0.6.
>          * Filtering Constrains: strict VS. no
>        * Results (See Table 13, Table 14, copied below for convenience)
>          * Lowering filtering standards degrades model performance
>          * Performance decreases observed across all categories without "0.8+strict filtering"
>          * For categories with limited data, although performance decreases, ”0.8+strict filtering” results in smaller performance drops.
>      * Conclusion
>        * Strict filtering leads to better results even with limited data
>
> Table 3 (in the paper pdf, extracted key metrics from Table 3 for convenient comparison): Fine-tuned SD XL Results
> | Category | Inital Scale (data pool) | Inital Quality (alignment score) | Finetuned Model Results (alignent & SemVarEffect) |
> |---|---|---|---|
> | Absolute location | 1.7k | 0.64 | improve |
> | Direction | 0.3k | 0.79 | worsen |
> | Height | 0.2k | 0.77 | worsen |
>
> Table 13 (in the revised pdf, the right table): Fine-tuned Results on Absolute_Location with different data filtering constrains
> | Filtering Constrains  | $\bar{S}(\uparrow)$ | $\gamma_{w}(\uparrow)$ | $\gamma_{wo}(\downarrow)$ | $\kappa(\uparrow)$ |
> |---------------------------|-------|-------|--------|-------|
> | zeroshot (baseline)       | 0.64  | 0.29  | 0.34   | -0.05 |
> | **0.8 + strict filtering**    | **0.65**  | **0.42**  | **0.35**   | **0.07**  |
> | 0.8 + no filtering        | 0.54  | 0.25  | 0.31   | -0.05 |
> | 0.6 + strict filtering    | 0.56  | 0.36  | 0.38   | -0.02 |
> | 0.8 + no filtering        | 0.57  | 0.31  | 0.40   | -0.09 |
>
> Table 14 (in the revised pdf, the right table): Fine-tuned Results on Direction with different data filtering constrains
> | Filtering Constrains  | $\bar{S}(\uparrow)$ | $\gamma_{w}(\uparrow)$ | $\gamma_{wo}(\downarrow)$ | $\kappa(\uparrow)$ |
> |---------------------------|-------|-------|--------|-------|
> | zeroshot (baseline)       | 0.79  | 0.20  | 0.15   | 0.05  |
> | **0.8 + strict filtering**    | **0.77**  | **0.25**  | **0.24**   | **0.02**  |
> | 0.8 + no filtering        | 0.75  | 0.24  | 0.24   | 0.00  |
> | 0.6 + strict filtering    | 0.75  | 0.19  | 0.28   | -0.09 |
> | 0.8 + no filtering        | 0.74  | 0.20  | 0.27   | -0.07 |

---

> ### Author Response · Authors · 2024-12-01
>
> **Question 4**: With the 4 metics you had mentioned, which one is the main metric that would make sense
> quickly for readers? It is a little hard to follow with different metrics presented at different parts,
> which I can imagine is a little confusing especially for readers who are looking for a single metric
> for benchmarking.
>
> **Response 4**: Thank you for this important question. **Our proposed $\kappa$ (SemVar-Effect) is the main metric, serving as a complementary measure to alignment $\bar{S}$ by validating whether high alignment scores truly reflect semantic understanding.** $\gamma_{w/}$ and $\gamma_{wo}$ are only used to explain why $\kappa$ increases or decreases.
>    * How to use the metric
>      * $\kappa$ evaluates semantic information transfer from inputs to outputs, rather than generation quality
>        * A model may achieve high $\kappa$ scores when maintaining consistent semantic variations, even if those variations are incorrect
>      * $\kappa$ validates the reliability of high alignment scores $\bar{S}$
>        * High $\bar{S}$ + Low $\kappa$: Indicates limited semantic understanding
>        * High $\bar{S}$ + High $\kappa$: Indicates effective semantic understanding
>
> **We updated these analysis in Appendix F.2.1 in the revised version.**

---

> > ### Comment · Reviewer_nmou · 2024-12-03
> >
> > I appreciate the good rebuttal, I believe that my questions were properly answered.
> >
> > Is there any plan to open-source this benchmark? This can be valuable contribution to the field.
> >
> > Based on the good-faith assumption that this would be open, and based on the rebuttal response, I am raising the score to 7.

---

> ### Author Response · Authors · 2024-12-03
>
> Thank you for your positive feedback and valuable suggestion. Yes, we absolutely plan to open-source our benchmark. To demonstrate our commitment, we have already started preparing:
>
> * The complete dataset, including:
>   * Training set and test set
>   * Small training set for anti-commonsense and plausible scenarios, including human-crafted/human-selected images (created during rebuttal)
> * Implementation resources:
>   * Scripts for dataset usage
>   * Evaluation scripts
>   * Detailed documentation and usage guidelines
>
> We will release everything through a public GitHub repository. A preliminary version of the repository can be previewed at [ https://anonymous.4open.science/r/dataset-657D/ ].  We believe making this benchmark publicly available will help advance research in relationship understanding in T2I models. We appreciate your recognition of the potential value of this benchmark to the community and thank you for improving the score.

---

### Official Review · Reviewer_UuMu · 2024-11-07

**Soundness:** 2
**Presentation:** 1
**Contribution:** 2
**Rating:** 5
**Confidence:** 3

**Summary:**

The paper presents a novel metric, SemVarEffect, and a corresponding benchmark, SemVarBench, designed to evaluate the causality between semantic variations in input text and output images for text-to-image (T2I) synthesis models. The authors identify a gap in current evaluation methods, which often rely on indirect metrics like text-image similarity, failing to adequately assess how models handle semantic changes due to word order. SemVarEffect measures the average causal effect of input semantic variations on output images, while SemVarBench provides a dataset with controlled linguistic permutations to test model performance. The experiments reveal that even state-of-the-art models struggle with semantic variations, indicating room for improvement in understanding and generating accurate visual representations from textual instructions.

**Strengths:**

- The paper directly addresses a critical issue in the evaluation of T2I models, which is the accurate interpretation of human instructions and evaluate the finegrained instruction following ability of the models.

**Weaknesses:**

1. The paper suffers from significant presentation issues, making the proposed method difficult to comprehend. The overall structure of the methodology section requires refinement:
   - Figure 2, which is crucial to understanding the proposed method, is challenging to interpret.
   - In Section 2.2, the term "localized change" needs clarification.
   - The definition of "integrated visual semantic variations" is also unclear.
   - Line 278 uses the phrase "evoke vivid mental images," but its meaning is ambiguous in the given context.
   - Table 2 includes numerous annotations, but offers no explanation of what the different scores represent.

2. In my opinion, the method appears overly complex. The core idea is straightforward: we anticipate that semantic alignment of generated images should be strong if the inputs are semantically identical, and less aligned if the inputs differ semantically. Thus, a simple metric like S(I_a, I_pu) - S(I_a, I_pi) could be employed. What advantages does the proposed causal relationship provide over this simple baseline? I would appreciate deeper insight and explanations on this matter.

3. Regarding model-based evaluation, there is no discussion on whether the MLLM used to compute the alignment score is effective for this task, nor is there any human evaluation. This omission raises concerns about the reliability and trustworthiness of the scores presented.

**Questions:**

See weakness part.

---

> ### Author Response · Authors · 2024-12-01
>
> We sincerely apologize for this oversight. Though we successfully submitted a revised version several days ago, we failed to verify if the correct version was uploaded. As we cannot locate the revised version in the system now, we have provided an anonymous link [
>  https://anonymous.4open.science/r/submission-41BF/EVALUATING_SEMANTIC_VARIATION_IN_TEXT-TO-IMAGE_SYNTHESIS.pdf ] containing our complete revised manuscript. We deeply appreciate your understanding and time.
>
> **Question 1**: The paper suffers from significant presentation issues, making the proposed method difficult to comprehend. The overall structure of the methodology section requires refinement:
> 1. Figure 2, which is crucial to understanding the proposed method, is challenging to interpret.
> 2. In Section 2.2, the term "localized change" needs clarification.
> 3. The definition of "integrated visual semantic variations" is also unclear.
> 4. Line 278 uses the phrase "evoke vivid mental images", but its meaning is ambiguous in the given context.
> 5. Table 2 includes numerous annotations, but offers no explanation of what the different scores represent.
>
> **Response 1**: We appreciate these detailed suggestions for improving the clarity of our presentation. We have thoroughly revised the methodology section to address each point.
>
> 1. *Figure 2 Clarification*.
>    * Our evaluation framework consists of three key components:
>      * Step 1 - Input Variations: Implementation of semantic change and maintenance interventions.
>      * Step 2 - Visual Semantic Evaluation: Two parallel processes assessing semantic change intervention and semantic maintenance intervention mentioned in Step 1. These evaluations produce respective visual semantic variation values.
>      * Step 3 - Causal Effect Calculation: Calculation of SemVarEffect by comparing the difference between the two visual semantic variation values, quantifying the causal effect of semantic variation.
>     * **In this revision, we revised Figure 2 and its caption in Line 70-76 to enhance its clarity and interpretability.**
>
> 2. *Localized Change*.
>    * "localized change" refers to decomposing complex semantic transformations into minimal discrete steps.
>      * Similar to how calculus breaks down continuous changes into infinitesimal differences.
>      * Though semantic variation are discrete, viewing them as a sequence of minimal steps justifies our focus on initial and final states.
>    * **In this revision, we deleted the phrase "localized change" in Line 132-136 and use "minimal discrete steps" to uniformly represent this process for clarity.**
>
> 3. *Integrated Visual Semantic Variations*.
>    * Thank you very much for reminding us that we did not clearly distinguish the summation process of Line 132 and Line 134 & 140.
>    * The phrase "integrated visual semantic variations" in Line 140 refers to sum the visual semantic variations from multiple text-image pairs to comprehensively measure these variations.
>    * **In this revision, we revised the "integrated visual semantic variations" in Line 140 to "the summation of visual semantic variations" for clarity.**
>
> 4. *Evoke Vivid Mental Images*.
>    * The text itself should describe something humans can visualize, regardless of whether models can successfully generate it.
>    * To avoid potential ambiguity, We elaborated this characteristic in Appendix.
>      * We decomposed it into four concrete, assessable standards in Appendix C.2 (Line 1130-1144):
>        * Visually Depicted Element
>        * Static/Multiple Exposure Scene
>        * Moderate Level of Details
>        * Quantifiable Comparison
>      * For detailed implementation, we provide examples violating these standards in Appendix C.2 Table 5, transforming subjective principles into objective annotation guidelines.
>     * **In this revision, we revised the phrase "evoke vivid mental images" to the more precise "describe something humans can visualize. If text is unimaginable to humans, it cannot be meaningfully visualized by AI models" in Line 210-211.**
>
>
> 5. *Table 2 Notation*.
>    * All notations in Table 2 have been introduced in Section 4.1 EXPERIMENTAL SETUP.
>      * Model Name abbreviations: Table 1
>      * Evaluation metrics and their meanings: Line 299-302 (Detailed explanations of metrics: Appendix B.4 and B.5)
>    * **In this version, we updated Table 2's caption in Line 336-340 to explicitly explain all metrics and notations to improve clarity.**
>    * For clarity, here are these metrics and their meanings:
>      * $\bar{S_{ii}}$ ($\uparrow$): text-image alignment score. $\bar{S_{ii}}=\frac{1}{\left|K\right|}\sum_{i \in K}S(T_i, I_i)$, where $K=\{a, pv, pi\}$.
>      * $\kappa$ ($\uparrow$): SemVar-Effect, measuring input-to-output variation contribution.
>      * $\gamma_{w}^I$ ($\uparrow$): visual semantic variation under semantic change.
>      * $\gamma_{wo}^I$ ($\downarrow$): visual semantic variation under semantic maintenance.

---

> ### Author Response · Authors · 2024-12-01
>
> **Question 2**: In my opinion, the method appears overly complex. The core idea is straightforward: we anticipate that semantic alignment of generated images should be strong if the inputs are semantically identical, and less aligned if the inputs differ semantically. Thus, a simple metric like $S(I_a, I_{pv}) - S(I_a, I_{pi})$ could be employed. What advantages does the proposed causal relationship provide over this simple baseline? I would appreciate deeper insight and explanations on this matter.
>
>
> **Response 2**: Thank you for your concern about the complexity.
>
>  * **Complexity**.
>    * **Our core idea of measuring semantic variations is straightforward, that is evaluating how input semantic variations are reflected in outputs.**
>    * **This complexity stems from the fundamental nature of the problem rather than arbitrary design choices.**
>      * Establish causal relationships between input semantic variations (as the cause) and output variations (as the effect)
>      * Quantify the degree of semantic transfer across modalities
>
>  * **Shortcomings of the Baseline Metric**.
>    * **It cannot differentiate semantic errors from valid creative variations due to its sole focus on visual similarity**
>      * Cannot differentiate
>          * True semantic errors
>            * Example: "three dogs" vs "one dog" might look visually similar but represents a semantic error
>          * Acceptable variations
>            * Example: For "a dog chasing a squirrel", "on the moon" and "in the forest" represent acceptable creative variations
>          * Unspecified variations
>            * Example: For "a dog", "a Labrador Retriever" and "a Chihuahua" are both dogs, even though they are not the same in the breed.
>
>    * **It lacks standards for meaningful comparison and interpretation.**
>      * Unable to determine what constitutes good performance
>      * Cannot fairly compare different types of semantic changes (e.g., object missing vs relationship variation)
>
>
>  * **Advantages of Our Method**. Our method provides two key advantages over the simple metric:
>    * **Theoretical Support and Study.**
>      * Grounded in equivariant mapping theory requiring corresponding input-output changes [1][2].
>      * Quantify the proportion of output variations caused by input semantic changes by employing causal framework
>      * Provides theoretically justified bounds (0.5-1.0) for performance evaluation:
>        * scores > 0.5 indicate semantically meaningful variations
>        * 0 < scores < 0.5 indicate variations insensitive to semantics
>        * scores ≤ 0 indicate random variations
>    * **Necessity of Text-Guided Meirtc for T2I Image Quality.**
>      * Traditional metrics focusing solely on visual feature similarity are insufficient for T2I evaluation
>        * Image-only metrics ignore text-image semantic alignment, such as FID and IS,  which are usually used in T2I synthesis in the previous years [3][4]
>        * Cross-modal nature of T2I requires joint consideration of both text and image variations
>      * Enables fine-grained quality assessment specific to T2I generation
>        * Semantic errors (e.g., generating "three dogs" for "a dog")
>        * Valid creative variations (e.g., for "a dog chasing a squirrel", images have different scenes: "on the moon" vs "in the forest")
>        * Acceptable variations in unspecified attributes (e.g., dog breed)
>
> [1] Hartford J, Graham D, Leyton-Brown K, et al. Deep models of interactions across sets[C]//ICML, 2018.
>
> [2] Blum-Smith B, Villar S. Machine learning and invariant theory[J]. arXiv preprint arXiv:2209.14991, 2023
>
> [3] Jayasumana S, Ramalingam S, Veit A, et al. Rethinking fid: Towards a better evaluation metric for image generation[C]//CVPR. 2024.
>
> [4] Huang K, Sun K, Xie E, et al. T2i-compbench: A comprehensive benchmark for open-world compositional text-to-image generation[J]. NeurIPS, 2023.

---

> ### Author Response · Authors · 2024-12-01
>
> **Question 3**: Regarding model-based evaluation, there is no discussion on whether the MLLM used to compute the alignment score is effective for this task, nor is there any human evaluation. This omission raises concerns about the reliability and trustworthiness of the scores presented.
>
> **Response 3**: We appreciate the reviewer’s concern about the reliability of evaluation. **Our choice of MLLMs is supported by both literature and empirical validation.**
>
> **We updated the human evaluation in Line 299-302, Line 355-358 and Appendix E.2.**
>
> Details are as follows.
>
>  * **Literature Evidence**
>    * Recent studies show GPT-4v/o has strong correlation with human judgments in T2I evaluation [1][2][3][4]
>    * Gemini-1.5-Pro and Claude-3.5-Sonnet rank consistently below GPT-4v/o but outperform traditional metrics (CLIP, PickScore, etc.)
>  * **Human Evaluation Results**
>     * Experiment Setting
>        * Testsets: 80 stratified samples from the testset
>        * Models: 4 advanced T2I models
>        * Evaluators: 3 raters evaluating all samples
>        * Total cost: \$ $200$
>    * Results & Conclusion
>      * Consistent performance trends between human and MLLM evaluations
>      * Moderate correlation coefficients with human judgments (Pearson's, Spearman's, Cohen's Kappa)
>    * Our human evaluation study demonstrates MLLM's reliability
>
> Table 10 (in revised version): Evaluation human rating results of different T2I models in understanding semantic variations.
> | T2I Models      | $\bar{S}(\uparrow)$ | $\gamma_{w}(\uparrow)$ | $\gamma_{wo}(\downarrow)$ | $\kappa(\uparrow)$ |
> |------------|-------|--------|---------|-------|
> | MidJ V6    | 0.59  | 0.37   | 0.49    | -0.12 |
> | DALL-E 3   | 0.63  | 0.53   | 0.50    | 0.03  |
> | CogV3-Plus | 0.69  | 0.52   | 0.33    | 0.19  |
> | Ideogram 2 | 0.74  | 0.50   | 0.31    | 0.19  |
>
>
> Table 11 (in revised version): Correlation coefficients between GPT-4o/v, Claude-3.5-Sonnet, Gemini-Pro-1.5 and human evaluations on SemVarEffect of CogView3-Plus.
> | Evaluators      | $\rho(\uparrow)$ | $\phi(\uparrow)$ | $\kappa_{cohen}(\uparrow)$ |
> |-----------------|-------|-------|-----------|
> | GPT-4o          | 0.54  | 0.53  | 0.53      |
> | GPT-4v          | 0.50  | 0.51  | 0.54      |
> | Claude-3.5-Sonnet| 0.42  | 0.37  | 0.37      |
> | Gemini-Pro-1.5  | 0.27  | 0.11  | 0.23      |
>
>
> [1] Ku M, Jiang D, Wei C, et al. Viescore: Towards explainable metrics for conditional image synthesis evaluation[J]. ACL, 2024.
>
> [2] Chen Z, Du Y, Wen Z, et al. MJ-Bench: Is Your Multimodal Reward Model Really a Good Judge for Text-to-Image Generation?[J]. arXiv preprint arXiv:2407.04842, 2024.
>
> [3] Lin Z, Pathak D, Li B, et al. Evaluating text-to-visual generation with image-to-text generation[C]//ECCV, 2024.
>
> [4] Ji P, Liu J. TlTScore: Towards Long-Tail Effects in Text-to-Visual Evaluation with Generative Foundation Models[C]//CVPR, 2024.

---

> ### Author Response · Authors · 2024-12-03
>
> We appreciate your time in reviewing our work. We understand that your schedule may be quite busy.  As the discussion phase draws close, we kindly request your attention to our responses. We have carefully addressed each of your concerns and clarified the misunderstandings in the review, including **clarity concerns**, **complexity concerns** and **evaluation reliability**.
>
> Regarding the **method complexity**, **as you expressed interest in** understanding the advantages of our causal framework over the simple baseline metric, we have provided a detailed analysis (please see our response above) focusing on:
>
> * Theoretical Support and Study
>   * Quantify the proportion of output variations caused by input semantic changes
>   * Standards for meaningful comparison and interpretation
> * Necessity of Text-Guided Metric for T2I Image Quality
>   * Cross-modal nature of T2I requires joint consideration of both text and image variations
>   * Enables differentiation between semantic errors and valid creative variations
>
> **Would you be particularly interested in reviewing this part of our response?**
>
> We would greatly appreciate your feedback on whether our explanations, especially regarding the method's complexity and necessity, have resolved these points. If our responses are satisfactory, we welcome your further consideration.

---

### Author Response · Authors · 2024-12-02

We sincerely thank all the reviewers for their constructive and thoughtful feedback. The reviewers' comments have helped us significantly improve our work. We have carefully addressed each concern and made substantial revisions to enhance the clarity and rigor of our paper.

The revised manuscript with tracked changes is available at [ https://anonymous.4open.science/r/submission-41BF/EVALUATING_SEMANTIC_VARIATION_IN_TEXT-TO-IMAGE_SYNTHESIS.pdf ], with all revisions highlighted in blue. In response to the reviewers' comments, we have addressed each question individually. Additionally, we provide a summary of the revisions:

* **Enhanced Framework Illustration and Methodology**
  - Improved Figure 2 and its caption to better demonstrate our three-component framework (Line 54-76)
  - Clarified SemVarEffect derivation and calculation process (Section 2.2, Line 128-140)
  - Elaborated on SemVarEffect's complementary role to traditional alignment metrics (Appendix F.2.1, Line 1588-1594)
  - Clarified the definition of Visualizability to eliminate ambiguity (Section 3, Line 210-211)

* **Expanded Experimental Details and Results**
  - Enhanced Table 2 with clearer notation explanations (Line 336-340)
  - Added human evaluation settings (Section 4.1, Line 299-302) and results (Section 4.2, Line 355-358)
  - Provided comprehensive human evaluation analysis with detailed results (Appendix E.2, Line 1477-1487, Table 10-11)

* **Enhanced Analysis of Model Behavior**
  - Refined analysis of token-level versus semantic-level improvements in fine-tuned models (Section 4.3, Line 440-446)
  - Provided experimental validation of token-level improvement hypothesis (Appendix F.4.1, Line 1784-1802, Table 21, Figure 16-17)
  - Explored the relationship between semantic variation understanding capability and training data imbalance (Section 4.3, Line 486-498; Appendix F.3.4, Line 1754-1771, Table 19-20)
  - Conducted additional analyses across multiple scenarios:
    - Impact of data filtering on fine-tuning performance (Appendix F.3.1, Line 1597-1616, Table 13-14)
    - Commonsense bias in fine-tuned models (Appendix F.3.2, Line 1618-1697, Table 15-17, Figure 15)
    - Model performance on relational sentences in plausible scenarios (Appendix F.3.3, Line 1718-1752, Table 18)
    - Shortcut learning phenomenon in data-limited categories (Appendix F.4.2, Line 1847-1861)

Thank you once again for your valuable time and effort in reviewing our paper. We hope these revisions address your concerns and improve the clarity and quality of our work.

---

### Comment · Area_Chair_fswv · 2024-12-02
**Dec 2 - last day for reviewers' questions**

Dear reviewers,

This is a kind reminder that Dec 2 marks the last day for reviewers to ask questions. As the paper received diverse reviews and the authors have provided the rebuttal, can you check the authors' responses and confirm whether you have any remaining concerns? Please take this chance to ask questions to help us make the final decision.

Thank you,

AC

---

### Meta-Review · Area_Chair_fswv · 2024-12-24

**Metareview:**

Summary: This paper focused on evaluating the causality between textual semantic variations and visual semantic variations in text-to-image synthesis task. This paper pointed out that existing T2I models often struggle to capture semantic variations caused by different word orders, and further proposed a novel metric called PermuteEffect and a benchmark named PermuteBench for better T2I evaluations. The key findings are that cross-modal alignment in UNet or Transformers is important in semantic variations.

Strengths: (1) The idea of interpreting and evaluating Text-to-Image qualities via causal perspective is interesting and novel. (2) An metric and evaluation benchmark with 11.4k examples (10.8k training data and 0.6k test data). (3) A first systematic study of semantic variations in T2I synthesis is conducted.

Remaining concerns after discussion are (1) the lack of comparison with different benchmarks and metrics, (2) rationale behind the metric definition, and (3) the balance between the method’s complexity and its practical benefits.

This paper received diverse ratings 8, 6, 5, 5 as final ratings. The AC checked reviewers' comments, authors' responses, and the revised paper. For the remaining concerns (1), the text-image alignment score metric is compared; for concern (2), Sec. 2.3 has explained the rationale behind the metric definition via causal perspective; for (3), the authors replied that "this complexity stems from the fundamental nature of the problem rather than arbitrary design choices", which AC agreed as the method come from the causality nature. Therefore, the AC recommend the paper as accept and encourages the authors to further revise the paper according to reviewers' comments.

**Additional Comments On Reviewer Discussion:**

The reviewers raised three remaining concerns. See Metareview for details.

---

### Decision · Program_Chairs · 2025-01-22

Accept (Poster)